# BEYOND MODEL SCALING: TEST-TIME INTERVENTION FOR EFFICIENT DEEP REASONING

## ABSTRACT

Large Reasoning Models (LRMs) excel at multi-step reasoning but often suffer from inefficient reasoning processes like overthinking and overshoot, where excessive or misdirected reasoning increases computational cost and degrades performance. Existing efficient reasoning methods operate in a closed-loop manner, lacking mechanisms for external intervention to guide the reasoning process. To address this, we propose Think-with-Me, a novel test-time interactive reasoning paradigm that introduces external feedback intervention into the reasoning process. Our key insights are that transitional conjunctions serve as natural points for intervention, signaling phases of self-validation or exploration and using transitional words appropriately to prolong the reasoning enhances performance, while excessive use affects performance. Building on these insights, Think-with-Me pauses reasoning at these points for external feedback, adaptively extending or terminating reasoning to reduce redundancy while preserving accuracy. The feedback is generated via a multi-criteria evaluation (rationality and completeness) and comes from either human or LLM proxies. We train the target model using Group Relative Policy Optimization (GRPO) to adapt to this interactive mode. Experiments show that Think-with-Me achieves a superior balance between accuracy and reasoning length under limited context windows. On AIME24, Think-with-Me outperforms QwQ-32B by 7.19% in accuracy while reducing average reasoning length by 81% under an 8K window. The paradigm also benefits security and creative tasks.

## 1 INTRODUCTION

Large Language Models (LLMs) (Achiam et al., 2023) have become the foundation of modern natural language processing. While effective across various tasks, they often struggle with complex reasoning that requires multi-step deduction and intermediate conclusions (Yang et al., 2024c). To address this limitation, Large Reasoning Models (LRMs)(Xu et al., 2025a) have emerged, such as OpenAI o1 (Xu et al., 2025c) and DeepSeek-R1 (DeepSeek-AI et al., 2025). These models are equipped with explicit thinking mechanisms that enable them to break down complex problems into logical steps. This design significantly improves performance on reasoning-intensive tasks, such as challenging mathematics (Art of Problem Solving, 2024) and formal logic (Yao et al., 2024).

*Unfortunately*, existing LRMs rely on fixed-length budgets (Aggarwal and Welleck, 2025) or greedy continuation of reasoning chains, lacking adaptive control over reasoning depth or direction (Liu et al., 2025c). This leads to two failure modes: *1) Overthinking.* Lacking a reliable stopping criterion, the model keeps extending the chain with repetitive steps, inflating overall token cost and latency (Chen et al., 2025b; Yang et al., 2025a). *2) Overshoot.* After reaching a valid intermediate solution, continued reasoning may introduce contradictions or unwarranted revisions that overwrite the correct answer and produce an incorrect final output He et al. (2025); Cuadron et al. (2025). These failures motivate on-the-fly control of reasoning depth and direction while preserving answer accuracy.

Recent works on reasoning efficiency can be categorized into two paradigms: **single-model optimization** and **multi-model collaboration**. Single-model optimization methods enhance efficiency within one model but rely on internal numerical signals that lack semantic nuance control (Yang et al., 2025b; Chen et al., 2025a). Multi-model collaboration improves accuracy through model collaboration but introduces coordination overhead (Pan et al., 2025; Akhauri et al., 2025). While effective to some extent, these methods internally determines how far to reason, whether to stop, and

whether it is on the right track. This self-referential control makes them hard to arrest redundant exploration and can let local mistakes snowball into final errors, under the known cognitive biases of LLMs (Echterhoff et al., 2024; Yu et al., 2024). Although external feedback, from many sources, has been validated as an effective strategy to mitigate models' internal biases (Shinn et al., 2023), existing works (Luo et al., 2024; Shinn et al., 2023) misalign the reasoning behaviors of LRMs. As we demonstrate empirically in Appendix B.1, since these strategies, often applied post-hoc or without structural alignment to the reasoning process, applying previous work on LRMs can not improve the efficiency of reasoning.

To address these limitations, we introduce Think-with-Me, a test-time intervention framework for efficient reasoning of LRMs. Our design builds on two key observations: (**Observation 1**) transitional conjunctions naturally segment the reasoning process and often mark phases of self-validation or exploration; (**Observation 2**) judicious use of such segments improves task accuracy, whereas overuse harms it. Building on these findings, we design Think-with-Me as an interactive reasoning paradigm that treats the generation of specific transitional conjunctions as intervention points and integrates external feedback based on multi-criteria from either LLM proxies or humans to adaptively terminate or extend the reasoning process as needed, reducing the redundant reasoning that leads to overthinking and even overshoot. This strategy reduces the overall token sequences while maintaining the task accuracy or even outperforming the strong baselines, achieving efficient reasoning. Finally, to ensure the target reasoning model reliably acts upon provided external feedback, we train the reasoning model through the Group Relative Policy Optimization (GRPO) (DeepSeek-AI et al., 2025) under the proposed interactive reasoning mode. Our main contributions are as follows:

- **Adaptive external-feedback intervention framework.** We propose Think-with-Me, a framework that incorporates independent external feedback to intervene in the reasoning process of LRMs. By adaptively adjusting or terminating the reasoning process based on external feedback, our method reduces redundant thinking, improves accuracy, and achieves efficient deep reasoning.

- **Empirical discoveries of intervention opportunities in LRM's reasoning.** We provide empirical evidence that transitional conjunctions naturally segment the reasoning and that appropriate use of these markers improves answer accuracy, whereas excessive use degrades it. We treat them as intervention points and introduce fine-grained external feedback to bound overall reasoning length, controlling the use of these conjunctions and balancing the accuracy–length trade-off.

- **Extensive experiments validate the effectiveness of Think-with-Me.** Compared to SOTA methods and model-based baseline, Think-with-Me reduces the overall reasoning length while maintaining comparative task performance. For AIME24@32, Think-with-Me outperforms QwQ-32B 7.19% in accuracy and reduces reasoning length by 81% in 8k window size.

## 2 PROBLEM STATEMENT

Let $\mathcal{LM} : \mathcal{V}^* \to \mathcal{V}$ denote a next-token prediction LLM, where $\mathcal{V}$ denotes the vocabulary set and $\mathcal{V}^*$ represents the space of all possible token sequences over $\mathcal{V}$. Given an input $x := (x_1, \ldots, x_n) \in \mathcal{V}^*$, where each $x_i \in \mathcal{V}$, the LLM predicts the next token in the sequence. A conventional LLM automatically generates a response sequence $y := (y_1, \ldots, y_m) \in \mathcal{V}^*$ by iteratively predicting each response token $y_j = \mathcal{LM}([x, y_{<j}])$ conditioned on the context $x$ and the previously generated tokens $y_{<j} := (y_1, \ldots, y_{j-1})$. For LRMs, the generation process is split into two stages. In the reasoning stage, the model first produces an intermediate sequence of reasoning tokens $r = (r_1, \ldots, r_k) \in \mathcal{V}^*$, and in the subsequent response stage, generates the final output sequence $y = (y_1, \ldots, y_m) \in \mathcal{V}^*$.

However, during reasoning, LLMs often suffer from two related issues: overthinking, engaging in excessive validation and exploration, wasting computational cost and response time; and overshoot, a performance degradation that occurs when the total reasoning steps $k$ exceeds the optimal count $k^*$. We introduce an adaptive strategy that intervenes after selected intermediate steps to optimize the trade-off between accuracy and computational cost. Formally, let $\pi = (\pi_1, \ldots, \pi_k) \in \{0, 1\}^k$ be a binary intervention policy, which is as follows:

$$\pi_i = \begin{cases} 1, & \text{intervene in step } i \\ 0, & \text{no intervention} \end{cases}. \tag{1}$$

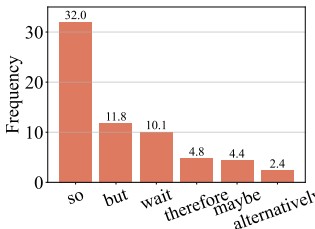 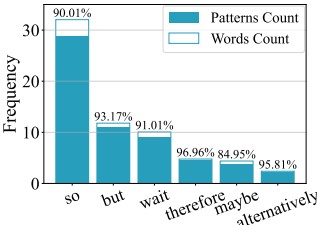 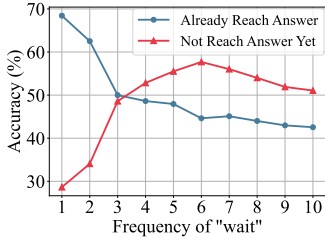

(a) Frequency analysis of common words in the thinking process.
(b) Pattern-word frequency ratio in reasoning traces.
(c) Impact of intermediate thinking correctness on final accuracy.

Figure 1: The key observations: (a) causal and transitional conjunctions are among the most frequent tokens in the reasoning stage. (b) these tokens align with semantic phase boundaries, providing natural checkpoints for intervention. (c) increasing the frequency of "wait" improves the task accuracy before the final answer is reached, but hurts after obtaining the correct answer or self-overuse.

Denote by $y_\pi(x)$ the final output produced under policy $\pi$. We aim to optimize as

$$\min_\pi \; \mathbb{E}_{(x,y^*)\sim\mathcal{D}}\Big[\underbrace{\mathcal{L}\big(y_\pi(x),\, y^*\big)}_{\text{task loss}} + \sum_{i=1}^{k} \pi_i\Big], \qquad (2)$$

where $\mathcal{L}(\cdot,\cdot)$ measures the discrepancy between the model output and ground truth $y^*$, $\sum_i \pi_i$ counts the number of interventions, correlating with total reasoning length. This formulation jointly optimizes accuracy and interaction frequency, maintaining responsiveness in interactive reasoning mode and reducing redundant reasoning that leads to overthinking and even overshoot.

## 3 TEST-TIME INTERVENTION FOR EFFICIENT DEEP REASONING

### 3.1 OBSERVATIONS

In this section, we report the observations that motivate **Think-with-Me**. While previous works (Muennighoff et al., 2025; Liu et al., 2025a) have analyzed LRM reasoning from multiple perspectives, we construct an evidential chain to further identify the triggers of reasoning stages and determine their impact on reasoning accuracy. Our working hypothesis is that *overthinking* stems from the model's tendency toward self-validation and unnecessary exploration—even after reaching a correct solution. We conduct evaluations on 200 uniformly sampled questions of varying difficulty from the MATH500 dataset (Lightman et al., 2024), using DeepSeek-R1-Distill-Qwen-32B (DeepSeek-AI et al., 2025) to generate complete reasoning traces and corresponding answers.

Figure 1a shows that the most frequent tokens in reasoning traces are causal and transitional conjunctions (e.g., "so", "wait"), indicating their universality in reasoning. The experiment details are in Appendix B.2.1. To understand their role, we conduct further experiments and have observations:

**Observation 1: Transitional conjunctions naturally act as stage markers that initiate self-validation or exploratory reasoning.** We summarize common patterns involving these conjunctions and measure their frequencies across traces. The result is in Figure 1b and more experiment details are in Appendix B.2.2. Transition conjunctions appear in a summarizing pattern in over 90% of cases, indicating the pervasiveness of this pattern. Based on the semantics of patterns, causal conjunctions close a sub-step, while transitional conjunctions open the next validation or exploration phase.

**Observation 2: Conjunction-triggered extension improves accuracy before correctness, but harms it after correctness or under self-overuse.** We segment traces at transitional conjunctions identified in Observation 1 and evaluate their effect on final accuracy by prompting Qwen2.5-72B-Instruct (Yang et al., 2024a). As shown in Figure 1c, using transitional conjunctions to *extend* reasoning increases accuracy when the current solution has not reached a correct final answer; once a correct final answer has been reached or when the model itself overuses, such extensions decreases accuracy. The same phenomenon holds on other challenging benchmarks (details in Appendix B.2.4).

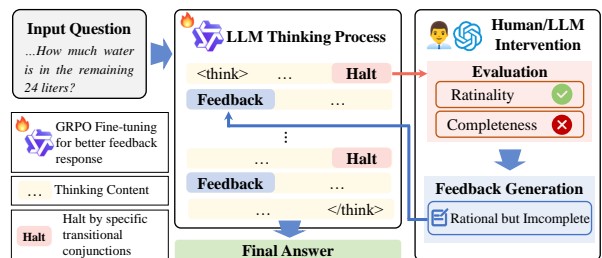 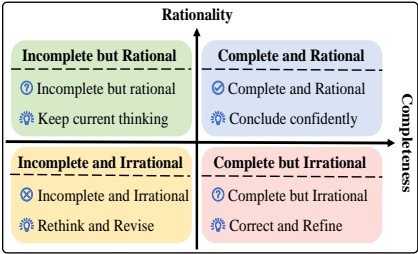

(a) The pipeline of intervene with external feedback in the reasoning process of the target model

(b) The reasoning status and corresponding feedback.

Figure 2: The illustration of Think-with-Me. Given a question, the LRM generates reasoning and is paused until it encounters specific trigger words. An LLM-based evaluator or human then assesses the reasoning for rationality and completeness, and inserts feedback wrapped in tags. The LRM resumes reasoning based on the updated context. This process repeats until the maximum times is achieved or generates </think> to end reasoning and produce the final answer.

## 3.2 THINK-WITH-ME

In this paper, we propose **Think-with-Me**, a novel reasoning paradigm for LRMs that leverages transitional conjunctions as natural intervention points for external feedback, enabling adaptive termination or extension of the ongoing reasoning and balancing accuracy-length trade-off. As illustrated in Figure 2, the target model pauses when it emits specified transitional conjunctions. An evaluator, either a human or an LLM proxy, assesses the current reasoning based on criteria and returns targeted feedback. The feedback is injected into the context, after which the model resumes reasoning. The interaction repeats until reaching the maximum number or the model generates </think> to terminate and then the final answer is produced. The pipeline is summarized in Algorithm 1.

### 3.2.1 INTERVENTION IN THE LLM'S REASONING PROCESS

We design **Think-with-Me**'s intervention mechanism around three core challenges: *when, what, and how to intervene?* First, guided by **Observation 1**, we set the generation of the target model of common transitional conjunctions (e.g., "wait") as intervention points. Second, informed by **Observation 2**, we introduce independent external feedback to direct the target model to either stop or extend: (i) stop reasoning when logically inferring to the final answers, or (ii) guide the model to improve incomplete or irrational reasoning. To formalize the potential effect of intervention, we model the LLM's reasoning process using information theory (Cover, 1999), providing a theoretical understanding.. The base model's uncertainty about target $T$ given context $C$ is quantified by the conditional entropy $H(T \mid C; \theta)$. When intervention feedback $F$ is provided, the uncertainty becomes $H(T \mid C, F; \theta)$. The following proposition characterizes how intervention reduces this uncertainty:

**Proposition 1 (Uncertainty Reduction by Intervention with target-towards external feedback).** *For any intervention feedback $F$, the conditional entropy satisfies:*

$$H(T \mid C, F; \theta) \leq H(T \mid C; \theta)$$

*with equality if and only if $I$ provides no relevant information about $T$. The uncertainty reduction is:*

$$\Delta H = H(T \mid C; \theta) - H(T \mid C, F; \theta) = I(T, F \mid C; \theta) \geq 0.$$

Proposition 1 follows directly from the non-negativity of conditional mutual information. As a direct consequence of information theory, Proposition 1 confirms that additional target-relevant information reduces total reasoning uncertainty. The reduction in uncertainty curtails the redundant self-validation and exploration that arise from uncertainty (Choi et al., 2025), improving reasoning efficiency. Proposition 1 serves as the theoretical explanation to the mechanism of Think-with-Me. As the intervention feedback is designed to provide guidance towards target, it enables $H(T \mid C, F; \theta) > H(T \mid C; \theta)$ and therefore $\Delta H > 0$, reducing the need for extensive self-exploration and verification in the reasoning process, allowing the model to converge more efficiently to obtain conclusions.

### 3.2.2 TARGETED FEEDBACK BASED ON PREDEFINED CRITERIA

To enable effective intervention, we evaluate the current reasoning along two content-agnostic criteria—*Rationality* and *Completeness* (Paul et al., 2024), without consulting other external materials. *Rationality* checks whether steps are logically valid and knowledge-grounded; *Completeness* checks whether the reasoning has reached a final answer in forms and semantics. These criteria induce four thinking statuses, each mapped to rule-based guidance (continue/stop/redirect/refine) for the ongoing reasoning, as shown in the right part in Figure 2. In Figure 2, Think-with-Me supports feedback from either a human-in-the-loop evaluator or an LLM proxy, enabling scalable deployment across diverse tasks. For LLM proxy, we prompt Qwen2.5-72B-Instruct to evaluate the current reasoning and provide feedback for ongoing reasoning. The prompt is constructed based on the rules from REFINER (Paul et al., 2024) and the details of the criteria and prompt template can be found in Appendix C.To enhance the distinction between the generated reasoning content and external feedback, we tag the feedback with special tokens `<reasoning_feedback>` and `</reasoning_feedback>`.

---

**Algorithm 1** Pipeline of Think-with-Me.

---

**Input:** Reasoning-enhanced LLM $\mathcal{M}$, Questions $Q$, Evaluator $\mathcal{E}$, Stop token set $S = \{s_1, \ldots, s_n\}$, Maximum intervention limit $I_{max}$.
1: Initialize set of finished thinking trace $\mathcal{R} \leftarrow \{\}$, set of finished answer part $\mathcal{A} \leftarrow \{\}$
2: **for** each question $q \in Q$ **do**
3:     Initialize generation sequence $gs \leftarrow$ `""`, intervention count $t \leftarrow 0$
4:     **while** `</think>` **not in** $gs$ **do**
5:         Generate until reaching `EOS` or any $s_i$ in $S$: $r_t \leftarrow \mathcal{M}(q \oplus gs)$
6:         Update $gs$: $gs \leftarrow gs \oplus r_t$
7:         **if** `</think>` **not in** $gs$ and $t \leq I_{max}$ **then**
8:             Get external feedback: $f \leftarrow \mathcal{E}(q \oplus gs)$
9:             Construct feedback information: $gf \leftarrow$ `<reasoning_feedback>` $f$ `</reasoning_feedback>`
10:           Update $gs$ with feedback: $gs \leftarrow gs \oplus gf$
11:           $t \leftarrow t + 1$
12:         **else if** `</think>` **not in** $gs$ and $t > I_{max}$ **then**         ▷ Exceed maximum intervention limit
13:           Last generate: $r_t \leftarrow \mathcal{M}(q \oplus$ `</think>`$)$         ▷ Insert `</think>` to force-exit thinking
14:           Update gs: $gs \leftarrow gs \oplus r_t$
15:     Split $gs$ by `</think>`: $gs_{list} \leftarrow \text{Split}(gs,$ `</think>`$)$
16:     Add $gs_{list}[1], gs_{list}[2]$ to $\mathcal{R}, \mathcal{A}$ respectively
**Output:** Thinking trace set $\mathcal{R}$, final answer set $\mathcal{A}$

---

### 3.2.3 TRAINING WITH GROUP RELATIVE POLICY OPTIMIZATION

We finetune the target reasoning model $\mathcal{LM}_\theta$ (policy $\pi_\theta$) to reliably react to external feedback. Starting from a pretrained model, we fine-tune using a reinforcement learning objective with Group Relative Policy Optimization (GRPO) (Shao et al., 2024; DeepSeek-AI et al., 2025). For efficient adaptation, we finetune a LoRA adapter with the frozen backbone model. During inference, it is merged into the backbone model $\mathcal{LM}_\theta^{\text{merge}}$ for efficient decoding without runtime overhead.

**Group Relative Advantage Estimation.** For each query $x$, we sample a group of $G$ candidate outputs $o_1, o_2, \ldots, o_G$ from the old policy $\pi_{\theta_{\text{old}}}$, score them using the reward function in Eqn. 3, and normalize the rewards: $\tilde{r}_i = \frac{r_i - \text{mean}(r)}{\text{std}(r)}$. These normalized rewards $\tilde{r}_i$ are used as the advantage for all tokens in the corresponding output $o_i$. The policy is updated by maximizing the learning objective.

**Remark.** Unlike standard GRPO, each $o_g$ is sampled *through* the proposed interactive reasoning paradigm to match the inference mode. Given a predefined stop token set $S$, each complete output $o_g$ is generated through multiple rounds of thinking intervention, denoted as $o_g = [t_1, f_1, \cdots, t_k, f_k, \cdots]$, where $t_i$ is the $i$-th thinking segment generated by the model and $f_i$ is the corresponding external feedback. Each reasoning segment $t_i = (t_{i,1}, \cdots, t_{i,s'})$ is sampled from the model $\mathcal{LM}_\theta$, conditioned on all previous interactions: $t_i \sim \mathcal{LM}_\theta([x, t_1, f_1, \cdots, t_{i-1}, f_{i-1}])$ and terminates at `</think>` or maximal interaction limit. This enhances the adaptability of the target model to the interactive reasoning mode.

**Reward Function Design.** Our reward function combines three components: a correctness reward $r_c$, a format reward $r_f$ and a length reward $r_l$. Formally, for a given question $x_i$ and corresponding

true answer $y_i$, one sampled complete output $o_k$ where $k \in \{1, 2, \ldots, G\}$, the total reward is:

$$r(o_k, y_i) = r_c(o_k, y_i) + r_f(o_k, y_i) + r_l(o_k, y_i). \tag{3}$$

The format reward $r_f$ encourages the model to maintain a correct response format that the generated output must contain feedback in the form of `<reasoning_feedback>` $f$ `</reasoning_feedback>` between the reasoning process wrapped with `<think>` and `</think>`. Specifically, $r_f$ is defined as:

$$r_f(o_k, y_i) = \begin{cases} +1, & \text{if the output satisfies the required format} \\ 0, & \text{otherwise} \end{cases}. \tag{4}$$

The correctness reward $r_c$ encourages the model to achieve high answer accuracy and it is defined as:

$$r_c(o_k, y_i) = \begin{cases} +1, & \text{if the final answer is correct} \\ 0, & \text{otherwise} \end{cases}. \tag{5}$$

Length reward discourages overthinking and token efficiency (Team et al., 2025a), which as:

$$r_l(o_k, y_i) = \begin{cases} 0.5 - \frac{\text{len}-\text{min\_len}}{\text{max\_len}-\text{min\_len}}, & \text{if the final answer is correct} \\ \min\left(0, 0.5 - \frac{\text{len}-\text{min\_len}}{\text{max\_len}-\text{min\_len}}\right), & \text{otherwise} \end{cases}, \tag{6}$$

where len is the length of $o_k$, min\_len / max\_len are the minimum / maximum length of $\{o_1, o_2, ..., o_G\}$.

**Learning Objectives.** The goal of learning is to maximize the expected long-term return $\mathcal{J}(\theta)$:

$$\mathcal{J}_{GRPO}(\theta) = \mathbb{E}[x \sim P(Q), \{o_i\}_{i=1}^G \sim \pi_{\theta_{old}}(O|x)]$$

$$\frac{1}{G}\sum_{i=1}^{G}\left\{\min\left[\frac{\pi_\theta(o_i|x)}{\pi_{\theta_{old}}(o_i|x)}\hat{A}_i, \text{clip}\left(\frac{\pi_\theta(o_i|x)}{\pi_{\theta_{old}}(o_i|x)}, 1-\varepsilon, 1+\varepsilon\right)\hat{A}_i\right] - \beta D_{KL}[\pi_\theta||\pi_{ref}]\right\}. \tag{7}$$

where $\varepsilon$ and $\beta$ are hyper-parameters, $\pi_\theta$ and $\pi_{\theta_{old}}$ are the current and old policy models.

## 4 EXPERIMENT

**Datasets and Metrics.** We train the target reasoning model on the MATH training dataset (Hendrycks et al., 2021) and evaluate on multi-task, multi-difficulty benchmarks including mathematical questions: GSM8K (Cobbe et al., 2021), MATH500 (Lightman et al., 2024), AIME24 (Art of Problem Solving, 2024), AIME25 (Art of Problem Solving, 2025), multidisciplinary reasoning: GPQA Diamond (Rein et al., 2023), and challenging code generation: LiveCodeBench (Jain et al., 2024). For small-scale datasets, we report AIME24@32 and AIME25@32.

We evaluate the performance on: **1) Accuracy (acc., %)**: The percentage of questions for which the final answer is correct. **2) Average Thinking Length (len.)**: The average number of tokens in a complete thinking sequence, encompassing the reasoning content and the external feedback.

**Baselines.** We compare Think-with-Me with DeepSeek-R1-Distill-Qwen2.5-7B/14B/32B (DeepSeek-AI et al., 2025), QwQ-32B (Team, 2025), Qwen2.5-72B-Instruct (Yang et al., 2024a) and Qwen2.5-Math-72B (Yang et al., 2024b) and with ablations: (i) the base model trained only with a length penalty (Team et al., 2025b), and (ii) prompting the base model to think concisely (Nayab et al., 2024). Referring to the related work (Appendix A), we include representative efficient reasoning methods. For training optimization, we include L1-Max (Aggarwal and Welleck, 2025) and s1-32B (Muennighoff et al., 2025). For test-time optimization, we include DEER (dynamic early exit) (Yang et al., 2025b), SEAL (adaptive reasoning) (Chen et al., 2025a), Speculative Thinking (Speculative Decoding) (Yang et al., 2025c), and Specreason (multi-model collaboration) (Pan et al., 2025).

**Implementation Details.** The base reasoning-enhanced model is DeepSeek-R1-Distill-Qwen2.5-7B. We set the reasoning to be forcibly terminated after 10 interactions with feedback. Feedback comes from both an LLM proxy and human volunteers under the same predefined criteria. Most experiments use an 8K context window and we report results for reasoning-enhanced models at 32K for fair comparison (DeepSeek-AI et al., 2025). The details of the experiment setting are in Appendix D.

Table 1: Accuracy (acc., %) and average thinking length (len.) of Think-with-Me where the external feedback provided by **LLM proxy** and baselines across benchmarks. Best and second-best results are in **bold** and underlined, respectively. WS denotes Window Size.

| WS | Method | GSM8K | | MATH500 | | AIME24 | | AIME25 | | GPQA | | Livecode | |
|---|---|---|---|---|---|---|---|---|---|---|---|---|---|
| | | acc. | len. | acc. | len. | acc. | len. | acc. | len. | acc. | len. | acc. | len. |
| | Qwen2.5-72B-Instruct | 94.92 | - | 83.76 | - | 16.67 | - | 15.00 | - | 10.61 | - | 10.29 | - |
| | Qwen2.5-Math-72B | 95.75 | - | 84.60 | - | 23.57 | - | 21.67 | - | 7.58 | - | 0.00 | - |
| | DeepSeek-R1-7B | 87.65 | 1110.10 | 84.05 | 1864.33 | 40.00 | 3374.50 | 35.71 | 6371.46 | 22.25 | 5020.86 | 22.29 | 6305.82 |
| | DeepSeek-R1-14B | 94.05 | 1037.15 | 84.13 | 2003.44 | 53.33 | 3257.70 | 38.89 | 6480.27 | 24.66 | 4411.10 | 32.00 | 6350.63 |
| | DeepSeek-R1-32B | 95.32 | 1199.42 | 85.98 | 2131.25 | 70.00 | 3134.43 | 44.44 | 8751.76 | 36.30 | 5028.59 | 34.86 | 6387.92 |
| | QwQ-32B | 95.52 | 1066.38 | 84.26 | 2540.38 | 66.66 | 4052.80 | 39.68 | 6428.53 | 37.42 | 5585.19 | 32.57 | 6305.63 |
| | length penalty only | 79.15 | 834.59 | 87.80 | 1355.85 | 45.71 | 2322.03 | 41.11 | 3514.69 | 26.35 | 4821.33 | **37.71** | 4909.14 |
| 8k | prompt to think concise | 86.50 | 699.62 | 88.20 | 1259.27 | 52.74 | 2573.43 | 35.56 | 3845.30 | 31.97 | 4120.47 | 34.86 | 5201.16 |
| | L1-Max | 93.94 | 2913.56 | 68.60 | 1615.70 | 30.52 | 1662.96 | 22.19 | 3817.53 | 28.57 | 1731.86 | 24.57 | 3649.40 |
| | s1-32B | **96.00** | 1023.26 | 90.00 | 2922.14 | 55.00 | 6047.17 | 52.92 | 5868.90 | **68.00** | 3888.79 | 30.00 | 4019.50 |
| | DEER | 91.00 | 392.25 | 87.00 | 1280.75 | 54.06 | 4127.76 | 49.79 | 4097.15 | 63.33 | 3136.17 | 30.80 | 3487.17 |
| | SEAL | 89.30 | **297.49** | 84.00 | 2096.87 | 51.35 | 6203.53 | 33.33 | 5919.77 | 57.33 | 3338.80 | 31.30 | 5987.83 |
| | Spectuve Thinking | 91.91 | 605.09 | 77.00 | **934.40** | 52.08 | **1124.03** | 26.67 | 3770.10 | 56.63 | 6805.80 | 32.57 | 9310.15 |
| | specureason | **96.00** | 1075.98 | 92.00 | 2380.78 | 50.00 | 6278.55 | 36.00 | 6431.01 | 50.51 | 5450.72 | 30.00 | 6327.84 |
| | **Ours(LLM proxy)** | 95.80 | 322.40 | **90.60** | 1081.87 | **73.85** | 1182.50 | **71.43** | **1964.22** | 65.22 | **1136.38** | 34.86 | **1248.64** |
| | DeepSeek-R1-7B | 92.72 | 1323.04 | 87.60 | 3463.52 | 50.00 | 5608.00 | 43.33 | 6320.00 | 27.45 | 17040.71 | 32.00 | 13332.02 |
| 32k | DeepSeek-R1-14B | 93.33 | 1227.62 | 90.70 | 3168.17 | 70.00 | 5948.55 | 58.33 | 6195.02 | 47.64 | 9678.51 | 46.28 | 9414.02 |
| | DeepSeek-R1-32B | 94.00 | 1215.42 | 91.75 | 3309.54 | 70.97 | 5758.77 | 60.00 | 7065.50 | 53.80 | 11733.01 | 53.14 | 10843.10 |
| | QwQ-32B | 95.29 | 1307.56 | 92.51 | 3265.72 | 73.33 | 8794.43 | 70.00 | 8794.43 | 53.15 | 11732.01 | 54.86 | 21648.67 |

Table 2: Accuracy (acc., %) and average thinking length (len.) of Think-with-Me where the external feedback provided by **human-in-the-loop** and baselines across datasets. Considering the workload of human involvement, 90 samples are randomly selected from every dataset. For small-scale datasets, we report AIME24@3 and AIME25@3. Other details are the same as Table 1.

| Method | GSM8K | | MATH500 | | AIME24 | | AIME25 | | GPQA | | Livecode | |
|---|---|---|---|---|---|---|---|---|---|---|---|---|
| | acc. | len. | acc. | len. | acc. | len. | acc. | len. | acc. | len. | acc. | len. |
| DeepSeek-R1-7B | 88.89 | 1290.73 | 86.67 | 1722.73 | 44.44 | 3903.83 | 38.89 | 6298.03 | 25.56 | 4880.62 | 26.67 | 6274.63 |
| length penalty only | 86.67 | 479.63 | 88.89 | 1740.83 | 45.56 | 2367.73 | 45.56 | 5259.15 | 28.89 | 5217.35 | 33.33 | 5018.09 |
| prompt to think concise | 90.00 | 442.43 | 87.78 | 1587.67 | 47.78 | 3143.75 | 42.22 | 3214.26 | 32.22 | 4461.60 | 26.67 | 4883.45 |
| L1-Max | 92.22 | 2065.12 | 83.33 | 1864.20 | 37.78 | 1662.96 | 36.67 | 2762.30 | 26.67 | 1853.33 | 16.67 | 3899.67 |
| s1-32B | 95.55 | 1029.41 | 90.00 | 2928.67 | 54.44 | 6651.48 | 52.22 | 5930.53 | 65.56 | 3627.19 | 30.00 | 4019.50 |
| DEER | 91.00 | 392.25 | 87.00 | 1280.75 | 54.06 | 4127.76 | 49.79 | 4097.15 | 57.29 | 3093.31 | 30.80 | 3480.86 |
| SEAL | 88.89 | **300.00** | 83.33 | 2090.69 | 51.11 | 6196.36 | 34.44 | 5889.54 | 56.67 | 3386.66 | 32.20 | 5692.74 |
| Spectuve Thinking | 91.11 | 676.70 | 85.56 | 1145.61 | 51.11 | **1098.96** | 40.00 | 3855.43 | 58.89 | 5691.08 | 31.11 | 9310.15 |
| specureason | **95.56** | 782.94 | 92.22 | 2372.07 | 52.22 | 6205.45 | 35.56 | 6155.71 | 48.89 | 6067.04 | 35.56 | 6337.84 |
| Ours (LLM proxy) | 94.44 | 399.60 | **92.23** | 1085.31 | **65.56** | 1263.07 | 55.56 | 2083.74 | **66.67** | 1163.56 | **36.67** | **1120.05** |
| Ours (human) | **95.56** | 388.80 | 90.00 | **1009.40** | 61.62 | 1597.00 | **61.11** | **1991.14** | 64.44 | **1067.09** | 34.44 | 1406.80 |

## 4.1 COMPARISON EXPERIMENTS

**Superior Reasoning Efficiency under Limited Context Windows.** From Table 1, Think-with-Me with LLM proxy averages 1182.50 tokens for 73.85% on AIME24 under an 8K window, outperforming QwQ-32B at 32K. In contrast, the length penalty only and the prompt to the concise setting fail to balance length and accuracy. This indicates that with external feedback, the target model reasonably stops or extends the reasoning process, reducing redundant reasoning without sacrificing accuracy and therefore curbing the overthinking and overshoot phenomenon.

**Dynamic Control of Reasoning Length via External Feedback Across Diverse Tasks.** From Table 1, we can clearly observe that Think-with-Me with LLM proxy demonstrates adaptive reasoning length control while maintaining competitive accuracy across **various** mathematical (GSM8K, MATH500, AIME), multidisciplinary (GPQA), and code generation (LiveCodeBench) tasks under an 8K context window constraint. This indicates that external feedback intervention effectively replaces fixed budgeting heuristics with flexible adaptive reasoning control for reasonably extending or terminating the reasoning process in real time, achieving an optimal efficiency-accuracy trade-off.

Table 3: The behavioral analysis metrics across datasets. Interact. (Average intervention counts), Stop. (Proportion of self-termination of the reasoning process not due to the maximum number of interactions), Feedback. (Average tokens for feedback content), thinking. (Average tokens for the whole thinking content, including feedback), Ratio. (Ratio of feedback length to reasoning).FbkT.(Average time for providing feedback in a complete reasoning process, in seconds.), ThkT.(Average time for the reasoning stage of the target reasoning model, in seconds.

| Feedback Source | Metric | GSM8K | MATH500 | AIME24 | AIME25 | GPQA | Livecode |
|---|---|---|---|---|---|---|---|
| LLM proxy | Interact. | 0.79 | 2.63 | 3.80 | 5.13 | 3.99 | 4.00 |
| | Stop. | 95.87 | 93.96 | 76.67 | 73.11 | 79.90 | 74.85 |
| | Feedback. | 72.49 | 146.43 | 389.20 | 352.76 | 242.27 | 312.00 |
| | Thinking. | 322.40 | 1081.87 | 1998.20 | 1964.22 | 1136.38 | 1248.64 |
| | Ratio. | 22.36 | 13.51 | 19.48 | 17.96 | 21.30 | 25.00 |
| | FbkT. | 2.81 | 11.97 | 21.84 | 22.06 | 16.36 | 21.65 |
| | ThkT. | 8.97 | 24.41 | 37.43 | 37.42 | 20.03 | 18.77 |
| Human | Interact. | 0.90 | 3.10 | 4.15 | 6.29 | 3.15 | 5.95 |
| | Stop. | 100.00 | 90.00 | 89.23 | 80.21 | 84.22 | 75.00 |
| | Feedback. | 19.00 | 33.10 | 46.77 | 55.85 | 34.63 | 63.65 |
| | Thinking. | 606.90 | 1221.50 | 1860.46 | 2383.74 | 1181.79 | 1504.85 |
| | Ratio. | 3.13 | 2.64 | 2.51 | 2.34 | 2.93 | 4.23 |
| | FbkT. | 2.83 | 13.44 | 27.45 | 60.25 | 18.26 | 34.59 |
| | ThkT. | 8.68 | 17.77 | 28.03 | 54.20 | 17.17 | 25.26 |

**Support multi-source external feedback including human-in-the-loop and LLM proxy.** From Table 1 and Table 2, feedback from both sources reduces reasoning length while maintaining competitive accuracy. On MATH500, our method with LLM-proxy attains 90.60% with 1081.87 tokens, and with human-in-the-loop achieves 90.00% with 1221.50 tokens outperforms the base model under a 32K window size. Considering the limitations of individual knowledge and professional fields, Think-with-Me with human-in-the-loop is not SOTA in accuracy across datasets, but strong in balancing accuracy and reasoning length. Details of the feedback source are in Appendix E.

### 4.2 THINKING BEHAVIOR UNDER THINK-WITH-ME PARADIGM

**Automatic self-termination by the target model.** From Table 3, under interactive reasoning mode, Think-with-Me with human feedback has self-termination exceeding 80% on most datasets and reaches 100% on GSM8K; with an LLM proxy, it remains high (73.11%–95.87%). Self-termination means the model emits `</think>` by itself. This indicates that the target model adapts to the interactive reasoning mode and Think-with-Me is operationally usable across tasks.

**Token and time cost of external feedback are acceptable from both LLM proxy and human.** From Table 3, feedback from the LLM-proxy averages 13%–25% of total length and feedback from human beings is even shorter at 2%–4%. In wall-clock terms, both LLM proxy and human complete all feedback for a question within seconds under pre-defined criteria. Furthermore, intervention intensity varies with task difficulty for LLM proxy and human: average interactions are 0.79/0.90 (LLM proxy/human) on GSM8K vs. 5.13/6.29 on AIME25, which is acceptable for interaction (Javernik et al., 2023). More details about time and computation cost are in Appendix F.

**The feedback from the LLM proxy and human feedback fit for different scenarios.** From Table 3, human feedback uses fewer tokens and occupies a smaller share of the reasoning trace, providing concise, targeted guidance which is crucial for the reliability in high-risk or creative cases. The LLM proxy produces longer feedback but makes the reasoning process fully automatic, enabling scalable tasks in batch or high-throughput settings. In practice, we recommend starting with the LLM proxy in Think-with-Me and taking over by humans for difficult or safety-critical tasks. The graphical interactive interface in Appendix I supports this hybrid workflow and seamless takeover.

### 4.3 ABLATION STUDY

**All components are necessary for the best accuracy–length trade-off.** From Table 4, removing major components degrades both accuracy and token efficiency. Fixed-length triggers (*w/o specific intervention point*) feedback on incomplete reasoning and weaken guidance; dropping feedback

Table 4: Ablation results of **Think-with-Me** across different datasets. w/o means without..

| Method | GSM8K | | MATH500 | | AIME24 | | GPQA | | LiveCode | |
|---|---|---|---|---|---|---|---|---|---|---|
| | acc. | len. | acc. | len. | acc. | len. | acc. | len. | acc. | len. |
| w/o specific intervene point | 75.45 | 598.98 | 80.00 | 2280.10 | 38.33 | 2303.52 | 42.34 | 2235.57 | 25.14 | 1504.77 |
| w/o special tokens | 91.13 | 618.22 | 86.40 | 2114.98 | 57.91 | 3592.33 | 46.43 | 2679.55 | 32.57 | 2193.00 |
| w/o trained model | 88.09 | 322.95 | 75.23 | 1190.85 | 33.75 | 2151.13 | 37.24 | 2011.89 | 28.00 | 1509.00 |
| w/o fine-grained feedback | 90.36 | 358.56 | 81.57 | 1785.19 | 44.06 | 2177.53 | 43.88 | 1833.67 | 31.43 | 1504.80 |
| w/o GRPO | 92.20 | 324.83 | 86.00 | 1404.15 | 63.33 | 3617.78 | 64.20 | 1523.46 | 28.65 | 1235.57 |
| Ours (pink background) | 95.80 | 322.40 | 90.60 | 1081.87 | 73.85 | 1998.20 | 65.22 | 1136.38 | 34.86 | 1248.64 |

Table 5: Parameter ablation of composite reward function across datasets (acc: accuracy; avg_len: average length; numbers in $(\cdot, \cdot, \cdot)$ denote sub-reward weights).

| Method | GSM8K | | MATH500 | | AIME24 | | GPQA | | LiveCode | |
|---|---|---|---|---|---|---|---|---|---|---|
| | acc. | len. | acc. | len. | acc. | len. | acc. | len. | acc. | len. |
| base model | 87.65 | 1510.10 | 84.05 | 1864.32 | 40.00 | 3374.50 | 22.25 | 5020.86 | 22.29 | 6305.82 |
| $(0, 0, 0)$ | 88.09 | 306.25 | 75.23 | 1067.59 | 36.67 | 2276.47 | 37.24 | 2011.89 | 28.00 | 1509.00 |
| $(1, 0, 0)$ | 88.56 | 342.64 | 85.00 | 1149.79 | 40.00 | 2166.57 | 40.31 | 1934.65 | 22.77 | 1439.30 |
| $(1, 1, 0)$ | 92.49 | 351.88 | 85.20 | 1168.44 | 45.56 | 2072.54 | 42.35 | 1980.23 | 25.35 | 1476.73 |
| $(1, 1, 1)$ | 95.80 | 322.40 | 90.60 | 1081.87 | 73.85 | 1998.20 | 65.22 | 1136.38 | 34.57 | 1248.64 |
| $(1, 0.5, 1)$ | 90.88 | 339.36 | 87.60 | 1240.04 | 42.22 | 2156.36 | 56.63 | 1305.60 | 30.71 | 1445.42 |
| $(1, 1, 0.5)$ | 89.64 | 330.69 | 87.20 | 1075.75 | 45.56 | 2370.23 | 52.55 | 1158.21 | 26.00 | 1444.75 |
| $(0.5, 1, 1)$ | 86.73 | 324.89 | 85.20 | 1145.67 | 44.44 | 2386.35 | 50.51 | 1384.81 | 28.58 | 1341.57 |

markers (*w/o special tokens*) blurs feedback and reasoning content; using a non-adapted target model (*w/o trained model*) reduces the responsiveness to feedback; and binary feedback (*w/o fine-grained feedback*) provides insufficient, non-actionable guidance. SFT training strategy (*w/o GRPO*) leads to suboptimal adjustment to the interactive reasoning paradigm. Therefore, explicit phased triggering, feedback tags, GRPO adaptation, and fine-grained feedback are all required for efficient reasoning. More ablation results about intervene strategies are in Appendix G.3.

**Composite reward needs balanced weighting.** From Table 5, under interactive reasoning mode, the equal-weight scalarization $(1, 1, 1)$ gives the strongest overall trade-off, whereas removing or down-weighting any term (e.g.,$(1, 0.5, 1)$ ) lengthens the reasoning phase on LiveCodeBench and $(0.5, 1, 1)$ slashes accuracy on AIME24. As the key points of the interactive reasoning paradigm, all three rewards are complementary and thus jointly necessary for stable, efficient, and accurate training under an interactive reasoning mode with external feedback.

## 4.4 MORE DISCUSSIONS

**Impact of LLM Proxy Capability on Performance of Think-with-Me.** From Table 6, Think-with-Me with proxies constructed by a larger scale performs better in improving accuracy while keeping the thinking efficiency compared with the base model. With higher consistency of human preference, an LLM proxy constructed by a larger-scale model provides higher-quality feedback, which is critical for guiding the target model through challenging reasoning tasks. More discussion about the advantages of Think-with-Me in extending window size is in Appendix H.3.

Table 6: The impact of different feedback qualities of LLM proxy on task performance. We use Fleiss' kappa (Fleiss and Joseph, 1971) to measure the consistency between LLM proxy and human preference as the quality of feedback from LLM proxies. More Details and results are in Appendix H.1.

| LLM proxy | Fleiss' K | AIME24 | | Livecode | |
|---|---|---|---|---|---|
| | | acc. | len. | acc. | len. |
| Qwen2.5-32b-Instruct | 0.45 | 63.33 | 1241.48 | 29.14 | 1344.27 |
| Qwen2.5-72b-Instruct | 0.51 | 73.85 | 1182.50 | 34.86 | 1248.64 |
| kimi-k2-instruct | 0.68 | 76.56 | 1193.67 | 40.57 | 1277.33 |

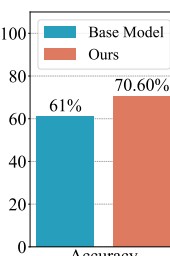 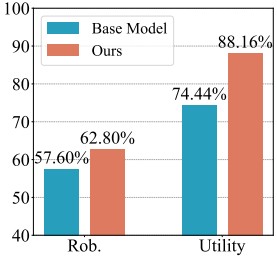 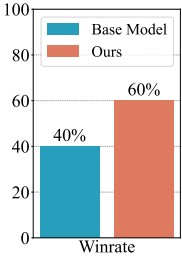

(a) Accuracy on IFEval.  (b) Robustness and Utility on SEP.  (c) Winrate on LitBench.

Figure 3: The effect of Think-with-Me by multi-source feedback in security and creative scenarios.

**Potential applications in diverse tasks.** We present case studies that mitigate overthinking and overshoot in Appendix H.4. We explore the effect of the proposed interactive reasoning mode in other critical (Wallace et al., 2024; Qin et al., 2024) and creative tasks (Wang et al., 2025) in Figure 3: with LLM proxy, we evaluate Think-with-Me on IFEval (Qin et al., 2024) for instruction following and on SEP (Wallace et al., 2024) for instruction hierarchy; with human-in-the-loop, we evaluate Think-with-Me on LitBench (Fein et al., 2025). These results indicate that Think-with-Me transfers across tasks and feedback sources. Experiment Details are in Appendix H.2.

## 5 CONCLUSION

We propose Think-with-Me, an interactive intervene framework that mitigates overthinking and overshooting for LRMs rooted from LRM's intrinsic reasoning behavior. Think-with-Me uses transitional conjunctions as intervention points for multi-source external feedback from humans or LLM proxies to adaptively adjust the reasoning process, providing explicit control points to pause, redirect, or terminate reasoning as needed. With GRPO training on an additional LoRA module to adapt to the proposed interactive reasoning mode, Think-with-Me reduces reasoning length while maintaining or even improving accuracy under a limited window size across multiple tasks with different difficulty levels. The simplicity of intervene mechanism supports the extension to more creative and safety-critical tasks, broadening LRMs' application boundaries and making reasoning-enhanced models more efficient, controllable, and collaborative.

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

# Supplementary Materials for
# "Efficient Deep Thinking with Human-in-the-Loop"

## CONTENTS

# A    RELATED WORK

**Efficient Reasoning Method.** Existing research in efficient reasoning can be divided into two paradigms:  single-model optimization and multi-model collaboration: **Single-Model Optimization** optimizes efficiency through training strategies, like L1 (Aggarwal and Welleck, 2025) and s1 (Muennighoff et al., 2025) or via internal numerical signals, like confidence Liu et al. (2025a) and latent states Chen et al. (2025a). Represent methods focus on test time include dynamic early exiting, like DEER Yang et al. (2025b), which uses pivotal tokens to decide when to output or rollback with just single strategy to control the reasoning length, and token-level control, like SEAL (Chen et al., 2025a). These methods primarily rely on suppressing reflection or enforcing early stopping, but Think-with-Me operates at the semantic level. It allows the model to adaptively stop or extend its reasoning based on the input question and its current state. Crucially, it provides directional, semantically-grounded feedback, which enhances reasoning focus, human readability, and supports multi-source feedback. **Multi-Model Collaboration** leverages the strengths of different models for complex tasks. Frameworks such as long-short model collaboration, like SpecReason (Pan et al., 2025), LLM routing (Ong et al., 2025), model integration (Xu et al., 2025b), and speculative decoding, like Specutive Thinking (Yang et al., 2025c) assign reasoning sub-tasks of varying difficulty to models with corresponding capabilities, improving overall accuracy and efficiency. These multi-model collaboration incurs communication overhead from loading/unloading processes across different models, while our method maintains a single, primary reasoning model.

**Note.** Although our method exhibits similarities to existing methods in forms of solutions, for instance, in the LLM proxy scenario, its solution strategy is similar to that of multi-model collaboration, we emphasize that our contribution is an interactive intervention mechanism rooted in the intrinsic reasoning behaviors of Large Reasoning Models (LRMs). Specifically, the framework leverages external feedback provided from multiple resources in accordance with content-agnostic, goal-oriented criteria to adaptively adjust the depth and direction of reasoning, mitigating the overthinking and overshoot problems in LRMs.

Think-with-Me enables in-depth human-LLM interaction and collaboration throughout the reasoning process. Its mechanism is straightforward: it is rooted in the model's intrinsic behaviors and low-barrier feedback provision criteria, which support the method's multi-task scalability. Additionally, our method explicitly defines an intervention-based control strategy for LRM reasoning.This strategy enhances the controllability of LRM reasoning, thereby broadening the application boundaries and practical value of LRMs.

**Thinking Evaluation.** While many works evaluate the generated result by just accuracy (Luo et al., 2025; Chern et al., 2023; Lu et al., 2023) or other metrics for the final answer, more works reveal that neglecting the quality of the intermediate steps can mask underlying problems, such as logical errors or redundancy in the thinking process (Hammoud et al., 2025; Xia et al., 2025). Some works (Sawada et al., 2023) measure the thinking quality by comparing the similarity between generated and reference solution steps. Although datasets like MATH-500 (Lightman et al., 2024) and GSM8K (Cobbe et al., 2021) provide the ground truth and standard solution, diverse thinking paths can lead to the same correct answer. Recent works attempt to design a set of evaluation criteria and construct an LLM-based proxy to efficiently and extensively provide evaluation results based on these criteria (Xia et al., 2025; Lee and Hockenmaier, 2025). To verify the effectiveness of Think-with-Me for efficient reasoning, diverse and large-scale experiment on different datasets is necessary, we apply LLM-based proxy to efficiently provide external feedback in the most of experiment and provide experimental results with human in the loop as much as possible.

# B  THE DETAILS OF PRE-EXPERIMENTS

## B.1  THE DETAILS OF DIRECTLY APPLYING THE PREVIOUS EXTERNAL FEEDBACK STRATEGY ON LRMS.

Table 7: Applying Self-Refine on LRMs with feedback from itself or Qwen2.5-72B-Instruct.

| Method | math500 | | gpqa | | LiveCode | |
|---|---|---|---|---|---|---|
| | accuracy | length | accuracy | length | accuracy | length |
| base model | 84.05 | 1864.33 | 22.25 | 5020.86 | 22.29 | 6305.82 |
| Self-Refine[r1] | 89.90 | 2335.98 | 28.49 | 8476.44 | 19.19 | 8523.69 |
| Self-Refine (Qwen2.5-72B-Instruct) | 87.00 | 2226.67 | 25.00 | 6335.25 | 29.63 | 7343.26 |
| Think-with-Me(Qwen2.5-72B-Instruct) | 90.6 | 1081.87 | 65.22 | 1136.38 | 34.86 | 1248.64 |

Several existing works (Madaan et al., 2023) have already explored, from various perspectives, the use of external feedback frameworks—such as human feedback, model feedback, or verifiers. However, they misalign with LRMs' reasoning behavior, and fail to optimize the accuracy–length trade-off for LRMs. Most methods apply feedback post-generation (Shinn et al., 2023) or at fixed answer stages (Luo et al., 2024), mismatching LRMs on timing, content, and form.

To further validate the empirical motivation of Think-with-Me and its superiority over existing external feedback methods, we compare with Self-Refine (Madaan et al., 2023), a representative feedback-based method. We apply Self-Refine to the base model in this paper, DeepSeek-R1-Distill-Qwen-7B, with two feedback sources: (1) self-feedback from the base model itself; (2) feedback from Qwen2.5-72B-Instruct (consistent with our LLM proxy setup). The results are shown in Table 7. Self-Refine (both self-feedback and Qwen2.5-72B-Instruct feedback) fails to reduce reasoning length while maintaining task accuracy, clearly demonstrating the limitations of existing feedback methods for LRM's efficient reasoning. These results confirm that existing feedback-based methods are misaligned with LRMs' reasoning behavior and further motivate the proposal of Think-with-Me, which is an interactive intervention mechanism—grounded in LRMs' intrinsic reasoning phase boundaries—that achieves the critical goal of shorter reasoning with higher accuracy for LRMs.

## B.2  THE DETAILS OF OBSERVATION EXPERIMENT

We use Qwen2.5-72B-Instruct to analyze the thinking behavior of a large reasoning model on different difficulty problems according to the following instruction, analyzing the key conclusions in the thinking process and how they are connected.

### B.2.1  THE DETAILS OF COUNTING THE APPEARANCE OF WORDS IN APPEARANCE.

In our analysis, we count the frequency of individual tokens as produced by the tokenizer and then aggregate variants of the same word into a single lemma. For example, all of the following variants are grouped under the word "so":

$$\{" \text{ so}", "\text{So},", "\text{So}"\},$$

and similarly for "wait", "but", etc. Punctuation, capitalization, and leading spaces are normalized before aggregation.

In Figure 1, the horizontal axis represents each cluster of similar reasoning examples (i.e., all examples that share the same inference pattern). Within each cluster, we count how many times each word appears across all reasoning steps. We ignore purely punctuation tokens (e.g. ",", ".") and some stop-token variants that carry no semantic weight. After normalization, we keep only the top 100 words by total frequency.

**Raw token counts (before aggregation)**:

The results are in Table 8. Here '\textvisiblespace' denotes a leading whitespace in the token. After summing these variants, we obtain the aggregated counts shown in Table 9.

Table 8: Raw Token Counts Before Aggregation

| Token Variant | Count |
|---|---|
| So, | 2417 |
| So | 1914 |
| \textvisiblespace so | 2076 |
| Wait | 1565 |
| \textvisiblespace Wait | 226 |
| \textvisiblespace wait | 223 |
| \textvisiblespace but | 1049 |
| But | 970 |
| \textvisiblespace But | 339 |
| Therefore | 729 |
| \textvisiblespace Therefore | 266 |
| Alternatively | 477 |
| \textvisiblespace maybe | 664 |
| Maybe | 213 |

Table 9: Top 7 words by total frequency in reasoning

| Word | Total count | Avg. per 500 examples |
|---|---|---|
| so | 6 407 | 12.81 |
| wait | 2 014 | 4.03 |
| but | 2 358 | 4.72 |
| therefore | 955 | 1.91 |
| alternatively | 447 | 0.89 |
| maybe | 877 | 1.75 |
| (others …) | … | … |

### B.2.2 THE COMMON CONTENT PATTERN FOR CAUSAL AND TRANSITIONAL CONNECTORS

This section details the identification of prevalent lexical patterns that frequently co-occur with specific causal and transitional connectors. We systematically analyzed the textual context surrounding these connectors (designated as 'Base Word' in Table 10) to determine the most common associated words or phrases (listed under 'Patterns'). The regular expressions employed for this extraction are also presented.

Table 10: The common content pattern for causal and transitional connectors.co-words means Co-occurrence words with specific base words.

| Base Word | Co-occurrence words | Combined Method |
|---|---|---|
| Wait / wait | make sure, might, maybe, if, new, another, check, verify, whether, possible, hold on, not | "Wait\s.{0,30}\s.+{co-words}"
"wait\s.{0,30}\s.+{co-words}" |
| But / but | if, new, another, check, verify, whether, maybe, possible, wait | "But\s.{0,30}\s.+{co-words}"
"but\s.{0,30}\s.+{co-words}" |
| Alternative / alternative | if, new, another, check, verify, whether, maybe, possible, wait | "Alternative\s.{0,30}\s.+{co-words}"
"alternative\s.{0,30}\s.+{co-words}" |
| So / so | can, is, would, be | "So\s.{0,30}\s.+{co-words}"
"so\s.{0,30}\s.+{co-words}" |

### B.2.3 THE DETAILS OF ANALYZING THE THINKING PROCESS

We report the prompt to analyze whether the current sub-section contains the correct final answer to the given question.

**Experimenal details of the observation 2**. To examine the impact of transitional conjunctions on reasoning dynamics, we prompt DeepSeek-R1-Distill-Qwen-32B to generate 16 reasoning traces per question from our test set, using a temperature of 1.0 and top-p of 0.95. We then filter the samples to ensure that, for each question, both correct and incorrect final answers are present across the generated traces. Using the transitional conjunctions identified in Observation 1 as segmentation points, we divide each reasoning trace into sub-reasoning units. For each segment preceding a conjunction, we instruct Qwen2.5-72B-Instruct to judge whether there exists a correct final answer to the question before the specific transition conjunction, based on the ground truth and a reference solution. The prompt template is provided below. We compute the final answer accuracy conditioned on whether the intermediate sub-reasoning already contains a correct solution. As shown in Figure 1c, the results demonstrate that transitional conjunctions, when used judiciously, can enhance accuracy by enabling extended reasoning. However, excessive use of such conjunctions often leads to degraded performance.

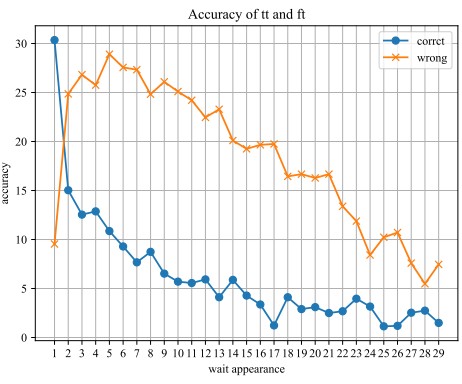 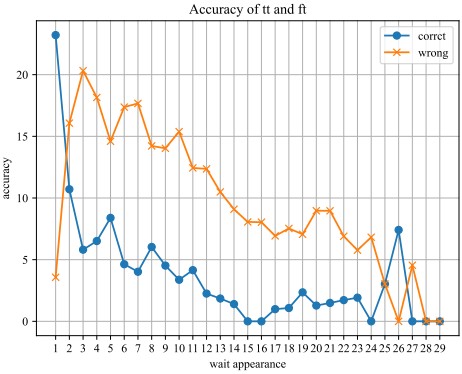

(a) The impact of extending thinking traces on final accuracy on the AIME dataset.

(b) The impact of extending thinking traces on final accuracy on the GPQA dataset.

---

**Evaluation Prompt for Subthinking**

**Task**

Your task is to analyze whether the current thinking is on the correct track based on the given corresponding question, standard solution, and correct answer.

**Standard**

There are two possible evaluation results and their corresponding criteria:

1. **Yes, the current thinking contains the right final answer:** The current thinking includes a standard solution or is close to or consistent with the standard solution, and the current thinking can obtain the correct answer in the future.

2. **No, the current thinking does not contain the right final answer:** The current thinking does not include the standard solution or is inconsistent with the standard solution, and the current thinking cannot obtain the correct answer in the future.

**Tips**

1. Your task is only to analyze whether the current thinking contains the right final answer or not, rather than giving the final answer.

2. Please do not provide any thinking or analytical process.

**Input:**

Given question: {question}
Standard solution: {solution}
Correct answer: {answer}
Current thinking: {current_thinking}

**Output:**

Your output must choose one of the two evaluation results, indicating whether the current thinking is on the correct track.
Your choice:

---

### B.2.4 THE OBSERVATION OF USING TRANSITIONAL CONJUNCTIONS IN MORE TASKS TO ENHANCE THE VALIDITY OF RESULTS.

We conducted additional experiments on the AIME (Art of Problem Solving, 2024) and GPQA (Rein et al., 2023) datasets to verify the universality of observation 2. With the same number of instances, the results are illustrated in Figure 4a, Figure 4b. The result conveys the same trend as the result on MATH500: using transitional conjunctions appropriately improves task performance, while overuse affects performance. This consistency indicates the universality of Observation 2.

## C    THE DETAILS OF EVALUATION CRITARIA

**The details of the evaluation criteria.** To provide accurate feedback for an effective intervention in the reasoning of LRM, we first assess the quality of the current reasoning. Many studies (Lee and Hockenmaier, 2025; OpenAI, 2023; Liu et al., 2025c) have found that numerical reasoning evaluation metrics cannot effectively aligned with human preferences, especially in some unconventional evaluation aspects(Lee and Hockenmaier, 2025). Therefore, theresearchers propose the LLM-as-a-Judge framework (Li et al., 2024), which instructs the large language models to adapt to human preferences for automated evaluation. Inspired by this framework, we constructed an evaluator to judge the LRM reasoning process by prompting a strong LLM (Lee and Hockenmaier, 2025), Qwen2.5-72b-Instruct.

Many works have focused on finding the reasoning trace (Hammoud et al., 2025; Xia et al., 2025), which divides the reasoning and studies the quality of current reasoning based on the rationality of structure and semantics. Following their studies, we evaluate the rationality and completeness of the current reasoning. Specifically, we divide each aspect into detailed sub-aspects, and refer to the corresponding professional literature to make specific definitions as the evaluation criteria. The details are shown as follows:

- **Rationality assesses whether the reasoning follows valid logical steps and is grounded in accurate knowledge.** It consists of three sub-criteria:
    1) Formal logic conforms. The current reasoning satisfies the basic reasoning rules. There are no situations such as skipping steps, missing steps, wrong steps, or incorrect reasoning (Gamut, 1991).
    2) Reasonable knowledge. The knowledge involved in the current reasoning conforms to common sense (Davis, 2023).
    3) Reasonable solution. The current reasoning uses a practical solution to solve the question (Wallace and Kiesewetter, 2024).

- **Completeness evaluates whether the reasoning fully addresses the question and reaches a usable conclusion.** It includes:
    1) Semantically draw the answer to the given question. Semantically, there is an answer to the given question or a conclusion with the same efficiency as the answer.
    2) Complies with the answer format. The reasoning contains content that specifies the format of the answer, such as \boxed{answer}.

## D    THE DETAILS FOR EXPERIMENTAL SETUP

### D.1    THE DETAILS OF DATASETS

**GSM8K**[1](Cobbe et al., 2021)(Grade School Math 8K) is a widely adopted standard benchmark for evaluating models' fundamental arithmetic thinking capabilities, consisting of 8.5K high-quality and linguistically diverse grade-school math word problems. These problems typically require 2 to 8 steps of step-by-step thinking to solve and include 1,319 test samples. The datasets are publicly available.

**MATH**[2](Hendrycks et al., 2021)(Mathematics Aptitude Test of Heuristic) is a highly challenging competition-level mathematics dataset designed to evaluate models' thinking abilities in advanced mathematics. In our experiments, we used the **MATH-500**[3] subset, sampled from the original MATH test set, for evaluation. This subset contains 500 carefully selected problems to ensure representativeness in terms of difficulty and distribution. The datasets are publicly available.

**AIME24**[4](Art of Problem Solving, 2024) and **AIME25**[5](Art of Problem Solving, 2025) are composed of problems from the 2024 and 2025 American Invitational Mathematics Examination (AIME)

---

[1]https://github.com/openai/grade-school-math
[2]https://github.com/hendrycks/math
[3]https://huggingface.co/datasets/HuggingFaceH4/MATH-500
[4]https://huggingface.co/datasets/Maxwell-Jia/AIME_2024
[5]https://huggingface.co/datasets/yentinglin/aime_2025

contests. Each dataset contains 30 competition-level mathematics problems that demand sophisticated, multi-step reasoning, thus serving as a challenging benchmark for advanced mathematical problem-solving. For statistical robustness, we perform 32 evaluation trials for each problem.

**GPQA**[6](Rein et al., 2023)(Grade-Level Problems in Question Answering) , a graduate-level Google-Proof Q&A benchmark comprising 448 expert-level multiple-choice questions in biology, physics, and chemistry. The questions are designed to be "Google-proof," testing deep domain knowledge over simple information retrieval.

**Livecode**[7](Jain et al., 2024) is a "live" updating benchmark designed for contamination-free code evaluation. It continuously collects new problems from competitive programming platforms such as LeetCode, AtCoder, and CodeForces. In our experiments, we use a set of 175 problems from the v6 release of the benchmark.

## D.2 The Details of Evaluation Matrices

**Accuracy (acc.)** The Accuracy metric quantifies the proportion of correctly answered instances within each dataset. The calculation process involves three distinct steps:

1. **Response Extraction:** For each instance in the dataset, the textual answer generated by the model during the inference phase is extracted. If the model's output is structured (e.g., within a specific "answer box" or designated field), only the content from this component is considered.

2. **Ground Truth Matching:** The extracted answer is then compared against the corresponding standard or ground truth answer provided in the dataset. This matching process is typically binary, determining whether the model's response is identical to the expected correct answer.

3. **Dataset-Level Aggregation:** Finally, the Accuracy for each dataset is computed as the ratio of successfully matched instances to the total number of instances in that dataset. This is expressed as: $\text{acc.} = \frac{\text{Number of correctly matched responses}}{\text{Total number of responses in the dataset}}$

**Average Token Length per Completed Thinking (len.)** The Average Token Length metric measures the average length of the model's generated responses in terms of tokens. This provides an indication of the verbosity or conciseness of the model's output. The calculation is performed as follows:

1. **Response Tokenization:** For each inference result (i.e., the generated textual response), we segment the text into tokens. To ensure a standardized and consistent measurement across all experiments, we utilize the official tokenizer from the DeepSeek-R1-Distill-Qwen-7B model for this process. This ensures that the length measurement is consistent.

2. **Token Count Calculation:** The number of resulting tokens is counted for each individual response.

3. **Dataset-Level Averaging:** The Average Token Length(len.) is then calculated as the arithmetic mean of these tokens across all instances within that specific dataset. This can be represented as: $\text{len.} = \frac{\sum_{i=1}^{N} \text{Tokens count of response}_i}{N}$ where $N$ is the total number of responses in the dataset.

## D.3 The Implementation Details for Baselines

**The Deployment Details of Model Baselines.** We evaluated several publicly available foundation models. For all foundation models listed below, inference was conducted with **temperature set to 0.6**, **top-p set to 1.0**, and utilized the **vLLM inference framework**:

- QwQ-32B
- Qwen-72B-Instruct
- Qwen-Math-72B-instruct
- DeepSeek-R1-Distill-Qwen-7B
- DeepSeek-R1-Distill-Qwen-14B

---

[6]https://huggingface.co/datasets/Idavidrein/gpqa
[7]https://huggingface.co/datasets/livecodebench/code_generation_lite

- DeepSeek-R1-Distill-Qwen-32B

**The Deployment Details of L1** (Aggarwal and Welleck, 2025) L1 utilizes Length Controlled Policy Optimization (LCPO), a reinforcement learning technique, to train language models capable of adhering to specified generation length constraints during reasoning tasks. We specifically compare against the **L1-Max** variant, designed to control the maximum length of the generated output (e.g., Chain-of-Thought) while maintaining task accuracy. For our reproduction experiments using L1-Max: the maximum target token length ('ngold') was set to **8192**, including the input prompt; inference employed deterministic decoding with **temperature set to 0.6** and **top-p set to 1**; all computations were performed on **NVIDIA A800*4**.

**Implementation Details of Speculative Thinking(**Yang et al., 2025c**).** The Speculative Thinking framework enhances a small model by delegating difficult reasoning spans to a larger model via structural cues and reflective phrases—using a dual-model setup: **DeepSeek-R1-Distill-Qwen-7B** as the speculative model ($M_S$, 1×A800) and **DeepSeek-R1-Distill-Qwen-14B** as the target ($M_T$, 1×A800). Both run on vLLM (Kwon et al., 2023) with tensor parallelism 6, and we sample 5 completions per instance. All intervention rules follow (Yang et al., 2025c), including trigger lexicons (affirmative/reflective/verification), takeover token budgets ($n_1, n_2, n_3$), and the reflection cap $C_{\text{thresh}}$.

**Length penalty only.**

We fine-tune the target reasoning model **DeepSeek-R1-Distill-Qwen-7B** on the MATH train dataset using *only* the length-penalty objective in Equation 6 by **GRPO**. Training and evaluation are conducted under **8K** context window. We stop at **epoch 106** when the validation loss plateaus, and use the "length-penalty-only" model to directly get the answer across datasets.

**Prompt to think concise.**

At inference time (no additional training), we follow the concise thinking instruction (Nayab et al., 2024) to prompt the base model, **DeepSeek-R1-Distill-Qwen-7B** to minimize unnecessary intermediate steps and stop once the solution is determined, while preserving correctness with an 8k window size. The detailed prompt is as follows:

> **The Prompt for Thinking Concise**
>
> {question} Think briefly about the answer.

### D.4 IMPLEMENTATION DETAILS OF THINK-WITH-ME

#### D.4.1 THINK-WITH-ME IMPLEMENTATION DETAILS.

This section provides detailed configurations and procedures for implementing the Think-with-Me paradigm.

- **Model training.** We build on **Deepseek-R1-Distill-Qwen2.5-7B** and start from extending its test vocabulary with `<reasoning_feedback>` and `</reasoning_feedback>` to accept external guidance. To ensure the efficiency of training and adaptation to new reasoning paradigm, A **LoRA** adapter is fine-tuned with **GRPO** on MATH train QA pairs. The Implementation of GRPO is based on the open source project[8] with a modified interactive reasoning sampling method. The details of LoRA are : $\alpha = 8$, $r = 4$, dropout= 0.1. GRPO: learning rate $1 \times 10^{-5}$, LRM temperature 1.0, trajectories = 8 per question, batch size = 1, gradient accumulation = 10. We train convergence using **PyTorch 2.6.0** (open-r1 implementation), with seed 42, max generation length **8192 tokens**, on $8\times$ **A800** GPUs.

- **Method deployment.** The LLM proxy is **Qwen2.5-72B-Instruct** (training-free), prompted with the predefined criteria in Appendix C. At test time, both the feedback proxy and the target reasoning model use temperature **0.6**. We cap the number of feedback rounds at **10**. Interventions are injected via the chat template by **appending the feedback enclosed by `<reasoning_feedback>` and `</reasoning_feedback>`** tags after the current partial reasoning in the assistant's turn, so the model continues generation conditioned on the updated context.

---

[8] https://github.com/huggingface/open-r1

### D.4.2 ELASTIC DEPLOYMENT REQUIREMENTS.

Our method is designed for elastic deployment, accommodating a wide spectrum of resource constraints. The most resource-efficient configuration requires only the target model to be deployed locally. In Human-in-the-Loop deployment, the human user naturally serves as the evaluator, eliminating the need for a separate LLM proxy. For large-scale automated testing, the evaluator role can be offloaded to external API-based models (e.g. Kimi-K2-Instruct). This minimal setup significantly lowers the barrier to entry by reducing hardware footprints and operational complexity.

For scenarios demanding lower latency and greater stability, co-locating the evaluator LLM with the target model is advantageous. While this requires additional computational resources, it offers minimal response times by eliminating network overhead from API calls and a stable and reproducible environment for large-scale batch experiments.

## E THE DETAILS OF COMPARISON EXPERIMENT

### E.1 THE DETAILS OF THE SETTINGS IN COMPARISON EXPERIMENT OF THINK-WITH-ME WITH LLM PROXY

In our experiments, the LLM proxy is designed to simulate high-quality external feedback. We utilize **Qwen2.5-72B-Instruct** as our LLM proxy, given its strong performance in instruction following and logical reasoning. The core of the proxy is a carefully constructed prompt that instructs LLM to act as an expert evaluator. For each reasoning step generated by our model, the proxy is tasked with evaluating its correctness and providing actionable feedback.

**The prompt template for LLM to act as an external evaluator.** The system prompt, presented within the box 3 environment, is structured to guide the Qwen2.5-72B-Instruct model in its designated role as an evaluator of cognitive processes. The prompt's primary directive is for the model to assess a provided 'thinking' sequence in relation to a specific 'question' and to offer constructive suggestions for improvement. The prompt template used in our experiments is detailed in in box 3.

---

**Prompt for Evaluator**

**Task**
Your task is to evaluate the provided thinking based on the given question and provide reasonable improving suggestions.

**Evaluation Criteria**
You need to evaluate whether the thinking is correct from the following aspects:

1. **Rationality:**
   - The current thinking satisfies the basic thinking rules. There are no situations such as skipping steps, missing steps, wrong steps, or incorrect thinking.
   - Reasonable knowledge. The knowledge involved in the current thinking conforms to common sense.
   - Reasonable solution. The current thinking uses a practical solution to solve the question.

2. **Completeness:**
   - Semantically draws the answer to the given question. There is an answer to the given question or a conclusion with the same efficiency as the answer.
   - Complies with the answer format. The thinking contains content that specifies the format of the answer, such as \boxed{answer}.

**Problem and Current thinking**
Given problem: {question}
Current inference result: {thinking}

**Tips**
1. Your task is only to evaluate the current thinking, not to provide the final answer to the question.
2. Please do not provide any thinking or analysis process unless it is necessary for the evaluation.

**Output Format**
Your analysis results must fall into one of four categories. Please answer strictly in the following format:

1. Rational and Complete. The current thinking is rational and contains the final answer, so it is complete.

2. Rational but Incomplete. The current thinking is rational but does not contain the final answer.

3. Irrational and Incomplete. The thinking is irrational and does not contain the final answer. The suggestion for improvement is:

4. Irrational but Complete. The thinking is irrational but contains the final answer. The suggestion for improvement is:

---

### E.2 The Details of the settings in comparison experiment of Think-with-Me with human-in-the-loop

To conduct experiment with human-in-the-loop, five graduate students Wang et al. (2023) majoring in computer science independently are invited to provide feedback to Think-with-Me on the interface shown in Figure 5. Human evaluators can give the evaluation result by pressing the buttons below and download the evaluation result by pressing the green buttons.

## F The Details of Time and Computation Cost of Methods

We report comprehensive total inference time(s) and deployment GPU requirement(MB) for compared methods, confirming our framework's practical advantages. From Table 11, while the LLM-proxy

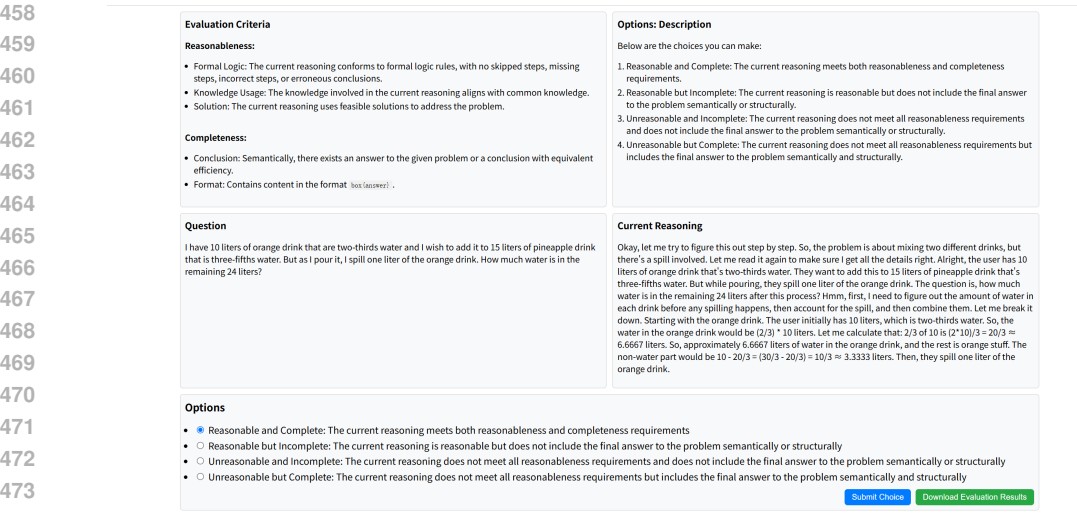

Figure 5: The interface for human evaluation

Table 11: Total time (s) and GPU memory (MB) of Think-with-Me and baselines across datasets. Minimum values are in **bold**, second minimum values are underlined.

| Method | MATH500 | | AIME24 | | GPQA | | LiveCode | |
|---|---|---|---|---|---|---|---|---|
| | total_time(s) | GPU(MB) | total_time(s) | GPU(MB) | total_time(s) | GPU(MB) | total_time(s) | GPU(MB) |
| DeepSeek-R1-7B | 35.48 | **40960** | **49.54** | **40960** | 79.48 | **40960** | 72.51 | **40960** |
| s1-32b(8k) | 127.71 | 73728 | 226.28 | 73728 | 170.47 | 73728 | 176.75 | 73728 |
| DEER | **21.75** | 65536 | 67.73 | 65536 | 47.26 | 65536 | 53.90 | 65536 |
| SEAL | 76.43 | **40960** | 247.40 | **40960** | 146.21 | **40960** | 254.14 | **40960** |
| specureason | 61.38 | 163842 | 250.09 | 163842 | 180.86 | 163842 | 208.46 | 163842 |
| ours(LLM proxy) | 36.38 | 188416 | 62.27 | 188416 | 36.39 | 188416 | **39.92** | 188416 |
| ours(human-in-the-loop) | 31.21 | **40960** | 55.48 | **40960** | **35.43** | **40960** | 59.85 | **40960** |

setting requires additional GPU memory (147456 MB) due to the local deployment of Qwen-72B-Instruct for the controlled LLM proxy, this is not necessary for realistic deployment. In practice, the feedback service can be provided by an external API and the computational overhead is solely the target 7B reasoning model, requiring only 40,960 MB of GPU memory. Think-with-Me exhibits practical utility. From Table 11, Think-with-Me exhibits a small time increment (or even shorter time) compared to the base model, demonstrating its practicality.

# G  THE DETAILS OF ABLATION EXPERIMENT

## G.1  THE DETAILS OF ABLATION EXPERIMENT ON CRITICAL METHOD EXPERIMENTS

In this section, we report the details of specific settings and key details in the ablation experiment.

### G.1.1  THE DETAILS OF ABLATION FOR SPECIFIC INTERVENTION POINTS

In this setting, we removed the model's ability to decide when to request feedback. Instead of using a special token to trigger an intervention, the reasoning process was automatically paused after generating a fixed length of 256 tokens. At this point, the generated text was evaluated by the LLM proxy, and feedback was provided for the model to continue its reasoning.

### G.1.2 THE DETAILS OF ABLATION FOR SPECIAL TOKENS

For this ablation, we eliminated the use of special tokens `<reasoning_feedback>` and `</reasoning_feedback>`. The feedback generated by the LLM proxy was directly injected into the reasoning context as plain text, immediately following the model's last generated token.

### G.1.3 THE DETAILS OF ABLATION FOR TRAINED MODEL

To assess the impact of our training, this ablation study used the base target reasoning model without GRPO training in our composite reward function. In this setting, the orignal reasoning model, DeepSeek-R1-Distilled-Qwen2.5-7B, performs reasoning with the proposed Think-with-Me framework, receiving feedback from the LLM proxy.

### G.1.4 THE DETAILS OF ABLATION FOR FINE-GRAINED FEEDBACK

In this setting, the four distinct fine-grained feedback options were substituted with a simplified binary classification system. This system categorizes the reasoning process as either "Complete Thinking" or "Incomplete Thinking." The prompt used to elicit this binary evaluation and the corresponding expected output formats are detailed in box 4 and box 5. This comparative analysis helps isolate the impact of the nuanced guidance provided by our fine-grained approach versus a more basic feedback signal.

---

**Evaluation Prompt for Subthinking**

**Task**
Your task is to evaluate the provided reasoning based on the given question and provide reasonable improvement suggestions.

**Evaluation Criteria**
1. *Rationality:*
– The reasoning follows basic logical rules (no skipped, missing, or incorrect steps).
– The knowledge used is consistent with common sense.
– The solution approach is practical and appropriate.

2. *Completeness:*
– The reasoning reaches a clear answer or conclusion equivalent to an answer.
– The reasoning specifies the answer format (e.g., contains "$\boxed{\text{answer}}$").

**Problem and Current Thinking**
Given problem: $\{question\}$
Current inference result: $\{reasoning\}$

**Tips**
1. Only evaluate the current reasoning; do not provide a new solution to the problem.
2. Avoid giving additional reasoning or analysis unless it is necessary for the evaluation.

**Output Format**
Your evaluation must fall into one of two categories (answer exactly as below):
1. `Complete Thinking`. The reasoning contains a valid final answer, so it is complete.
2. `Incomplete Thinking`. The reasoning does not contain the final answer to the question.

---

---

**Feedback Output Format**

**Complete**
`<reasoning_feedback>`
1. The current reasoning process is accurate and comprehensive, leaving no important details or logical gaps.
2. Generate the token `</think>` right away to halt any additional reasoning steps promptly.
3. Relying on the now-completed reasoning process that has been carried out, directly provide the final solution to the problem at hand.

`</reasoning_feedback>`
**Incomplete**
`<reasoning_feedback>`
1. The current reasoning is incomplete.
2. You need to continue reasoning to get the final answer. `</reasoning_feedback>`

---

## G.2 The Details of Ablation Experiment on the weight in Composite Reward Function

### G.2.1 The Setting of base model

In this setting, we used the untrained base model to solve problems via directly prompting. The setup does not utilize our interactive framework Think-with-Me, special tokens, or any feedback mechanism from an LLM proxy. This baseline represents the model's raw problem-solving capability before the application of our method.

### G.2.2 The Details of Ablation Experiment on the weight in Composite Reward Function

Setup $(0, 0, 0)$ is the baseline for all reward-trained models. It uses the untrained base model operating within the full interactive framework Think-with-Me. The model can receive feedback from the LLM proxy, but it has not undergone any training. To understand the effect of each reward, we started with an untrained baseline within our framework and incrementally added reward components, analyzing the impact at each stage. For each configuration, the model was trained using GRPO with the specified reward until the loss converged. The final trained model was then evaluated in the interactive thinking mode, guided by the LLM proxy.

### G.2.3 The Details of Weight Adjustment in Composite Reward Function

This study investigates the importance of maintaining a balance between the three reward components. Having established the full composite reward function with a default equal weighting of $(1, 1, 1)$, we explored other configurations by adjusting the relative weights of each component. This included experiments that down-weighted the correctness reward, such as $(0.5, 1, 1)$, and others that down-weighted the format reward, like $(1, 0.5, 1)$. The training and evaluation for these experiments was identical to that described in the previous section: each model variant was trained with GRPO using its specific reward weighting until convergence and then evaluated under the interactive thinking mode with LLM proxy feedback.

## G.3 More Discussion for Intervene Point

We compared other intervention points under the same 8K context window: (1) intervening after every sentence (after "."), (2) intervening every 256 tokens, and (3) intervening at the \n\n token. To accommodate the high frequency of interventions in these strategies, the maximum number of interventions was increased to 1,000. All other experimental settings remained consistent. The result is shown in Table 12. From Table 12, intervening at transitional conjunctions outperforms other strategies in balancing accuracy and length. Based on Observation 1, transitional conjunctions serve as natural stage markers in the reasoning process, where interventions at these points provide accurate feedback after a complete reasoning segment. In contrast, the compared strategies often intervene

during an ongoing reasoning stage, which leads to feedback being generated based on incomplete reasoning segments, resulting in less accurate guidance and consequently worse performance.

Table 12: The result of intervene through different points.

| Method | math500 | | AIME24 | | gpqa | | livecode | |
|---|---|---|---|---|---|---|---|---|
| | accuracy | length | accuracy | length | accuracy | length | accuracy | length |
| Base model | 84.05 | 1864.33 | 40.00 | 3374.50 | 15.23 | 2757.1 | 22.29 | 6305.82 |
| Intervene after every sentence | 82.4 | 2202.42 | 51.11 | 2963.37 | 16.67 | 3902.93 | 15.43 | 1911.11 |
| Intervene every 256 tokens | 80.00 | 2280.10 | 38.33 | 2303.52 | 20.70 | 2360.85 | 25.14 | 1504.77 |
| Intervene at "\n\n" | 86.60 | 1541.06 | 56.35 | 3943.43 | 26.67 | 1848.23 | 29.14 | 1657.08 |
| Intervene at transitional conjunctions (ours) | **90.60** | **1081.87** | **73.85** | **1182.50** | **29.90** | **1136.38** | **34.86** | **1248.64** |

## H   THE DETAILS OF MORE DISCUSSION

### H.1   THE DETAILS OF THE EVALUATION OF THE CONSISTENCY BETWEEN LLM PROXIES WITH DIFFERENT MODEL SCALES AND HUMAN PREFERENCE.

**Experimental Setup**   To evaluate the consistency between LLM proxies and human preferences, we conducted a human-in-the-loop experiment using a test set of 90 samples from the MATH500 dataset. We maintained consistent generation parameters (e.g., temperature, top-k) to ensure a fair comparison. At specific feedback points in the thinking process, we simultaneously collected feedback from both a human evaluator and an LLM proxy. The proxy's feedback then guided the subsequent reasoning steps, allowing a direct measurement of its impact. This procedure was replicated for several LLM proxies of varying scales.

**Evaluation Metric**   To quantify the consistency between the feedback provided by the LLM proxies and the human evaluators, we employ Fleiss' Kappa ($\kappa$) (Fleiss and Joseph, 1971). This metric is designed to assess the reliability of agreement among a group of raters. A $\kappa$ value in the range of $[0.4, 0.6]$ is interpreted as moderate agreement, while a value in $[0.6, 0.8]$ signifies substantial agreement.

**Results and Analysis**   The results of our evaluation are presented in Table 13. The table shows the performance of each proxy model on three mathematical reasoning benchmarks. For each benchmark, we report the Fleiss' Kappa score calculated from our human-in-the-loop experiment. Our findings reveal a correlation between the scale of the LLM proxy and its alignment with human preferences. As shown, models with higher Fleiss' Kappa scores consistently achieve higher accuracy across all benchmarks. This trend strongly suggests that larger, more capable models serve as more reliable proxies for human feedback, thereby leading to better overall performance.

Table 13: More Evaluation Result of LLM proxies with varying model scales.

| Method | Fleiss'K | GSM8K | | MATH500 | | AIME24 | |
|---|---|---|---|---|---|---|---|
| | | acc. | len. | acc. | len. | acc. | len. |
| Qwen2.5-32b-Instruct | 0.45 | 93.33 | 411.41 | 87.20 | 1177.60 | 63.33 | 1241.48 |
| Qwen2.5-72b-Instruct(ours) | 0.51 | 95.80 | 322.40 | 90.60 | 1081.87 | 73.85 | 1182.50 |
| kimi-k2-instruct | 0.68 | 96.67 | 318.53 | 92.67 | 762.60 | 76.56 | 1193.67 |

### H.2   THE DETAILS OF APPLYING INTERACTIVE REASONING ON MORE TASKS.

**Evaluation on Instruction Following Task**   We assess the interactive reasoning ability to adhere to instructions using the IFEval benchmark (Qin et al., 2024). The evaluation strictly follows the methodology proposed in the original paper. We compare the performance of the base model against our method Think-with-Me, where both are provided with a vanilla prompt with additional reminder "Think carefully about your decision", matching the interactive reasoning mechanism. In our setup, the interactive feedback is supplied by an LLM proxy. To quantify the capability, we

report the accuracy, defined as the proportion of prompts for which the responses satisfy all verifiable instructions within the prompt.

**Evaluation on Instruction Hierarchy Task**   To evaluate the interactive reasoning capacity involving prioritized instructions, we use the SEP dataset (Zverev et al., 2024). This task challenges the model to execute a main, high-priority instruction while ignoring a secondary, low-priority one. Feedback during the reasoning process is provided by an LLM proxy. To match the interactive reasoning mechanism, we test the input with an additional reminder, "Think carefully about your creation" and measure performance using two metrics: $Rob.$(robustness), which represents the proportion of low-priority instructions successfully ignored, and $Utility$, which quantifies the model's performance on the main task. The results in Figure 3b illustrate that Think-with-Me can effectively navigate conflicting instructions.

**Evaluation on Creative Task**   We explore the application of Think-with-Me in creative domains through a story generation task, evaluated on the LitBench benchmark (Fein et al., 2025). Unlike the previous tasks, this setup incorporates human-in-the-loop feedback to guide the creative writing process. The evaluation is based on a preference comparison, where outputs from our method are judged against base model DeepSeek-R1-Distilled-Qwen2.5-7B with additional requirement "Carefully consider your creation, I need creative content". Following the LitBench setup, we collected 20 samples, and the creative preferences were determined by an LLM-based evaluator (Claude-3.7-sonnet). The win rate, depicted in Figure 3c, highlights the potential of our interactive method in collaborative human-AI creative endeavors. The prompt used to judge is detailed in Box 6.

---

**Judgement prompt for Creative Task**

You are evaluating two creative writing responses (A and B) to the same writing prompt. These responses are similar to those posted on Reddit writing subreddits like r/WritingPrompts.
Your task is to predict which response would receive more upvotes from the Reddit community. Reddit users typically upvote creative writing that is engaging, original, well-written, and emotionally resonant.
When making your prediction, consider what makes content popular on Reddit:
- Originality and uniqueness of ideas
- Engaging narrative style and pacing
- Emotional impact and relatability
- Clever twists or satisfying conclusions
- Technical quality of writing
Story A:
{story_a}
Story B:
{story_b}
This is an experiment to test how well language models can predict human preferences in creative writing as expressed through Reddit's voting system.
Your verdict MUST follow this exact format:
Reasoning: [explain which response would likely get more Reddit upvotes and why]
Preferred: [A or B] (the one you predict would get more upvotes)

---

### H.3  POTENTIAL TO SACLE TO 32K WINDOW SIZE

Instead of locally deploying all models, we extend our method to the 32k context window by calling an external API for the LLM proxy service. Specifically, we use the API service of Qwen2.5-72B-Instruct (equipped with a 128k context window) as the LLM proxy, while locally deploying the 7B target reasoning model with a 32k context window on a single NVIDIA A800 GPU. From Table 14, our method still maintains the advantage in the accuracy–length trade-off across multiple challenging datasets when scaled to the 32k context window.

Table 14: The result of methods under 32k window size

| Method | MATH500 | | AIME24 | | GPQA | | Livecode | |
|---|---|---|---|---|---|---|---|---|
| | accuracy | length | accuracy | length | accuracy | length | accuracy | length |
| DeepSeek-RI-7B | 87.60 | 3463.52 | 50.00 | 5608.00 | 16.67 | 3798.23 | 22.00 | 13332.02 |
| DeepSeek-RI-14B | 90.70 | 3168.17 | 70.00 | 5948.55 | 20.00 | 3853.17 | 46.28 | 9414.02 |
| DeepSeek-RI-32B | 91.75 | 3309.54 | 70.97 | 5758.77 | 30.00 | 4358.77 | 53.14 | 10843.10 |
| QwQ-32B | 92.51 | 3265.72 | 73.33 | 8794.43 | 33.33 | 4698.03 | 54.86 | 21648.7 |
| Ours (LLM proxy) | 94.2 | 1371.23 | 75.94 | 1779.24 | 37.78 | 1496.38 | 37.30 | 1671.02 |

## H.4 CASE STUDY

This section presents a qualitative analysis through case studies to illustrate the practical advantages of our proposed method over the baseline model. We focus on two common undesirable behaviors observed in large language models: namely, "Overthinking" and "Overshot." "Overthinking" refers to scenarios where models engage in unnecessarily complex or prolonged reasoning processes before arriving at a conclusion. "Overshot" means that the model answered correctly initially, but then answered incorrectly during subsequent thinking. For each case, we will juxtapose the reasoning steps and final outputs generated by our method against those produced by the base model, thereby highlighting the improvements achieved.

### H.4.1 DETAILS OF CASE DECOMPOSITION

Here we have placed a thinking result using based model and Think-with-Mefor the same problem from GSM8K, and have broken down the thinking in detail to quickly understand the key points in the thinking results. The result is shown in Figure 6

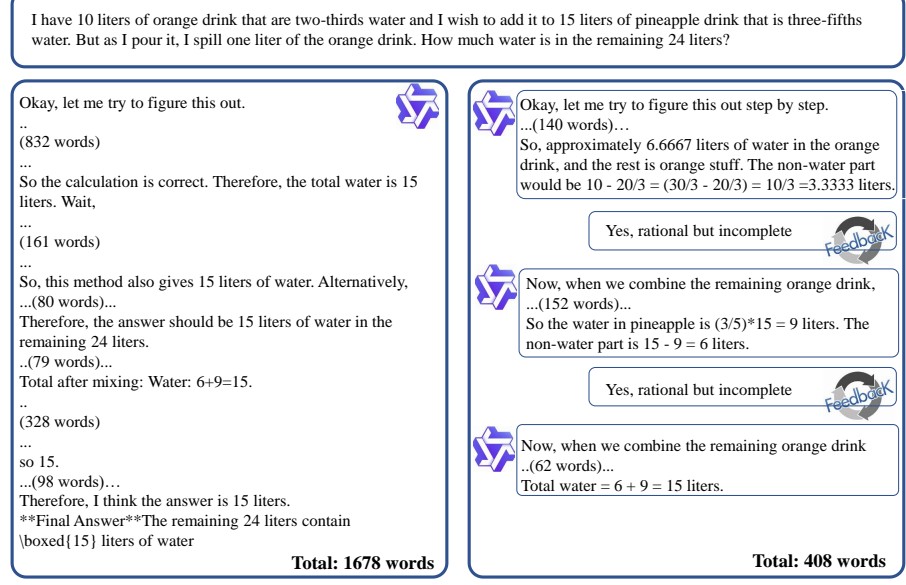

Figure 6: The typical case contains the reasoning result from DeepSeek-Distill-Qwen2.5-7B and Think-with-Me.

### H.4.2 THE CASE FOR MITIGATING THE OVERTHINKING PHENOMENON

This subsection presents a case study specifically illustrating the phenomenon of "overthinking." Box 7 and Box ?? detail the reasoning process and the resultant response generated by our proposed method, respectively. A key aspect of our approach is the integration of evaluator feedback, which allows the model to discern the precision of its thought process and subsequently make corrective adjustments. This mechanism is instrumental in mitigating "overthinking." For comparative purposes,

Box 8 displays the base model's reasoning and its answer, which evidently exhibit the "overthinking" problem when addressing the user's query.

---

**The reasoning process of Think-with-Me**

**User:**
What is $1^{(2^{235423523})}$?

**Assistant:**
Okay, so I need to figure out what $1^{(2^{235423523})}$ is. Hmm, let me break this down step by step. First, I remember that any number raised to any power is just multiplying that number by itself that many times. So, for example, $2^3 = 2 \times 2 \times 2 = 8$. But in this case, the base is 1, and the exponent is $2^{235423523}$. That exponent is a massive number, right? It's 2 raised to the power of 235,423,523. That must be an astronomically huge number. But

**Evaluator:**
4. Incorrect reasoning. The current reasoning completely does not contribute to the question, and a new reasoning is needed. Suggestions for exploring new reasoning: Focus on the properties of the number 1 when raised to any power, recognizing that $1^n = 1$ for any real number $n$. This will lead to the correct and immediate conclusion without the need to consider the magnitude of the exponent.

**Assistant:**
I know that 1 raised to any power is always 1. It's a fundamental rule in exponents. So, regardless of what the exponent is, as long as the base is 1, the result will always be 1.
Let me think about why that is the case. When you raise a number to a power, you're essentially multiplying that number by itself multiple times. For example, $1^2 = 1 \times 1 = 1$, $1^3 = 1 \times 1 \times 1 = 1$, and so on. No matter how many times you multiply 1 by itself, you'll always end up with 1.
So, even though the exponent here is a gigantic number like $2^{235423523}$, multiplying 1 by itself that many times doesn't change the outcome. It's still just 1.
I don't think there's any exception to this rule. Whether the exponent is zero, positive, negative, or even a fraction, as long as the base is 1, the result remains 1. For example, $1^0 = 1$, $1^{-1} = 1$, and $1^{1/2} = 1$. All of these still equal 1.
Therefore, applying this to the given problem, $1^{(2^{235423523})}$ must equal 1, no matter how large the exponent is.
I don't think I need to calculate the actual value of $2^{235423523}$ because it's unnecessary for solving this problem. The key insight is recognizing the base is 1, which simplifies everything.
So, in conclusion, the value of $1^{(2^{235423523})}$ is 1. </think>
The value of $1^{(2^{235423523})}$ is 1 because any number raised to any power remains 1 if the base is 1.
$\boxed{1}$

---

1836
1837
1838
1839
1840
1841
1842
1843
1844
1845
1846
1847
1848
1849
1850
1851
1852
1853
1854
1855
1856
1857
1858
1859
1860
1861
1862
1863
1864
1865
1866
1867
1868
1869
1870
1871
1872
1873
1874
1875
1876
1877
1878
1879
1880
1881
1882
1883
1884
1885
1886
1887
1888
1889

## The reasoning process of baseline model

**User:**
What is the value of

$$What\ is\ (1^{(2^{235423523})})?$$

**Assistant:**

Okay, so I need to figure out what $1^{(2^{235423523})}$ is. Hmm, let me break this down step by step. First, I know that exponentiation is when you raise a number (the base) to a certain power (the exponent). In this case, the base is 1, and the exponent is $2^{235423523}$. That seems like a really huge exponent! But wait, the base is 1, so maybe that simplifies things?

Let me recall some basic exponent rules. I remember that any number raised to any power is just that number multiplied by itself that many times. So, for example, $2^3 = 2 \times 2 \times 2 = 8$. But here, the base is 1, so no matter how many times you multiply 1 by itself, you're always going to get 1.

Wait, so $1^2 = 1 \times 1 = 1$, $1^3 = 1 \times 1 \times 1 = 1$, and so on. So, regardless of the exponent, as long as the base is 1, the result is always 1. That makes sense because multiplying 1 by itself any number of times doesn't change its value.

So, applying that to this problem, $1^{(2^{235423523})}$ should just be 1, right? Because no matter how big the exponent is, it's still just 1 multiplied by itself that many times, which is still 1.

Let me double-check to make sure I'm not missing something. Is there any situation where 1 raised to a power isn't 1? Hmm, I know that in some cases, especially with limits in calculus, $1^\infty$ is considered an indeterminate form because it can approach different values depending on the context. But in regular exponentiation with finite numbers, $1^n$ is always 1, regardless of how large $n$ is.

So, in this case, $2^{235423523}$ is a finite number, albeit a very large one. Therefore, 1 raised to that power should still be 1.

Is there another way to look at this? Maybe using logarithms or something? Let me see. If I take the logarithm of $1^{(2^{235423523})}$, it would be $\log(1^{(2^{235423523})})$. But the logarithm of 1 is 0, so that would be $0 \times (2^{235423523})$, which is still 0. Then, exponentiating back, $e^0 = 1$. So that also gives me 1.

Another thought: maybe using properties of exponents. I know that $a^{b^c} = a^{(b^c)}$, which is the same as $(a^b)^c$, but actually, that's not the case. Wait, no, $a^{b^c}$ is different from $(a^b)^c$. The first one is $a$ raised to the power of $b^c$, whereas the second is $a^b$ raised to the power of $c$, which is $a^{b \times c}$. So, in our case, it's definitely 1 raised to the power of $2^{235423523}$, which is different from $(1^2)^{235423523}$. But regardless, since the base is 1, it's still 1.

Let me also think about the properties of exponents involving 1. I remember that 1 is the multiplicative identity, so multiplying by 1 doesn't change the value. Similarly, exponentiating 1 doesn't change the value. So, no matter how many times you exponentiate 1, it's still 1.

Another angle: if I consider the general case, $1^n$ for any integer $n$, positive, negative, or zero. Well, except for $n = 0$, but even $1^0 = 1$ by definition. So, in all cases, it's 1. Therefore, even with such a massive exponent, it doesn't change the outcome.

Wait, could there be a case where $1^{\text{something}}$ isn't 1? For example, in modular arithmetic? Hmm, in modular arithmetic, sometimes things wrap around, but unless specified, regular exponentiation is just 1. So, unless we're working modulo something, which isn't indicated here, it should still be 1.

Also, thinking about the definition of exponentiation, $a^b$ is the product of $b$ copies of $a$ multiplied together. So, if $a = 1$, then no matter how many copies you have, multiplying them all together is still 1. So, that's another confirmation.

Is there a way this could be something else? Maybe if the exponent is zero? Wait, $2^{235423523}$ is definitely not zero. 2 raised to any positive integer is at least 2, so $2^{235423523}$ is a huge number, but not zero. So, $1^{(\text{that huge number})}$ is still 1.

Wait, what if the exponent was zero? Then it would be $1^0 = 1$. But since the exponent here is $2^{235423523}$, which is definitely not zero, so that doesn't apply.

Also, in some contexts, like in limits, as I mentioned before, $1^\infty$ is indeterminate, but here it's a finite exponent, so it's just 1.

Another thought: maybe if the exponent was a variable, but in this case, it's a specific number, so that's not an issue.

---

### Base model thinking and answer–Overthinking(Continue)

Wait, could I write this as $1^{2^{235423523}} = (1^2)^{235423523}$? But no, exponentiation is right-associative, so $a^{b^c} = a^{(b^c)}$ and not $(a^b)^c$. So, in this case, it's 1 raised to $2^{235423523}$, not $(1^2)^{235423523}$. But since both are 1, it still ends up as 1.

Wait, let me clarify the associativity. So, exponentiation is right-associative, meaning that $a^{b^c}$ is evaluated as $a^{(b^c)}$ and not $(a^b)^c$. So, in our case, it's definitely 1 raised to $2^{235423523}$, which is 1, and not $(1^2)^{235423523}$, which is also 1. So, in either case, it's 1, but the right-associative property just tells us the order of operations.

So, even if we considered the other way, it still results in 1. So, that doesn't change the outcome.

Is there a way this could be something other than 1? I can't think of any. All the exponent rules I know point to it being 1.

Wait, let me consider the definition of exponentiation. For any integer $a$, $a^0 = 1$ if $a \neq 0$, and $a^1 = a$. So, for $a = 1$, $1^n$ is 1 for any integer $n$. Since $2^{235423523}$ is an integer, positive, it's still 1.

Is the exponent allowed to be that large? I mean, in regular exponentiation, the exponent can be any integer, whether small or large. So, even if it's a huge number, 1 raised to it is still 1.

Wait, could there be some kind of overflow or something in computer science? But since this is a mathematical problem, not a computational one, we don't have to worry about that.

Another thought: if I use logarithms, as I did earlier, $\log(1^{(2^{235423523})}) = 0 \times (2^{235423523}) = 0$, and then $e^0 = 1$. So that also gives me 1.

Alternatively, using natural logarithm: $\ln(1^{(2^{235423523})}) = (2^{235423523}) \times \ln(1) = 0$, so exponentiating gives $e^0 = 1$. Same result.

Is there any other mathematical operation or function that might change this? For example, if I take the factorial of 1? But no, that's not relevant here. Or maybe the square root or something? The square root of 1 is 1, but that's not the operation we're dealing with here.

Wait, another angle: in some contexts, especially in abstract algebra, 1 can be the multiplicative identity in different structures. But regardless of the structure, multiplying 1 by itself any number of times will still give 1. So, even in more abstract settings, $1^n = 1$.

Also, thinking about complex numbers, 1 is still 1. Even if you raise 1 to any power in the complex plane, it's still 1. So, no change there.

Wait, maybe I'm overcomplicating this. Let me step back. The problem is $1^{(2^{235423523})}$. The base is 1, exponent is $2^{235423523}$. Since 1 raised to any power is 1, regardless of how big the exponent is, the result is 1.

Therefore, after considering all these angles, I can't find any reason why this wouldn't be 1. It seems pretty straightforward. Therefore, the Final Answer: is $\boxed{1}$.

### H.4.3 THE CASE FOR MITTIGATING OVERSHOOT

This subsection presents a case study specifically illustrating the phenomenon of "overshot," where a model continues processing despite having already reached a correct internal answer. Box 10 and Box ?? detail the reasoning process and the resultant response generated by our proposed method, respectively. A crucial feature of our method is its enhanced ability to recognize when a sufficient answer has been formulated internally, prompting a more timely termination of the redundant reasoning process. This prevents the model from unnecessarily elaborating on or potentially altering an already adequate solution, thereby effectively addressing "overshot." In contrast, Box 11 showcases the base model's reasoning and response, which clearly demonstrate the "overshot" tendency: even after an appropriate answer is likely identified internally, the model continues its deliberation, leading to a less optimal final output.

---

**The reasoning process of Think-with-Me**

**User:**
Each week, between 30 and 50 students show up for an archery class run by Betty and Wilma. Usually the students break up into groups of equal size for target practice. However, this week, Betty noticed that she could not break the students up into multiple groups of equal size. Wilma noticed that if she and Betty both joined the students in practicing, they still could not break the archers up into groups of equal size. How many students showed up to the archery class this week?

**Assistant:**
Okay, so I have this problem about Betty and Wilma's archery class. Each week, between 30 and 50 students show up. Usually, they break up into groups of equal size for target practice. But this week, Betty noticed she couldn't break them into multiple groups of equal size. Then Wilma noticed that even if she and Betty joined, they still couldn't break them up into equal groups. I need to figure out how many students showed up this week. Alright, let me break this down. So, normally, the number of students can be divided into groups of equal size. That means the number of students is divisible by some number greater than 1, right? So, the number of students must be a composite number, not a prime number. But this week, Betty couldn't break them into multiple groups of equal size. That suggests that the number of students this week is actually a prime number because prime numbers can't be divided into equal groups other than 1 and themselves. So, the number of students is prime. But

**Evaluator:**
Rational but incomplete. The current reasoning is practical and rational, but it does not contain the final answer to the question. So it is correct but incomplete.

**Assistant:**
So, Betty couldn't split them into multiple groups, implying the number is prime. Then Wilma noticed that even if both she and Betty joined, making the total number of students N + 2, they still couldn't split them into equal groups. So, N + 2 is also prime. Therefore, we're looking for a prime number N between 30 and 50 such that N + 2 is also prime. That means N and N + 2 are twin primes.
Now, let's list the prime numbers between 30 and 50:
31, 37, 41, 43, 47
Now, let's check which of these primes have another prime two greater than them:

- 31: 31 + 2 = 33, which is not prime (divisible by 3)
- 37: 37 + 2 = 39, which is not prime (divisible by 3)
- 41: 41 + 2 = 43, which is prime
- 43: 43 + 2 = 45, which is not prime (divisible by 5)
- 47: 47 + 2 = 49, which is not prime (7x7)

So, the only pair where both N and N + 2 are prime is 41 and 43.
Therefore, the number of students this week must be $\boxed{41}$.

---

```
The reasoning process of Base model
```

**User:**
Each week, between 30 and 50 students show up for an archery class run by Betty and Wilma. Usually the students break up into groups of equal size for target practice. However, this week, Betty noticed that she could not break the students up into multiple groups of equal size. Wilma noticed that if she and Betty both joined the students in practicing, they still could not break the archers up into groups of equal size. How many students showed up to the archery class this week?

**Assistant:**
Okay, so I need to figure out how many students showed up to the archery class this week. Let's see. The problem says that usually, the students break up into groups of equal size, but this week Betty noticed she couldn't do that. Then Wilma said that even if she and Betty joined, they still couldn't break them up into equal groups.

Hmm. So, let me break this down. Normally, the number of students is between 30 and 50. Let me denote the number of students as N. So, N is between 30 and 50.

Now, the students usually break up into groups of equal size. That means N is a number that can be divided into equal groups, so N must be a composite number because it has factors other than 1 and itself. But this week, Betty noticed that she couldn't break them into multiple groups of equal size. So, that means that N doesn't have any divisors other than 1 and itself? Wait, no, that would make N a prime number, but if N is prime, then the only way to break them into groups is one group or N groups, but the problem says they usually break into multiple groups, meaning more than one group but less than N. So, if N is prime, they can't break them into multiple groups, which is what Betty noticed this week.

But wait, the problem says usually they can break into groups of equal size, so N is composite. But this week, Betty noticed that she couldn't break them into multiple groups, implying that N is prime? That seems contradictory because if N is prime, usually they can break into groups, but this week they can't. Maybe I need to think differently.

Wait, no. Maybe the usual grouping is into groups of a certain size, say, groups of 2, 3, 4, etc. So, if N is a number that is composite, it can be divided into equal groups, but this week, for some reason, Betty couldn't. Maybe the number of groups she was thinking of is such that it's not a divisor of N. So, Betty was trying to divide them into a specific number of groups, which didn't work, but it's not necessarily that N is prime. It could be that N is composite, but Betty was trying to split them into a group size that doesn't divide N.

Wait, let me re-read the problem: "However, this week, Betty noticed that she could not break the students up into multiple groups of equal size." So, Betty noticed that she couldn't break them into multiple groups of equal size. So, maybe she tried to split them into, say, 2 groups or 3 groups, and it didn't work. So, the number of students is such that it can't be divided into equal groups of 2, 3, etc., but usually, they can. Hmm, so maybe N is a number that is prime? Because if N is prime, you can't split them into groups of 2, 3, etc., without having a remainder.

But wait, usually, they can break up into groups, so N is composite. So, this week, Betty noticed that she couldn't break them into multiple groups, so maybe she tried to split them into groups of a certain size, but that size doesn't divide N. But it's not necessarily that N is prime, because N could be composite but have some divisors that don't work.

Wait, maybe the key is that N is a prime number. Because if N is prime, then you can't split them into multiple equal groups unless you have 1 group or N groups. But the problem says that usually, they break into multiple groups, so N is composite. Hmm, but this week, Betty noticed that she couldn't break them into multiple groups, so maybe the number of students is such that it's prime? But that contradicts the usual situation. Wait, maybe I need to think about the problem differently.

Let me parse the problem again: "However, this week, Betty noticed that she could not break the students up into multiple groups of equal size." So, Betty tried to split them into multiple groups, but she couldn't. So, the number of students, N, is a number such that it cannot be divided into equal groups of a certain size. But usually, they can be divided into equal groups, so N is composite. So, this week, Betty tried to divide them into, say, 2 groups, but N is odd, so she couldn't. Alternatively, she tried to split them into 3 groups, but N isn't a multiple of 3. Hmm.

**The reasoning process of Base model (Continue)**

But then, Wilma noticed that if she and Betty both joined the students in practicing, they still couldn't break the archers up into groups of equal size. So, when adding 2 more students (Betty and Wilma), the new number of students is N + 2, and this number also cannot be divided into equal groups of any size. So, both N and N + 2 are numbers that cannot be divided into equal groups of multiple sizes, meaning both are prime numbers? Wait, if N is composite, but N + 2 is prime, that's possible. Or both N and N + 2 are composite but cannot be divided into equal groups? Hmm, that seems conflicting because if N is composite, it can be divided into equal groups of certain sizes, unless those specific group sizes don't divide N. Wait, maybe N is a prime number? Because if N is prime, you can't split them into multiple groups, but if N is prime, usually they can't split them either. But the problem says that usually, they can break into groups, so N is composite. Hmm, this is confusing.

Wait, let me think of it another way. If N is such that it's composite, but it's also such that N + 2 is prime. So, Betty couldn't split N into multiple groups, but N + 2 is prime, so even if Betty and Wilma join, they still can't split into multiple groups. So, maybe N is a composite number where N is one less than a prime number. So, N is composite, N + 2 is prime.

But let's test this idea. Let's list numbers between 30 and 50, and see which are composite, and then check if N + 2 is prime.

Let me list the numbers from 30 to 50:

30, 31, 32, 33, 34, 35, 36, 37, 38, 39, 40, 41, 42, 43, 44, 45, 46, 47, 48, 49, 50.

Now, composite numbers are all except the primes: 31, 37, 41, 43, 47.

So, composite numbers between 30 and 50: 30, 32, 33, 34, 35, 36, 38, 39, 40, 42, 44, 45, 46, 48, 49, 50.

Now, for each composite number N, check if N + 2 is prime.

Let's go one by one:

30: 30 + 2 = 32, which is composite. So, N + 2 is composite. Not good.

32: 32 + 2 = 34, composite. Not good.

33: 33 + 2 = 35, composite. Not good.

34: 34 + 2 = 36, composite. Not good.

35: 35 + 2 = 37, which is prime. Okay, so 35 is a candidate.

36: 36 + 2 = 38, composite. Not good.

38: 38 + 2 = 40, composite. Not good.

39: 39 + 2 = 41, which is prime. Okay, 39 is another candidate.

40: 40 + 2 = 42, composite. Not good.

42: 42 + 2 = 44, composite. Not good.

44: 44 + 2 = 46, composite. Not good.

45: 45 + 2 = 47, which is prime. Okay, 45 is another candidate.

46: 46 + 2 = 48, composite. Not good.

48: 48 + 2 = 50, composite. Not good.

49: 49 + 2 = 51, composite. Not good.

50: 50 + 2 = 52, composite. Not good.

So, the composite numbers N in 30-50 where N + 2 is prime are 35, 39, and 45.

Now, let's check these numbers.

First, N = 35. 35 is 5 * 7, so it's composite. N + 2 = 37, which is prime. So, Betty couldn't split 35 into multiple equal groups because 35 is composite but maybe she tried to split it into groups that don't divide 35, but wait, 35 can be divided into 5 groups of 7 or 7 groups of 5. So, actually, she should be able to split them, but the problem says she couldn't. Hmm, that seems contradictory.

Wait, maybe Betty tried to split them into 2 groups, but 35 is odd, so you can't split into 2 equal groups. Similarly, 35 is not divisible by 3: 35 / 3 is about 11.666, so not an integer. So, maybe Betty tried to split them into groups of 2 or 3, which didn't work, so she couldn't. So, N = 35 is a possible answer.

Similarly, N = 39. 39 is 3 * 13. So, you can split them into 3 groups of 13 or 13 groups of 3. So, usually, they can split into multiple groups. But this week, Betty couldn't.

---

### The Reasoning Process of Base Model(Continue)

So, perhaps Betty tried to split them into 2 groups, but 39 is odd, so can't split into 2 equal groups. Similarly, 39 is not divisible by 4: 39 / 4 is 9.75, so not integer. So, maybe she tried to split into 2 or 4 groups, which didn't work. So, N = 39 is another possible answer.

Similarly, N = 45. 45 is 9 * 5, so it can be split into 5 groups of 9 or 9 groups of 5. So, usually, they can split into multiple groups. But this week, Betty couldn't. Maybe she tried to split into 2 groups, which is not possible because 45 is odd. Similarly, 45 isn't divisible by 4: 45 / 4 = 11.25, so no. So, N = 45 is another candidate.

So, we have three candidates: 35, 39, and 45.

Now, the problem says that this week, Betty couldn't break them into multiple groups, and Wilma noticed that if they both joined, they still couldn't break them into equal groups. So, N is such that N is composite, and N + 2 is prime.

But we have multiple possibilities. How do we choose between 35, 39, and 45?

Wait, let's think about the initial condition: "However, this week, Betty noticed that she could not break the students up into multiple groups of equal size."

So, if Betty couldn't break them into multiple groups, it means that N is a composite number, but she couldn't split them into a specific number of groups, say, k groups, where k is a divisor of N. But N is composite, so it has some divisors other than 1 and itself. So, if N is composite, it can be split into multiple groups, but Betty couldn't.

Wait, maybe Betty tried to split them into a prime number of groups, but N is not a multiple of that prime. For example, if she tried to split them into 2 groups, which is prime, but N is odd, so it can't be split into 2 equal groups.

But then, if N is even, you can split into 2 groups, so she could have done that. So, N must be odd, because she couldn't split into 2 groups. So, N is odd.

Looking back at our candidates: 35, 39, 45 are all odd, so that's consistent.

Now, let's see if N is 35, 39, or 45, which one is the correct answer.

Wait, maybe we can think of another condition. The problem says that when Betty and Wilma join, so N + 2, they still can't break the archers into equal groups. So, N + 2 is prime, as we saw earlier.

But also, N must be such that N is composite, but N + 2 is prime.

So, in this case, all our candidates satisfy that N is composite, N + 2 is prime.

But perhaps there's another condition: that N is such that it can't be split into multiple groups because it's a prime power? Hmm, not necessarily.

Wait, let me think differently. Maybe N is a composite number, but N is a semiprime, meaning it's the product of two primes, so it can only be split into two groups of equal size or into groups of size equal to the prime factors.

So, if N is a semiprime, it can be split into two equal groups or into groups equal to its prime factors.

But if Betty couldn't split them into multiple groups, maybe she tried to split them into more than two groups, but N is a semiprime, so it can't be split into more than two equal groups unless those groups are size 1 or N.

Wait, but semiprimes can sometimes be split into more groups if the number of groups is a divisor. For example, 15 is 3 * 5, so it can be split into 3 groups of 5 or 5 groups of 3. So, if N is a semiprime, it can be split into two equal groups or into groups equal to the prime factors. So, if Betty tried to split them into 3 groups, but N isn't a multiple of 3, then she couldn't. So, if N is a semiprime, and 3 doesn't divide N, then Betty couldn't split them into 3 groups.

Similarly, if N is 35, which is 5 * 7, she couldn't split into 3 groups because 35 isn't a multiple of 3.

Similarly, N = 39 is 3 * 13, so she couldn't split into 5 groups because 39 isn't a multiple of 5.

Similarly, N = 45 is 9 * 5, which is $3^2 * 5$, so she couldn't split into 7 groups or something else.

Wait, maybe the key is that N is such that it's a semiprime where the two prime factors are distinct and greater than 2, so that N is odd and can't be split into 2 groups, and also, N + 2 is prime.

But I think all three numbers 35, 39, 45 satisfy that.

Wait, maybe the problem is that N is a composite number where N is not a multiple of 2, 3, or 5, but N + 2 is prime.

---

**The reasoning process of Base model (Continue)**

Wait, 35 is 5 * 7, so it's a multiple of 5. 39 is 3 * 13, so it's a multiple of 3. 45 is 9 * 5, which is $3^2 * 5$.

So, perhaps N is a composite number that is a multiple of 3 or 5, but N + 2 is prime. So, 35, 39, and 45 all fit that.

Hmm.

Alternatively, maybe the problem is that N is such that it's a composite number, and N + 2 is a prime, but N is also a perfect square or something.

Wait, 35 is not a perfect square, 39 is not, 45 is not.

Alternatively, maybe N is a composite number such that N is one less than a prime, but N + 2 is prime. Wait, no, N + 2 is prime in our case.

Wait, maybe I should look for N such that N is composite, N + 2 is prime, and N is the smallest such number in the range 30-50. But 35 is the first one.

Wait, let's check N = 35: 35 is composite, 35 + 2 = 37 is prime. So, that fits.

But let's see if N = 35, can Betty split them into multiple groups? 35 can be split into 5 groups of 7 or 7 groups of 5. So, if Betty tried to split them into 5 groups, she could, but if she tried to split them into 3 groups, she couldn't. So, if Betty couldn't split them, maybe she tried to split them into 3 groups, which doesn't work because 35 isn't divisible by 3.

Similarly, N = 39: 39 can be split into 3 groups of 13 or 13 groups of 3. So, if Betty tried to split them into 5 groups, which 39 isn't divisible by, she couldn't.

N = 45: 45 can be split into 5 groups of 9 or 9 groups of 5. So, if Betty tried to split them into 7 groups, which 45 isn't divisible by, she couldn't.

So, all three numbers fit the conditions.

Wait, so how do we choose between them? The problem says that this week, Betty couldn't break them into multiple groups of equal size. So, N is composite, but Betty couldn't split them, meaning that N is composite, but N isn't divisible by the number of groups she tried to split them into.

But since the problem doesn't specify what group size Betty was trying, maybe we need another condition.

Wait, maybe N is such that it's composite, but N is also a prime power? But 35 is 5*7, 39 is 3*13, 45 is $3^2 * 5$. So, they're all composite with multiple prime factors.

Wait, maybe N is such that N is a multiple of a prime, but N + 2 is prime. So, N is a multiple of 5 or 3 or something, and N + 2 is prime.

But all three numbers are multiples of 3 or 5.

Wait, maybe N is the only composite number in the range where N + 2 is prime. But no, all three are.

Alternatively, perhaps N is such that N is a Carmichael number or something, but that might be overcomplicating.

Wait, maybe I need to think about the fact that when Betty and Wilma join, making N + 2, which is prime, so they still can't split them into equal groups. So, N + 2 is prime, so they can't split them into multiple groups, which is consistent.

But N is composite, so it can be split into equal groups, but Betty couldn't split them into equal groups, so maybe she tried to split them into a specific number of groups that didn't divide N.

Wait, but all three numbers N = 35, 39, 45 can be split into multiple equal groups, except when Betty tried to split them into groups that don't divide N.

But the problem doesn't specify what group size Betty was trying, so maybe all three numbers satisfy the condition.

But the problem is asking for a specific number, so I must be missing something.

Wait, let's reread the problem:

"How many students showed up to the archery class this week?"

Betty noticed that she couldn't break the students up into multiple groups of equal size.

---

---

**The reasoning process of Base model (Continue)**

Wilma noticed that if she and Betty both joined the students in practicing, they still couldn't break the archers up into groups of equal size.

So, perhaps the key is that N is composite, so Betty could have broken them into equal groups, but she couldn't. That suggests that N is composite, but Betty was trying to split them into a prime number of groups, which didn't divide N.

But N is composite, so it must have some prime factors. So, if Betty tried to split them into a prime number of groups, but N isn't divisible by that prime, then she couldn't.

So, for example, if Betty tried to split them into 2 groups, but N is odd, so she couldn't. Similarly, if she tried to split them into 3 groups, but N isn't a multiple of 3.

So, in our case, N is either 35, 39, or 45.

Now, let's check if N is 35: 35 is 5 * 7, so it can be split into 5 groups of 7 or 7 groups of 5. So, if Betty tried to split them into 2, 3, 4, 6, etc., groups, she couldn't. So, she couldn't break them into multiple groups of equal size. So, that works.

Similarly, for N = 39: 39 is 3 * 13. So, Betty couldn't split them into 2, 4, 5, etc., groups, but she could into 3 or 13. So, if she tried to split them into 2 groups, she couldn't. So, that works too.

N = 45: 45 is 9 * 5, so it's $3^2 * 5$. So, Betty couldn't split them into 2, 4, 7, etc., groups, but she could into 3, 5, 9, or 15. So, if she tried to split them into 2 groups, she couldn't. So, that works too.

So, all three numbers fit.

Wait, maybe the problem is that N is such that N is composite, but N is also a perfect square or something.

But 35 is not a perfect square, 39 isn't, 45 isn't.

Alternatively, maybe N is a multiple of 3 and 5, like 45, so it's a multiple of 15. But 35 is a multiple of 5 and 7, and 39 is a multiple of 3 and 13.

Alternatively, maybe N is the only number in that range where N is composite, and N + 2 is prime, but also N is such that N is the product of two primes, both greater than 2.

So, 35 is 5*7, both primes greater than 2. 39 is 3*13, 3 is a prime, but 3 is not greater than 2 in terms of being odd, but it's still a prime. 45 is $3^2 * 5$, so it's not a product of two distinct primes.

Wait, so 35 is the product of two distinct primes, both greater than 2. 39 is the product of two distinct primes, one of which is 3. 45 is the product of a prime squared and another prime.

So, perhaps the answer is 35, because it's the only one where both prime factors are greater than 2. Hmm, but that might be a stretch.

Alternatively, maybe the problem is expecting N to be a number such that N is composite, N + 2 is prime, and N is the smallest such number in the range.

But 35 is the first one in the range 30-50.

Wait, let me think of another way. Maybe N is a composite number where N is not a multiple of 2, 3, or 5, but that seems restrictive.

Wait, 35 is 5*7, so it's a multiple of 5. 39 is 3*13, multiple of 3. 45 is 9*5, multiple of 3 and 5.

Wait, so they all are multiples of 3 or 5. So, maybe all three numbers are possible.

But the problem is expecting a specific answer, so maybe I need to think differently.

Wait, let me check the initial conditions again.

Betty couldn't split them into multiple groups of equal size. So, N is composite, so Betty should have been able to split them, but she couldn't. So, maybe she tried to split them into a number of groups equal to a prime, but N is not a multiple of that prime.

Similarly, when Wilma joins, N + 2 is prime, so they still can't split them into multiple groups.

But is there another condition?

Wait, the key might be that N is such that it's composite, but N is also a prime power. Wait, 35 is not a prime power, 39 isn't, 45 isn't.

Alternatively, maybe N is a Carmichael number, but 35 is 5*7, which is not a Carmichael number because 5 - 1 doesn't divide 35 - 1, which is 34. 5 - 1 = 4, 34 divided by 4 is 8.5, which is not an integer. Similarly, 7 - 1 = 6, 34 divided by 6 is about 5.666, not integer. So, 35 isn't a Carmichael number.

### The Reasoning Process of Base Model (Continue)

Similarly, 39: 3 - 1 = 2, 38 divided by 2 is 19, which is integer. 13 - 1 = 12, 38 divided by 12 is about 3.166, not integer. So, 39 isn't a Carmichael number either.

45: 3 - 1 = 2, 44 divided by 2 is 22, integer. 5 - 1 = 4, 44 divided by 4 is 11, integer. So, 45 is a Carmichael number? Wait, 45 is $3^2 * 5$. For a number to be a Carmichael number, it must be square-free and satisfy Korselt's criterion, which is that for all prime divisors p of n, p - 1 divides n - 1.

So, 45 is not square-free because it's $3^2 * 5$. So, it's not square-free, so it can't be a Carmichael number. So, that idea doesn't hold.

Hmm. Maybe I need to think of another approach.

Wait, let me consider the initial statement: usually, they can break into groups of equal size, but this week they couldn't. So, N is composite, but this week, Betty couldn't. So, maybe N is a composite number, but it's a prime power, like $2^5 = 32$, which is composite, but it can only be split into 2 groups of 16 or 16 groups of 2. So, if Betty tried to split them into 3 groups, she couldn't. So, N = 32 is a possibility.

But 32 is in the range 30-50, and 32 + 2 = 34, which is composite. So, does N = 32 fit?

So, N = 32 is $2^5$, composite. Betty couldn't split them into multiple groups, maybe 3 groups, since 32 isn't divisible by 3. And when Wilma joins, N becomes 34, which is composite, so they still can't split into equal groups because 34 can be split into 2 groups of 17 or 17 groups of 2, but 17 is prime, so they can't split them into more than 2 groups unless they try to split into 17 groups, which is impractical.

Wait, but 34 is 2 * 17, so they can split into 2 groups of 17 or 17 groups of 2. So, if Betty and Wilma join, making 34, they can split into 2 groups of 17 or 17 groups of 2. So, they can split them, which contradicts the problem statement that they still couldn't break them into equal groups.

So, N = 32 is out because when they add 2, N + 2 = 34 is composite, but they can split them into equal groups.

So, N + 2 must be prime for them to still not be able to split them into equal groups. So, N + 2 must be prime, so N + 2 is prime. So, N must be such that N is composite, and N + 2 is prime.

So, that brings us back to 35, 39, 45.

So, all three numbers satisfy N composite, N + 2 prime.

But the problem is asking for a specific number. So, maybe I need to think of another condition.

Wait, let me think about the fact that N is such that it's composite, and it's the product of two primes, both of which are greater than 2. So, 35 is 5 * 7, both primes greater than 2. 39 is 3 * 13, 3 is a prime, but 3 is not greater than 2 in a meaningful way, but still, it's a prime. 45 is $3^2 * 5$, so it's not a product of two distinct primes.

So, maybe 35 is the answer because it's the product of two primes, both greater than 2, so it can't be split into 2 groups, and when adding 2, it becomes 37, which is prime, so can't be split into multiple groups.

Alternatively, maybe 39 is the answer because it's the product of 3 and 13, both primes, and when adding 2, it becomes 41, which is prime.

==Wait, 39 + 2 is 41, which is prime.==

Similarly, 35 + 2 is 37, which is prime.

45 + 2 is 47, which is prime.

So, all three numbers fit.

Wait, maybe the key is that N is the only composite number in that range where N is composite, and N + 2 is prime, and N is a semiprime.

But 35, 39, and 45 are all semiprimes or products of primes with powers.

Wait, maybe the problem is expecting N to be 35 because 35 is the first such number in the range where N is composite, N + 2 is prime, and N is a product of two distinct primes, both greater than 2.

Alternatively, maybe the problem is expecting the answer 35, 39, or 45, but it's expecting one.

Wait, maybe I need to think of another angle.

---

**The reasoning process of Base model (Continue)**

Wait, the problem says that usually, they can break into groups of equal size, so N is composite, but this week, Betty couldn't. So, N is composite, but Betty couldn't split into multiple groups, which might mean that N is a prime power, like $2^5 = 32, 3^3 = 27$, but 27 is below 30.

Wait, 32 is in the range, 32 is composite, but it's a prime power. So, can you split 32 into multiple equal groups? Yes, into 2 groups of 16, 4 groups of 8, 8 groups of 4, 16 groups of 2. So, Betty could split them into multiple groups, but the problem says she couldn't. So, that contradicts.

So, N is composite, but Betty couldn't split them into equal groups, which suggests that N is composite, but it's not a multiple of some number that Betty tried to split them into. So, N is composite, but Betty tried a prime number of groups, which didn't divide N.

So, for example, if N is 35, Betty couldn't split into 3 groups because 3 doesn't divide 35.

If N is 39, Betty couldn't split into 5 groups because 5 doesn't divide 39.

If N is 45, Betty couldn't split into 7 groups because 7 doesn't divide 45.

So, all three fit.

Wait, maybe the problem is that N is such that N is composite, N + 2 is prime, and N is square-free.

35 is square-free, 39 is square-free, 45 is not square-free because it's 9 * 5. So, 45 is not square-free.

So, maybe N is square-free. Then, 35 and 39 are square-free, 45 isn't.

So, perhaps N is 35 or 39.

But the problem is expecting a specific number.

Wait, maybe 35 is the answer because it's the smallest composite number in the range where N + 2 is prime.

But 35 is not the smallest, 30 is composite, but 30 + 2 = 32, which is composite. So, 30 doesn't fit.

Similarly, 32 doesn't fit because N + 2 is composite.

So, 35 is the first composite number where N + 2 is prime.

But 35 is 5 * 7, so it's square-free.

Similarly, 39 is 3 * 13, square-free.

So, both are square-free.

Hmm, maybe both are possible answers, but the problem is expecting one.

Wait, let me check the original problem again.

"How many students showed up to the archery class this week?

Betty noticed that she could not break the students up into multiple groups of equal size.

Wilma noticed that if she and Betty both joined the students in practicing, they still could not break the archers up into groups of equal size. How many students showed up to the archery class this week?"

Wait, so the problem is presented as a single answer. So, perhaps I need to think of another condition.

Wait, maybe N is such that N is composite, N + 2 is prime, and N is a multiple of 3 or 5.

But all three numbers 35, 39, 45 are multiples of 3 or 5.

Alternatively, maybe N is such that N is the product of two primes, both of which are greater than 2, making N square-free and odd.

So, 35 is 5 * 7, both primes greater than 2.

39 is 3 * 13, 3 is not greater than 2, but still, it's a prime.

45 is $3^2 * 5$, not square-free.

So, 35 is the only one where both prime factors are greater than 2.

Therefore, N = 35.

So, I think the answer is 35.

Final Answer The number of students who showed up to the archery class this week is $\boxed{35}$.

## I   INTERACTIVE INTERFACE

We have implemented a graphical interactive interface that is compatible with Think-with-Me reasoning paradigm for inserting external feedback as needed during the reasoning process: **1.** users **choose to have LLM proxy automatically** evaluate or **manually provide feedback** at the beginning of the reasoning stage, in Figure 7. **2.** The interactive interface **supports fine-grained reasoning feedback** based on multi-criteria and **optional suggestions for subsequent reasoning**, in Figure 8. **3.** The target model **generates </think> to terminate the reasoning** ( or be forced to insert </think> after maximal interactions) and provides the answer to the question, in Figure 9.

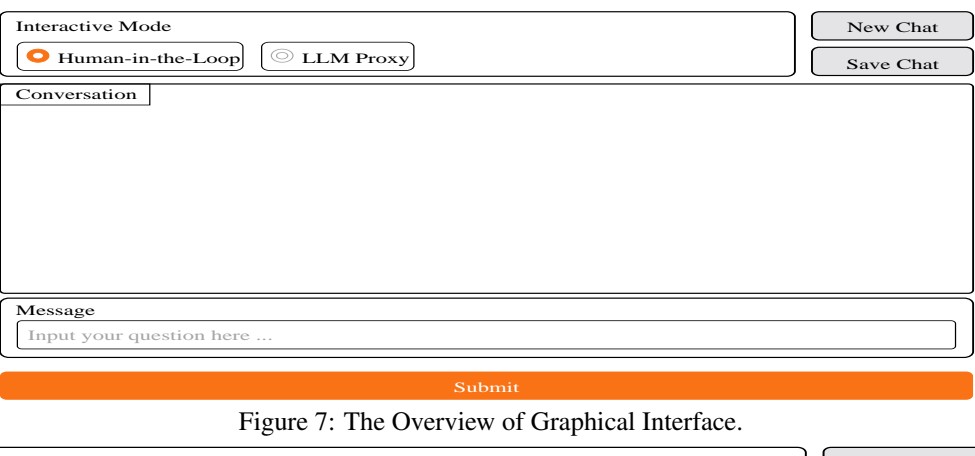

Figure 7: The Overview of Graphical Interface.

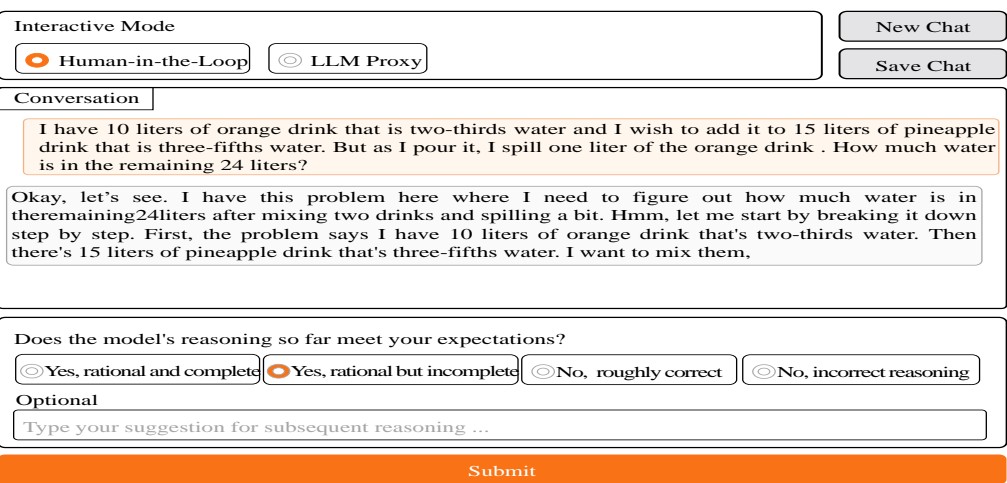

Figure 8: Four feedback options after halting the thinking process by specific transitional conjunctions.

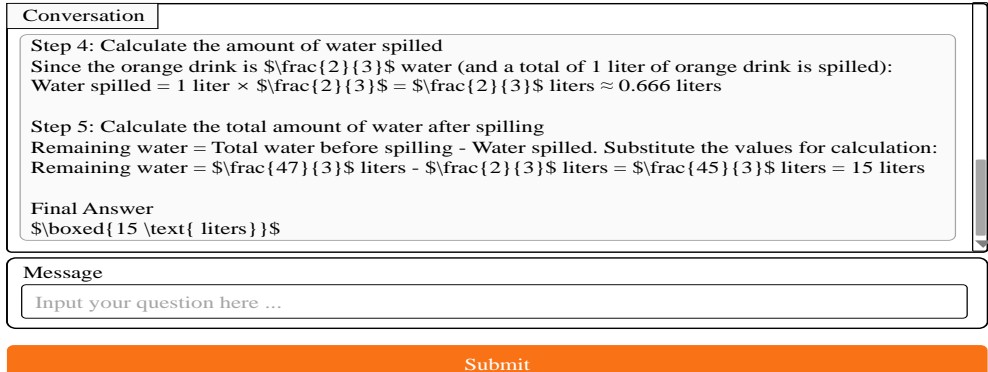

Figure 9: Get the final answer to the given questions after generating </think>.

## J  LIMITATION AND FUTURE WORK

### J.1  LIMITATION

**Incorrect external feedback from LLM proxy can degrade performance.** Owing to the known self-bias in LLMs (Wataoka et al.), the LLM proxy may produce misleading feedback that distorts subsequent reasoning. Besides, if human subjectivity is too strong or the LLM's performance is poor, the resulting erroneous feedback affects subsequent reasoning. Although our target reasoning model is trained with a composite reward that includes an accuracy term to help it implicitly down-weight harmful feedback, our method lacks an explicit, built-in mechanism to correct erroneous external feedback. Consequently, Think-with-Me remains vulnerable to error or low-quality external feedback.

**Lack of Long context processing capability.** Despite significant advancements in large language models' long-text processing abilities and acceptable window lengths (Liu et al., 2025b), the issue of information loss in long-context processing remains prevalent. Besides, it is hard for humans to process long contexts under cognitive pressure (Stadler et al., 2024). Although the reasoning length is absolutely reduced compared to the base model, the lack of long context processing capability affects the evaluation of long reasoning in complex task processing, subsequently undermining feedback accuracy.

### J.2  FUTURE WORK

**Feedback Verification Module.** Integrate a lightweight feedback verification module, like multi-proxy consensus voting or factuality checking via retrieval-augmentation, to filter misleading feedback. Further refine the composite reward with an explicit feedback quality term, and explore contrastive training to enhance the model's robustness against biased/low-quality signals.

**Leverage context compression.** Leverage hierarchical context compression, like key information distillation and long-context attention optimizations, like sliding window with memory enhancement, to summarize the reasoning content while mitigating information loss before making feedback. Design human-AI collaborative evaluation paradigms, like structured reasoning segment reviews, to reduce cognitive burden in long reasoning assessments.

## USE OF LARGE LANGUAGE MODELS DISCLOSURE

In accordance with the ICLR 2026 policy on LLM usage, we disclose that our study did not use any LLM to generate scientific content or perform major experiments. The only use of an LLM (ChatGPT-5) was to polish the English writing and improve presentation quality; all core methodology, experiments, and analyses were authored and verified by the human authors.

