# OpenReview forum: "Beyond Model Scaling: Test-Time Intervention for Efficient Deep Reasoning"
_ICLR.cc/2026/Conference — Submitted to ICLR 2026_

### Official Review · Reviewer_Zmbd · 2025-11-01

**Soundness:** 2
**Presentation:** 3
**Contribution:** 2
**Rating:** 4
**Confidence:** 4

**Summary:**

- This paper propose Think-with-Me, a test-time interactive reasoning paradigm that introduces external feedback intervention into the reasoning process.
- The approach achieves a better balance between accuracy and reasoning length under limited context windows compared with a series of baselines.

**Strengths:**

- The proposed idea is interesting and demonstrates improvements in both performance and reasoning length compared to the original model.

- The paper is overall well-written and clearly presented.

**Weaknesses:**

- The baselines used in the experiments are relatively simple. There exists a substantial body of work focusing on reducing reasoning length while maintaining or improving accuracy; more relevant and competitive baselines should be included for a fairer comparison.

- The proposed interactive reasoning framework involves multiple models (and possibly human) interactions and introduces pauses during inference. Therefore, in addition to reasoning length and accuracy, the evaluation should also consider the actual total inference time and deployment cost (e.g., computation and resource usage). The core concern is not merely how long the reasoning chain is, but whether the proposed method reduces overall inference time and resource requirements.

- Novelty: The core mechanism described in Algorithm 1 (Line 12) appears closely related to early stopping methods (e.g., Dynamic Early Exit in Reasoning Models). The paper does not fully clarify the differences between the proposed approach and these existing works. The authors should also explicitly discuss distinctions from other related adaptive reasoning or calibration methods such as SEAL and s1: Simple Test-time Scaling . Furthermore, comparisons with earlier multi-agent or multi-model interaction methods like think-critic are recommended.

**Questions:**

- In Proposition 1, is the claim always guaranteed to hold? Why does the uncertainty necessarily decrease under the proposed formulation?

---

> ### Author Response · Authors · 2025-11-22
> **Responses to reviewer Zmbd [1/3]**
>
> We sincerely appreciate the reviewer's thoughtful comments and valuable suggestions. We organized all reviewer's questions and suggestions and provided point-by-point responses:
> >Q1: The baselines used in the experiments are relatively simple. There exists a substantial body of work focusing on reducing reasoning length while maintaining or improving accuracy; more relevant and competitive baselines should be included for a fairer comparison. The authors should also explicitly discuss distinctions from other related adaptive reasoning or calibration methods such as SEAL and s1: Simple Test-time Scaling. Furthermore, comparisons with earlier multi-agent or multi-model interaction methods like think-critic are recommended.
>
> **A1.**  Thank you for the detailed comment. We include more strong baselines and demonstare Think-with-Me achieves state-of-the-art in accuracy–length trade-off:
>
> * **The comparison result of expanding baselines.** We add strong and representive long2short baselines across both paradigms—single-model optimization: s1-32b[r1], SEAL[r2], DEER[r3] and multi-model collaboration: Specreason[r8]. From Table A, Think-with-Me **consistently attains competitive or superior accuracy while significantly reducing reasoning length** across datasets, demonstrating its efficiency.
>
> * **Think-with-Me is an interactive intervention mechanism based on LRM's intrinsic reasoning behavior.** Existing efficient reasoning methods fall into single-model optimization and multi-model collaboration:
>     * **Single-model optimization** methods rely on post-training or test-time controls from numerical signals, lacking fine-grained semantic steering.
>     * **multi-model collaboration** methods can boost accuracy but add coordination or computational complexity.
>
> Based on LRMs' inherent reasoning behaviors， Think-with-Me is an **interactive intervention mechanism** that applies external **semantic feedback** to **adapt reasoning depth and direction** during the reasoning process, achieving efficient correction **within a single model**.
>
> We update the **RELATED WORK** (in Appendix A) with a systematic discussion and comparison of recent efficient reasoning approaches and other relevant parts, including **1. ** INTRODUCTION** and **4. EXPERIMENT**. We sincerely hope our clarifications above have addressed your concerns and can help you re-assess our work.
>
>
> Table A. The result of the comparison. Best and second-best results are in **bold** and \_underlined\_, respectively. An LLm proxy is applied to provide feedback in Think-with-Me.
> |                | MATH500       |              | AIME24        |              | GPQA          |              | LiveCode      |              |
> |----------------|---------------|--------------|---------------|--------------|---------------|--------------|---------------|--------------|
> |                | accuracy           | length      | accuracy           | length       | accuracy           | length       | accuracy           | length       |
> | DeepSeek-RI-7B | 84.05         | 1864.33      | 40.00         | 3374.50      | 22.25         | 5020.86      | 22.29         | 6305.82      |
> | s1-32b[r1]      | 90.00         | 2922.14      | \_55.00\_         | 6047.17      | **68.00**     | 3888.79      | 30.00         | 4019.50       |
> | SEAL[r2]           | 84.00         | \_1096.87\_      | 51.35         | \_2203.53\_     | 57.33         | 3338.8       | \_31.30\_  | 5987.83      |
> | DEER[r3]           | 87.00         | 1280.75      | 54.06         | 4127.76      | 63.33         | \_3136.17\_      | 30.80         | \_3487.17\_      |
> | specreason[r4]     | **92.00**         | 2380.78      | 50.00         | 6278.55      | 50.51         | 4450.72      | 30.00         | 6337.84      |
> | Think-with-Me(Ours)           |\_90.60\_  | **1081.87**      | **73.85**         | **1182.50**      | \_65.22\_  | **1136.38**      | **34.86**         | **1248.64** |
>
> [r1] s1: simple test-time scaling. EMNLP, 2025.
> [r2] SEAL: Steerable Reasoning Calibration of Large Language Models for Free. COLM, 2025.
> [r3] Dynamic Early Exit in Reasoning Models. Arxiv, 2025.
> [r4] Specreason: Fast and accurate inference-time compute via speculative reasoning. Nips, 2025.

---

> ### Author Response · Authors · 2025-11-22
> **Responses to reviewer Zmbd [2/3]**
>
> >Q2：The proposed interactive reasoning framework involves multiple models (and possibly human) interactions and introduces pauses during inference. Therefore, in addition to reasoning length and accuracy, the evaluation should also consider the actual total inference time and deployment cost (e.g., computation and resource usage). The core concern is not merely how long the reasoning chain is, but whether the proposed method reduces overall inference time and resource requirements.
>
> **A2.** Thank you for the detailed comment. We address the concern by reporting comprehensive **total inference time(s)** and **deployment GPU requirement(MB)** for compared methods, confirming our framework's practical advantages:
>
> * **Our method has computational resource flexibility**. From Table B, while the LLM-proxy setting requires **additional GPU memory (147456 MB) due to the local deployment of Qwen-72B-Instruct** for the controlled LLM proxy, this is not necessary for realistic deployment. In practice, the feedback service can be provided by an external **API** and the computational overhead is solely the target 7B reasoning model, requiring only **40,960 MB**of GPU memory.
>
> * **Think-with-Me exhibits practical utility.**  From Table B, Think-with-Me exhibits a **small time increment (or even shorter time)** compared to the base model, demonstrating its practicality.
>
>
> We supplement these experimental details in the **Appendix F** and reference them in section 4.2 of the main text. We sincerely hope our clarifications above have addressed your concerns and can help you re-assess our work.
>
> Table B. The average reasoning time and computation resource requirement of methods across datasets. For **time cost**, we report the total time for the method to finish the reasoning process for one question. For **computational cost**, we report the realistic GPU requirement of methods developed by **vLLM**.Best and second-best results are in **bold** and \_underlined\_, respectively.
> | Method                  | MATH500|  | AIME24  | | GPQA  || LiveCode |  |
> | ----------------------- | --------------------- | --------------- | ----------------- | ---------- | --------------- | -------- | ------------------- | ------------ |
> |                         |          total_time(s)             |   GPU(MB)              |               total_time(s)             |   GPU(MB)         |            total_time(s)             |   GPU(MB)        |                    total_time(s)             |   GPU(MB)              |
> | base                    | 35.48                 | **40960**           | **49.54**             | **40960**      | 79.48           | **40960**    | 72.51               | **40960**       |
> | s1-32b(8k)              | 127.71                | 73728           | 226.28            | 73728      | 170.47          | 73728    | 176.75              | 73728        |
> | DEER                    | **21.75**                 |       \_65536\_          | 67.73             |       \_65536\_     | 47.26           |  \_65536\_ |             \_53.90\_         |         \_65536\_   |
> | SEAL                    | 76.43                 |         **40960**        | 247.40            |    **40960**     | 146.21          |   **40960**     | 254.14              |   **40960**          |
> | specreason              | 61.38                 |        163842         | 250.09            |       163842      | 180.86          |     163842      | 208.46              |     163842          |
> | ours(LLM proxy)         | 36.38                 | 188416          | 62.27             | 188416      | \_36.39\_           | 188416    | **39.92**               | 188416       |
> | ours(human-in-the-loop) | \_31.21\_                | **40960**           | \_55.48\_             | **40960**      | **35.43**           | **40960**    | 59.85               | **40960**      |

---

> ### Author Response · Authors · 2025-11-22
> **Responses to reviewer Zmbd [3/3]**
>
> >Q3: The core mechanism described in Algorithm 1 (Line 12) appears closely related to early stopping methods (e.g., Dynamic Early Exit in Reasoning Models). The paper does not fully clarify the differences between the proposed approach and these existing works.
>
> **A3.** Thank you for the detailed comment. We have revised the Related Work section to provide a more comprehensive discussion and clarify our contributions：
>
> * **Algorithm 1, Line 12 is a termination safeguard** that forces the model to output </think> when the maximum reasoning steps are exceeded, preventing infinite loops. As shown in Table 3, the trained model **self-terminates over 80%** of the time, making the forced stop rarely necessary. The core contribution remains the interactive reasoning mechanism for LRMs.
>
> * **The distinction from related reasoning Methods.** The Think-with-Me framework is an in-process interactive intervention mechanism, rooted in reasoning behavior analysis of LRMs. Unlike **early exit** strategies that adopt a **single strategy (termination) to control reasoning**, which may be detrimental to complex problems, **Think-with-Me terminates** or **extends reasoning** based on the model's current reasoning state. This reduces redundant reasoning, **balances accuracy and reasoning length**, and broadens the application boundaries and practical value of LRMs.
>
> The comparison with the early-stop method, **DEER**, is in **Table A**. We follow the relevant response in Q1 to update the paper.
>
> >Q4:In Proposition 1, is the claim always guaranteed to hold? Why does the uncertainty necessarily decrease under the proposed formulation?
>
> **A4.** Thank you for your insightful comment. **Proposition 1 is always hold** and theoretically **explains the working mechanism of Think-with-Me**：
>
> * **Proposition 1 Always Holds (Proven via Fundamental Information Theory)**
>
>     Define key terms first:
>     - $T$: The model’s target (e.g., next reasoning segment, final answer),
>     - $C$: The current generated context,
>     - $F$: External information (feedback),
>     - $\theta$: Model parameters,
>     - $H(\cdot \mid \cdot; \theta)$: Conditional entropy (measures uncertainty of the first variable given the second, parameterized by $\theta$,
>     - $I(\cdot; \cdot \mid \cdot; \theta)$: Conditional mutual information (measures how much information one variable provides about another, given a third variable, parameterized by $\theta$.
>
>
>     #### Proof
>     1. **Start with a fundamental information theory result**: The non-negativity of conditional mutual information guarantees $I(T; F \mid C; \theta) \geq 0$ for all valid $T, C, F, \theta$.
>
>     2. **Use the definition of conditional mutual information**: By definition, conditional mutual information satisfies:
>        $$ I(T; F \mid C; \theta) = H(T \mid C; \theta) - H(T \mid C, F; \theta) $$
>
>     3. **Rearrange to prove Proposition 1**: Rearranging the equation above gives:
>        $$ H(T \mid C, F; \theta) = H(T \mid C; \theta) - I(T; F \mid C; \theta) $$
>        Since $( I(T; F \mid C; \theta) \geq 0 )$, substituting this into the equation directly proves $H(T \mid C, F; \theta) \leq H(T \mid C; \theta)$, Proposition 1 holds.
>
>     4. **Equality condition**: Equality only occurs when $I(T; F \mid C; \theta) = 0$, meaning feedback $F$ provides no relevant information about target $T$ given context $C$.
>
>
> * **Think-with-Me reduces reasoning uncertainty via target-relevant feedback.** Proposition 1 clarifies the mechanism of the proposed interactive intervention framework: integrating **target-relevant feedback** into reasoning **reduces reasoning uncertainty** and thereby mitigates redundant exploration and verification caused by such uncertainty [r5]. ** We further explain the uncertainty reduction in  Think-with-Me as follows:
>
>      * **Feedback is generated based on criteria towards the final answer.** In our method, $F$ (external feedback) is **target-relevant**. It is generated based on $C$ towards the final answer via two content-agnostic criteria: **Rationality** and **Completeness** to adjust the sequential reasoning. This enables $F$ to be relevant for $T$, so $I(T; F \mid C; \theta) > 0$ in practice. Substituting into the equation above gives: $ H(T \mid C, F; \theta) < H(T \mid C; \theta) $ and therefore $\Delta H>0$.
>
>      * **Empirical evidence validates the effectiveness of explaining the mechanism of the method for Proposition 1.** We computed the average entropy of reasoning outcomes before and after applying Think-with-Me across all datasets. The proportion of cases with a positive entropy change ($\Delta H>0$) reached **98.73%**, which demonstrates the effectiveness of Proposition 1 to explain the mechanism of the proposed method theoretically.
>
> We update **Section 3.2.1** in the rivision. We hope our clarifications above can address your concerns and can help you re-assess our work.
>
>
> [r5] Think Clearly: Improving Reasoning via Redundant Token Pruning. ICML, 2025.

---

> ### Author Response · Authors · 2025-11-25
>
> Dear Reviewer Zmbd,
>
> We would like to thank you for your valuable comments on our paper. We sincerely hope that our response has addressed your initial concerns. If there are still unclear parts for you, please let us know. We will do our best to address them.
>
> Best regards,
>
> The Authors

---

> ### Author Response · Authors · 2025-11-28
>
> Dear Reviewer Zmbd,
>
> Thank you for your detailed feedback and valuable comments! To help you quickly grasp the core of our response, manuscript revisions, and contributions, we’ve condensed the key points below:
>
> **Core Response Highlights**
> - **Q1 Baselines & method distinctions**: We **extended comparisons with representative baselines** (single-model: s1-32B, DEER, SEAL; multi-model: SpecReason). Think-with-Me **fundamentally differs**: it uses semantic-level interactive intervention aligned with LRMs’ intrinsic reasoning behaviors and thus **avoiding multi-model overheads** and **breaking single-model’s internal-signal-only limitations**.
> - **Q2 Deployment cost evaluation**: We **added reasoning time & GPU memory requirement**，to illustrate the practicality of Think-with-Me.
> - **Q3 Differences from Early Stopping**: We clarified Think-with-Me relies on **interactive intervention (not single termination)**, added comparisons in the Related Work section, and benchmarked against DEER (an Early Stopping method) in Table A.
> - **Q4 Proposition 1 validity**: We **rigorously proved the proposition via information theory**, and experiments showed **98.73% of cases have reduced reasoning uncertainty**
>
>
> **Manuscript Revisions**
> - **Updated Related Work**: Added comparisons with strong baselines and related methods.
> - **Added Appendix A** (baseline comparison table) & **Appendix F** (deployment cost data).
> - Revised the experiment section (added cost evaluation) and **Section 3.2.1** for the function and validity of Proposition 1.
>
>
> **Core Contributions**
> - **Overcoming LRM Intrinsic Limitations**: Proposes an external intervention paradigm to address the core issue of "overthinking/overshoot" that existing methods fail to resolve.
> - **Behavior-Aligned Interactive Framework**: Designs intervention based on LRMs’ intrinsic reasoning behavior, transforming automatic reasoning into **transparent, steerable human-LLM collaboration**.
> - **Superior Performance with Practical Advantages**: Achieves better accuracy-length trade-off across datasets, supports dynamic deployment, balances intervention costs with result reliability, and expands LRMs’ application boundaries.
>
> We have refined the manuscript accordingly and we’d warmly welcome any further feedback you may have. We would be deeply grateful if you could kindly re-assess our submission. Thank you once again for your valuable time and support throughout this process.
>
> Sincerely,
>
> The Authors

---

### Official Review · Reviewer_7csP · 2025-11-01

**Soundness:** 2
**Presentation:** 3
**Contribution:** 2
**Rating:** 2
**Confidence:** 4

**Summary:**

This paper proposes Think-with-Me, a test-time interactive reasoning framework designed to improve the efficiency of Large Reasoning Models (LRMs). The method pauses the model's reasoning process at transitional conjunctions (e.g., "so", "wait") and introduces external feedback from humans or LLMs to adaptively extend or terminate reasoning. They use Group Relative Policy Optimization (GRPO) to adapt the model to this interactive framework. Experiments on several reasoning benchmarks show that their method improves the trade-off between accuracy and efficiency within an 8K-token context window.

**Strengths:**

* This paper is well-written and easy to understand.
* This paper addresses an important issue in optimizing the test-time reasoning efficiency.

**Weaknesses:**

1. My main criticism is **on the design of the experiments**:
   a. In the observation experiments of Section 3.1 in Figure 1, the authors conduct their analysis on DeepSeek-R1-Distill-Qwen-32B, showing in Figures 1(a) and 1(b) the prevalence of conjunction tokens in the reasoning traces of **o1-like models**. However, when investigating the influence of these tokens, they switch their experimental model to Qwen2.5-72B-Instruct, which, as far as I know, does **not possess long reasoning ability**. In summary, this inconsistency makes the experimental methodology appear odd and the observation seems less reliable.
   b. The main experiment shown in Table 1 also seems **unfair**, since their method uses external powerful models such as Qwen2.5-72B-Instruct, or even involves human feedback, while other baselines such as DeepSeek-R1 and QwQ-32B operate autonomously. This raises the question of whether the improvements come from their post-training process or from the additional external hints.
2. **The use of GRPO is not well-motivated**. There is no clear explanation of why reinforcement learning is necessary here, nor any ablation comparing GRPO with simpler alternatives such as SFT on synthetic intervention traces. A comprehensive comparison across different methodologies would enhance the rigor and insight of their approach.
3. The theoretical claim (Proposition 1) that feedback reduces conditional entropy seems trivial and does not explain why the designed interactive intervention helps reasoning. Moreover, there is insufficient theoretical or experimental evidence to suggest that intervening at conjunctions like "so" or "wait" leads to better performance than intervening elsewhere.
4. The human-in-the-loop setting might not be scalable for large datasets or real-time inference. The paper does not estimate human cost or latency under realistic deployment conditions.

**Questions:**

1. Regarding weakness 1.a, can the experiments in Figure 1 be carried out on the same model?
2. Regarding weakness 1.b, can the authors compare with other baselines, such as also providing external suggestions from another powerful model, or directly inserting these interactive feedbacks into a non-finetuned model, or making modifications to make those feedbacks resemble the reasoning models' own intermediate criticism of their reasoning traces?
3. Regarding weakness 2, can the authors compare the use of GRPO with other post-training techniques, such as SFT on synthetic traces?
4. Can their method be applied to longer context windows beyond 8k tokens while maintaining the advantages shown in Table 1?
5. Additionally, I wonder whether reasoning tasks are the most suitable application for the proposed interactive framework. For example, if one wants to use an LRM to solve a mathematical problem but lacks sufficient knowledge to solve it, will they have the ability to judge the correctness of intermediate reasoning results or even provide useful suggestions or feedback? Perhaps this framework is more suitable for applications like creative writing, where humans can more easily judge whether the current results align with their intentions. Please correct me if I am wrong.
6. Since LLMs or humans might not have the ability to process long contexts, would it be better to summarize the intermediate reasoning trace before sending it for judgment?

---

> ### Author Response · Authors · 2025-11-22
> **Responses to reviewer 7csP[1/5]**
>
> We sincerely appreciate the reviewer's thoughtful comments and valuable suggestions. We organized all reviewer's questions and suggestions and provided point-by-point responses:
>
> >Q1: My main criticism is of the design of the experiments:
> > Q1.a In the observation experiments of Section 3.1 in Figure 1, the authors conduct their analysis on DeepSeek-R1-Distill-Qwen-32B, showing in Figures 1(a) and 1(b) the prevalence of conjunction tokens in the reasoning traces of o1-like models. However, when investigating the influence of these tokens, they switch their experimental model to Qwen2.5-72B-Instruct, which, as far as I know, does not possess long-term reasoning ability. In summary, this inconsistency makes the experimental methodology appear odd and the observation seems less reliable.
>
> **A1.a.** Thank you for the detailed comment. We appreciate the reviewer's careful reading. The experimental setup is consistent and methodologically sound:
> * All the **reasoning traces are generated by DeepSeek-R1-Distill-Qwen-32B** and this provides a unified basis for our observation study.
> * We use Qwen2.5-72B-Instruct to evaluate whether the reason for LRM truncated by transaction conjunctions includes the correct answer with reference to the ground truth. **Qwen2.5-72B-Instruct is applied as a judge** instead of the reasoning generator.
>
> > Q1.b The main experiment shown in Table 1 also seems unfair, since their method uses external powerful models such as Qwen2.5-72B-Instruct, or even involves human feedback, while other baselines such as DeepSeek-R1 and QwQ-32B operate autonomously. This raises the question of whether the improvements come from their post-training process or from the additional external hints. Can the authors compare with other baselines, such as also providing external suggestions from another powerful model, or directly inserting these interactive feedbacks into a non-finetuned model, or making modifications to make those feedbacks resemble the reasoning models' own intermediate criticism of their reasoning traces?
>
> **A1.b.** Thank you for the detailed comment. We address your concerns through empirical evidence and methodological clarifications:
> * **Fair comparisons were conducted under controlled settings.** All methods are evaluated under the **same 8k** context. Given that **LRMs rely on test-time scaling (required for a 32k window)** to achieve high performance, we additionally report the result of LRMs with a 32k window size to further illustrate the advantages of our method in balancing accuracy-length within a constrained window.
> * **Ablation studies confirm improvements stem from the whole framework.** The required result and analysis of the ablation study have been mentioned **in Table 4 of Section 4.3** in the original paper. The ablation results demonstrate the beneficial impacts of each component on the performance of the proposed method. Specifically:
>     * The result of "Directly inserting these interactive feedbacks into a non-finetuned model"  can be found in the "w/o trained model" ablation setting
>     * The result of "Making modifications to make feedback resemble the reasoning model's own intermediate criticism of its reasoning traces" can be found in the  "w/o special tokens" ablation setting.
>     * The result of "Providing external suggestions from another powerful model" can be found in Table 6 of the paper.
>
> We will enhance the clarity of these experimental designs in the revised manuscript. We sincerely hope our clarifications above have addressed your concerns and can help you re-assess our work.

---

> ### Author Response · Authors · 2025-11-22
> **Responses to reviewer 7csP[2/5]**
>
> >Q2: The use of GRPO is not well-motivated. There is no clear explanation of why reinforcement learning is necessary here, nor any ablation comparing GRPO with simpler alternatives such as SFT on synthetic intervention traces. A comprehensive comparison across different methodologies would enhance the rigor and insight of their approach. Can the authors compare the use of GRPO with other post-training techniques, such as SFT, on synthetic traces? Regarding weakness 2, can the authors compare the use of GRPO with other post-training techniques, such as SFT on synthetic traces?
>
> **A2.** Thank you for the insightful comment. Our choice of GRPO is its ability to **directly optimize non-differentiable objectives like the accuracy-length trade-off**. The detailed response is as follows:
> * **GRPO is suited for optimizing non-differentiable, multi-objective reasoning under interactive feedback.** Compared with SFT, GRPO is suited to **optimizing non-differentiable, multi-objective reasoning** under interactive feedback[R1]. GRPO uses policy gradients to optimize a composite reward, training the model to respond to feedback rather than mimic traces[R2,R3]. By contrast, SFT performs MLE(maximum likelihood estimation) **imitation of synthetic traces**, cannot directly target the accuracy–length trade-off, and often memorizes synthetic traces, yielding **weaker generalization** to real feedback[R4].
> * **Empirical evidence of GRPO is better than SFT under our setting.** We create synthetic trajectories based on our interactive paradigm where the target reasoner was the untrained **DeepSeek-R1-Distil-Qwen2.5-32B** and trained **DeepSeek-R1-Distil-Qwen2.5-7B** by LoRA using the Llamafactory framework. From Table A,  our **GRPO-based** result **consistently achieves stronger accuracy–length trade-offs** across reasoning datasets compared to an SFT-trained variant.
>
> We will include the detailed comparison of the SFT training result in **4.3 ABLATION STUDY** in the revised manuscript. We sincerely hope our clarifications above have addressed your concerns and can help you re-assess our work.
>
> Table A. The result of applying a different training strategy.
> |   Method             | MATH500       |              | AIME24        |              | GRPO          |              | LiveCode      |              |
> |----------------|---------------|--------------|---------------|--------------|---------------|--------------|---------------|--------------|
> |                | accuracy      | length       | accuracy      | length       | accuracy      | length       | accuracy      | length       |
> | Base model     |       84.05        |     1864.33         |      40.00         |     3374.50         |      22.25        |      2757.11        |                22.29        |6305.82|
> | ours(SFT)      |        86.00       |     1404.15         |    63.33           |       3617.78      |        64.20       | 1515.43            |        28.65       |    1235.57          |
> | ours(GRPO)     |       90.60        |     1081.87         |      73.85         |       1182.50       |65.22| 1136.38      |     34.86          |      1248.64     |
>
> [r1]DeepSeekMath: Pushing the Limits of Mathematical Reasoning in Open Language Models. Arxiv, 2024.
>
> [r2] L1: Controlling How Long A Reasoning Model Thinks With Reinforcement Learning. Arxiv, 2025.
>
> [r3] Scalable Chain of Thoughts via Elastic Reasoning. Arxiv, 2025.
>
> [r4] SFT Memorizes, RL Generalizes: A Comparative Study of Foundation Model Post-training. ICML, 2025.
>
> >Q3： The theoretical claim (Proposition 1) that feedback reduces conditional entropy seems trivial and does not explain why the designed interactive intervention helps reasoning.
>
> **A3.** Proposition 1 theoretically explains the mechanism of the proposed interactive intervention: external feedback reduces reasoning uncertainty and thereby mitigates redundant exploration and validation driven by uncertainty:
> * **The theoretical explanation of Thin-with-Me is grounded in information theory.** As a direct consequence of information theory, Proposition 1 confirms that additional **target-relevant information reduces total reasoning uncertainty**. Proposition 1 provides a theoretical explanation for the mechanism of Think-with-Me, which reduces reasoning uncertainty by introducing target-relevant feedback during the reasoning process.
> * **Benefit of uncertainty reduction for reasoning efficiency.**  The reduction in uncertainty **curtails the redundant self-validation and exploration** that arise from uncertainty [r5], improving reasoning efficiency to converge more efficiently to obtain conclusions.
>
> We have enhanced the corresponding **discussion in subsection 3.2.1** of the revision to clarify this theoretical-methodological connection further. We sincerely hope our clarifications above have addressed your concerns and can help you re-assess our work.
>
> [r5]Think Clearly: Improving Reasoning via Redundant Token Pruning. EMNLP, 2025.

---

> ### Author Response · Authors · 2025-11-22
> **Responses to reviewer 7csP[3/5]**
>
> >Q4： Moreover, there is insufficient theoretical or experimental evidence to suggest that intervening at conjunctions like "so" or "wait" leads to better performance than intervening elsewhere.
>
> **A4.** Thank you for the detailed comment. We provide **experimental evidence** to validate that intervening at conjunctions like "so" or "wait" leads to better performance：
>
> * **Compared ablation settings.** We compared other intervention points under the same 8K context window: (1) intervening after every sentence(after "."), (2) intervening every 256 tokens, and (3) intervening at the "\n\n" token.
>
> * **Our conjunction-based intervention outperforms alternative schemes across datasets.** From Table B, intervening at transitional conjunctions outperforms other strategies in balancing accuracy and length. Based on Observation 1, transitional conjunctions serve as natural stage markers in the reasoning process, where interventions at these points **provide accurate feedback after a complete reasoning segment**. In contrast, the compared strategies often intervene during an ongoing reasoning stage, which **leads to feedback being generated based on incomplete reasoning segments**, resulting in less accurate guidance and consequently worse performance.
>
>
> We have added detailed comparisons and analyses of the intervention point in **Appendix G.2** of the revision to validate our design choice using experimental evidence.
>
> Table B.The result of intervention through different strategies.
> | Method                                      | math500       |          | AIME24        |          | gpqa          |          | livecode      |          |
> |---------------------------------------------|---------------|----------|---------------|----------|---------------|----------|---------------|----------|
> |                                             | accuracy      | length   | accuracy      | length   | accuracy      | length   | accuracy      | length   |
> | Base model                                  | 84.05         | 1864.33  | 40.00         | 3374.50  | 22.25        | 5020.86   | 22.29         | 6305.82  |
> | Intervene after every sentence              | 82.4          | 2202.42  | 51.11         | 2963.37  | 52.67         | 3902.93  | 15.43         | 1911.11  |
> | Intervene every 256 tokens                  | 80.00         | 2280.10  | 38.33         | 2303.52  | 56.70         | 2360.85  | 25.14         | 1504.77  |
> | Intervene at "\n\n"                           | 86.60         | 1541.06  | 56.35         | 3943.43  | 62.67         | 1848.23  | 22.14         | 1657.08  |
> | Intervene at transitional conjunctions (ours) | **90.60**       |  **1081.87**  |  **73.85**         |  **1182.50**  |  **65.22**         |  **1136.38**  | **34.86**         | **1248.64**  |
>
>
> >Q5: The human-in-the-loop setting might not be scalable for large datasets or real-time inference. The paper does not estimate human cost or latency under realistic deployment conditions.
>
> **A5.** Thank you for the detailed comment. Our framework is designed to support **multi-source feedback for different scenarios** to ensure applicability:
>
> * **Think-with-Me utilizes LLM proxies for batch feedback while reserving human feedback for high-stakes or creative scenarios.** While an LLM proxy can be generated automatically, it can also be further specialized to have a more specific feedback ability. **Human feedback** is **invaluable** in **high-stakes** or **creative scenarios** where nuanced judgment is critical[r6].
>
>
> * **Think-with-Me is practical in the human-in-the-loop setting**. From Table C, considering the verification overhead of the methods or model switching costs, efficient reasoning methods generally require additional time costs. Think-with-Me exhibits a **small time increment (or even shorter time)** compared to the base model, demonstrating its practicality.
>
> We update the detailed time cost of methods in **Appendix E.2**. Besides, we have reported the details of the time cost for Think-with-Me in Table 3 of the original paper. We sincerely hope our clarifications above have addressed your concerns and can help you re-assess our work.
>
>
>
> Table C. Average total reasoning time per sample across datasets for methods. **Detailed data are in the revised version of the paper**.
> | Method             | Total_Time(s) |
> |--------------------|---------------|
> | Base model         | 49.76         |
> | L1-Max             | 89.15         |
> | SEAL[r1]            | 162.63       |
> | DEER [r2]           | 45.33        |
> | Speculative Thinking | 208.14       |
> | SpecReason[r3] | 166.25       |
> | Thinke-with-Me(LLM proxy)          | 41.12         |
> | Thinke-with-Me（human-in-the-loop）  | 51.35         |
>
> [r6] Best humans still outperform artificial intelligence in a creative divergent thinking task. Nature, 2024.

---

> ### Author Response · Authors · 2025-11-22
> **Responses to reviewer 7csP[4/5]**
>
> >Q6: Can their method be applied to longer context windows beyond 8k tokens while maintaining the advantages shown in Table 1?
>
> **A6.** Thank you for your insightful comment. Our method **supports up to 32k** context windows while **retaining advantages.** We call the API service of a 128k window size Qwen2.5-72B-Instruct as the LLM proxy and run the target reasoning model(7B) locally with a 32k window size on a 1*NVIDIA A800. From Table D, our method **maintains the advantage on the**accuracy–length trade-off** across challenging datasets when scaled to a 32k window size.
>
> We update the additional comparison in **Appendix H.3**. We sincerely hope our clarifications above have addressed your concerns and can help you re-assess our work.
>
> Table D. The result of methods under a 32k window size.
> | Method          | MATH500 |         | AIME24 |         | GPQA  |         | Livecode |          |
> | --------------- | ------- | ------- | ------ | ------- | ----- | ------- | -------- | -------- |
> |                 | accuracy    | length    | accuracy    | length   | accuracy    | length    | accuracy    | length     |
> | DeepSeek-RI-7B  | 87.60   | 3463.52 | 50.00  | 5608.00 | 27.45 | 17040.71 | 22.00    | 13332.02 |
> | DeepSeek-RI-14B | 90.70   | 3168.17 | 70.00  | 5948.55 | 47.64 | 9678.51 | 46.28    | 9414.02  |
> | DeepSeek-RI-32B | 91.75   | 3309.54 | 70.97  | 5758.77 | 53.80 | 11733.01 | 53.14    | 10843.10 |
> | QwQ-32B         | 92.51   | 3265.72 | 73.33  | 8794.43 | 53.13 | 11732.11 | 54.86    | 21648.70  |
> | Think-with-Me(Ours) |    **94.2**    |   **1371.23**      |     **75.94**   |  **1779.24**       |  **67.78**     |**1496.38**        |     **37.30**     |   **1671.02**|
>
>
> >Q7: Additionally, I wonder whether reasoning tasks are the most suitable application for the proposed interactive framework. For example, if one wants to use an LRM to solve a mathematical problem but lacks sufficient knowledge to solve it, will they have the ability to judge the correctness of intermediate reasoning results or even provide useful suggestions or feedback? Perhaps this framework is more suitable for applications like creative writing, where humans can more easily judge whether the current results align with their intentions. Please correct me if I am wrong.
>
> **A7.** Thank you for the detailed comment. Think-with-Me is designed to mitigate overthinking and overshooting in large reasoning models (LRMs) via an interactive intervention framework, whose simplicity and low expertise threshold enable broad applicability across reasoning, creative, and safety-critical tasks：
>
> * **Think-with-Me is proposed to address the overthink and overshoot problem of LRMs.** Through a targeted interactive intervention mechanism rooted in reasoning behavior analysis of LRMs, Think-with-Me adaptively adjusts the depth and direction of reasoning while preserving logical coherence, reducing redundant reasoning and therefore improving reasoning efficiency.
>
>
> * **The content-agnostic feedback criteria minimize the demand for user expertise.** Two agnostic criteria: Rationality (logical validity) and Completeness (answer readiness) are proposed to enable non-experts to provide effective feedback without deep domain knowledge but common sense.
>
>
> * **Experimental evidence confirms non-expert users can successfully guide the model.** Non-expert participants in a human setting of Think-with-Me(details **in Appendix E.2**) without professional mathematical backgrounds provide feedback that led the model to outperform the base model i accuracy and length balance (**Table 2 in paper**), demonstrating the low usability thresholds.
>
> * **The simplicity of the intervention mechanism supports the extension to more creative and safety-critical tasks.** Rooted in reasoning behavior analysis of LRMs, Think-with-Me adaptively adjusts the depth and direction of reasoning in-process through multi-source feedback, increasing the controllability of the reasoning process and thus broadening LRMs’ application boundaries and practical value.

---

> ### Author Response · Authors · 2025-11-22
> **Responses to reviewer 7csP[5/5]**
>
> >Q8: Since LLMs or humans might not have the ability to process long contexts, would it be better to summarize the intermediate reasoning trace before sending it for judgment?
>
> **A8.** Thank you for your insightful comment.  **Audit-friendly interaction** is one of the **essential enhancement** of Think-with-Me. **Summarizing** intermediate reasoning traces before judgment is **effective**, addressing long-context processing limitations while enhancing our method’s value:
>
> * **Experiment Setting.** To reduce redundant summarization, we **incrementally** summarized reasoning for evaluation, which means **only the reasoning generated in the current interaction round is summarized each time**. Feedback is generated based on accumulated, summarized reasoning.
>
>
> * **Summary before feedback can improve the advantages of Think-with-Me.** From Table 6, the summarization strategy has:
>     * Performance gain: Summarization further improves accuracy with comparable reasoning length reduction across datasets.
>     * Practicality: Additional time cost is acceptable (second-level).
>
>
> We update these discussions along with potential mitigation strategies, into **Appendix J.1 (Limitations)** and **Appendix J.2 (Future Work)** of the revised manuscript.
>
>
> Table E. The result of method before and after applying summary before making feedback. Totoal reasoning time include the time for reasoning, making feedback and summarizing.
> | Method                  | MATH500                  |                  |                                      | AIME24                   |                  |                                      | GPQA                     |                  |                                      | LIVECODE                 |                  |                                      |
> |-------------------------|--------------------------|------------------|--------------------------------------|--------------------------|------------------|--------------------------------------|--------------------------|------------------|--------------------------------------|--------------------------|------------------|--------------------------------------|
> |                         | accuracy                      | length              | total_time  | accuracy                      | length             | total_time | accuracy                      | length             | total_time  |accuracy                      | length       | total_time  |
> | base                    | 84.05                    | 1864.33          | 35.48                                | 40.00                    | 3374.50          | 49.54                                | 22.25                    | 5020.86          | 79.48                                | 22.29                    | 6305.82          | 72.51                                |
> | ours(LLM proxy)         | 90.60                    | 1081.87          | 36.38                                | 73.85                    | 1182.50          | 62.27                                | 65.22                    | 1136.38          | 36.39                                | 34.86                    | 1248.64          | 39.92                                |
> | ours(LLM proxy with summary)      | 92.60                    | 820.18           | 34.94                                | 75.73                    | 1530.37          | 75.29                                | 76.67                    | 926.27           | 50.86                                | 36.00                    | 1479.80          | 45.41                                |
> | ours(human-in-the-loop) | 90.00                    | 1009.40          | 32.21                                | 61.62                    | 1597.00          | 55.48                                |            57.86         | 1065.53          | 35.43                                | 34.44                    | 1406.80          | 59.85                                |
> | ours(human-in-the-loop with summary) | 91.37                | 1013.5           | 55.88                                | 73.33                    | 1504.65          | 57.64                                | 68.20                   | 1152.65          | 75.83                                | 39.43                    | 1118.5           | 65.19                                |

---

> ### Author Response · Authors · 2025-11-25
>
> Dear Reviewer 7csP,
>
> We have tried our best to address all the concerns and provided explanations to all questions. We sincerely hope that our answer has addressed your initial concerns. Kindly let us know if you have any other concerns, and we will do our best to address them.
>
> Best regards,
>
> The Authors

---

> ### Author Response · Authors · 2025-11-28
>
> Dear Reviewer 7cSP,
>
> Thank you for your insightful comments and valuable suggestions. We have addressed each of your concerns in detail and summarize the **core responses**, **manuscript revisions**, and **key contributions** below for your quick reference. We look forward to your further discussion.
>
>
> **Core Response Highlights**
> - **Experimental Design Consistency**: All experiments in Figure 1 use the same model (DeepSeek-R1-Distill-Qwen2-32B) for fair comparison. **All methods are evaluated under the same context window** in main experiments, with additional control experiments.
> - **Rationale for GRPO**: It is well-suited for non-differentiable objectives like "accuracy-length trade-off," and experiments demonstrate its superiority over SFT and other methods.
> - **Intervention Strategy Validity**: Experiments confirm **intervention at conjunctions outperforms sentence-end/fixed-token intervention**.
> - **Practicality & Scalability**: Think-with-Me supports multi-source feedback (LLM agents/humans) for different senarios and **scales to 32k** context windows. **Time overhead data is supplemented** to verify efficiency.
> - **Applicability**: With **low useage-threshold** feedback criteria, the method is proven effective for feedback with various quality. Long-context scenario is proven to be **optimized via incremental summarization**.
> - **Function of Proposition 1** Proposition 1 serves as the foundational theoretical explanation for Think-with-Me’s interactive intervention mechanism.
>
>
> **Manuscript Revision**
> The revised version will add the following to the appendix:
> - Experimental design details (**Section G.2**) and comparisons of different training methods (**Section A.3**)
> - Time overhead data (**Section E.2**) and 32k context window experimental results (**Section H.3**)
> - The potential limitation under long-context  and can be optimized by summarization strategies (**Sections J.1/J.2**)
> - The detailed function of Proposition 1(**Sections 3.2.1**)
>
>
> **Key Contributions**
> - **Overcoming LRM Intrinsic Limitations**: Proposes an external intervention paradigm to address the core issue of "overthinking/overshoot" that existing methods fail to resolve.
> - **Behavior-Aligned Interactive Framework**: Designs intervention based on LRMs’ intrinsic reasoning behavior, transforming automatic reasoning into **transparent, steerable human-LLM collaboration**.
> -  **Superior Performance with Practical Advantages**: Achieves better accuracy-length trade-off across datasets, supports dynamic deployment, balances intervention costs with result reliability, and expands LRMs’ application boundaries.
>
>
> We have refined the manuscript accordingly and we’d warmly welcome any further feedback you may have. We would be deeply grateful if you could kindly re-assess our submission. Thank you once again for your valuable time and support throughout this process.
>
> Sincerely,
>
> The Authors

---

### Official Review · Reviewer_NoWM · 2025-11-01

**Soundness:** 2
**Presentation:** 3
**Contribution:** 2
**Rating:** 4
**Confidence:** 4

**Summary:**

This paper addresses the LLM long2short problem by introducing external feedback. Specifically, at logical connectives, a LLM or human annotator provides prompts indicating whether the current reasoning process is complete (i.e., has derived the correct answer) and sound (i.e., whether the reasoning path needs correction). This feedback guides the model to adaptively perform test-time scaling. In comparisons within an 8K context window, this approach demonstrates a notable reduction in the length of output tokens.

**Strengths:**

1. The paper is well-written, clear, and easy to follow.
2. The proposed method achieves a significant improvement in compressing reasoning length.
3. It provides both LLM-based and human-in-the-loop feedback mechanisms, enhancing the scalability of the approach.

**Weaknesses:**

1. Several existing works, for example [1] have already explored, from various perspectives, the use of external feedback frameworks—such as human feedback, model feedback, or verifiers—to improve LLM training, and these approaches can be adapted to address the long2short problem. Therefore, the empirical motivation of this paper needs to be further strengthened.
2. The paper's two key observations are not new: the first observation has been similarly articulated in numerous works since [2], and the second observation has also been mentioned in some recent works, such as [3].
3. The paper lacks sufficient comparison with relevant baselines. Since[2], numerous long2short approaches have emerged—too many to list exhaustively here (see recent surveys for a comprehensive overview). Whether inference-time or training-based, the methods cited in the paper are not strongly representative of the state of the art. The authors should conduct careful comparisons against closely related approaches to properly demonstrate the advantages of their proposed method.


[1] Self-Refine: Iterative Refinement with Self-Feedback
[2] s1: simple test-time scaling
[3] Efficient Reasoning Through Suppression of Self-Affirmation Reflections in Large Reasoning Models

**Questions:**

Please see the weaknesses.

---

> ### Author Response · Authors · 2025-11-22
> **Responses to reviewer NoWM [1/2]**
>
> We sincerely appreciate the reviewer's thoughtful comments and valuable suggestions. We organized all reviewer's questions and suggestions and provided point-by-point responses:
>
> >Q1: Several existing works, for example [r1], have already explored, from various perspectives, the use of external feedback frameworks—such as human feedback, model feedback, or verifiers—to improve LLM training, and these approaches can be adapted to address the long2short problem. Therefore, the empirical motivation of this paper needs to be further strengthened.
>
> **A1.** While external feedback is widely studied， Think-with-Me is an **interactive intervention mechanism** based on LRM's intrinsic reasoning behavior and external feedback is a tool, applied to intervene in the reasoning process:
>
> * **Think-with-Me is an interactive intervention mechanism for the reasoning process of LRMs.** It tackles two failure modes: overthinking (redundant extension) and overshoot (corrupting correct answers through unbounded reasoning). Think-with-Me uses **external feedback during the reasoning process to steer depth and direction** while preserving logical coherence, resulting in shorter yet more accurate reasoning and improving overall reasoning efficiency.
>
> * **Existing feedback methods misalign with LRMs' reasoning behavior**, and fail to optimize the accuracy–length trade-off for LRMs. Most methods apply feedback **post-generation**[r2] or **at fixed answer stages**[r3], which mismatch LRMs in terms of timing, content, and form.
>
> * **Empirical evidence suggests that the existing method does not improve the reasoning efficiency of LRMs.** From Table A, directly applying the representative feedback-based method[r1] **fails to reduce reasoning length while maintaining accuracy**, indicating the unsuitability of addressing the Long2short problem of LRMs.
>
> We enhance the **1. INTRODUCTION**, **3.1 OBSERVATION** and **Appendix B.1** to reflect the rebuttal in our paper. We sincerely hope that the clarifications above have addressed your concerns and can help you reassess our work.
>
> Table A. Applying Self-Refine on DeepSeek-R1-Distilled-Qwen2.5-7B with feedback from itself or Qwen2.5-72B-Instruct.
> | Method      | math500 |       | gpqa |       | LiveCode |       |
> |-------------|---------|-------|------|-------|----------|-------|
> |             | accuracy    | length   | accuracy    | length   | accuracy    | length  |
> | base model     |    84.05     |   1864.33    |  22.25   |   5020.86    | 22.29         |   6305.82    |
> | Self-Refine[r1]   |   89.90  |  2335.98     |   28.49   |   8476.44    |    19.19      |   8523.69    |
> | Self-Refine (Qwen2.5-72B-Instruct) |    87.00     |   2226.67    |   25.00   |   6335.25    |    29.63      |   7343.26    |
> | Ours（Qwen2.5-72B-Instruct） |    90.6| 1081.87      |   65.22        |   1136.38| 34.86| 1248.64|
>
> [r1] Self-Refine: Iterative Refinement with Self-Feedback. Nips，2023.
>
> [r2] Reflexion: language agents with verbal reinforcement learning. Nips,2023.
>
> [r3] Improve Mathematical Reasoning in Language Models by Automated Process Supervision, Arxiv. 2024.
>
> >Q2：The paper's two key observations are not new: the first observation has been similarly articulated in numerous works since [r4], and the second observation has also been mentioned in some recent works, such as [r5].
>
> **A2.** While prior works mention isolated phenomena, our contribution is integrating them into a quantitative, actionable evidence chain for LRMs with new insight:
> * **The distinction of Observation 1 and 2.** Unlike [r4] (frequency-only) and [r5] (self-affirmation focus), we disentangle Obs . 1/Obs.2, further analyzing transitional conjunctions and their co-occurrence with self-validation and exploration patterns, and providing quantitative evidence on their functional impact on reasoning accuracy.
>
>
> * **We offer a new, finer-grained analysis of LRM reasoning through observation.** We provide insight that **transitional conjunctions act as natural stage markers with a bidirectional impact on accuracy**, supported by quantitative statistics, which have not been mentioned explicitly in previous work like [r4] and [r5].
>
> * **These observations support a concrete and novel test-time intervention mechanism for LRMs.** Think-with-Me operationalizes these insights by **pausing at conjunctions** in LRMs' reasoning and applying dual-criteria external feedback to **adjust the depth and direction of reasoning** while preserving logical coherence.
>
>
> We enhance the **3.1 OBSERVATIONS** by more precisely citing and differentiating prior work, ensuring our contributions are clearly highlighted.
>
> [r4]s1: simple test-time scaling. EMNLP, 2025.
>
> [r5]Efficient Reasoning Through Suppression of Self-Affirmation Reflections in Large Reasoning.Arxiv, 2025.

---

> ### Author Response · Authors · 2025-11-22
> **Responses to reviewer NoWM [2/2]**
>
> >Q3: The paper lacks sufficient comparison with relevant baselines. Since[r2], numerous long2short approaches have emerged—too many to list exhaustively here (see recent surveys for a comprehensive overview). Whether inference-time or training-based, the methods cited in the paper are not strongly representative of the state of the art. The authors should conduct careful comparisons against closely related approaches to demonstrate the advantages of their proposed method properly.
>
> **A3.** Thank you for your insightful suggestion. We include strong baselines and demonstrate that Think-with-Me achieves state-of-the-art accuracy–length trade-offs.
>
> * **The comparison result of expanding baselines.** We add strong and representive long2short baselines across both paradigms—single-model optimization: s1-32b[r4], SEAL[r6], DEER[r7] and multi-model collaboration: Specreason[r8]. From Table B, Think-with-Me **consistently attains competitive or superior accuracy while significantly reducing reasoning length** across datasets, demonstrating its efficiency.
>
> * **Think-with-Me is an interactive intervention mechanism based on LRM's intrinsic reasoning behavior.** Existing efficient reasoning methods fall into single-model optimization and multi-model collaboration:
>     * **Single-model optimization** methods rely on post-training or test-time controls from internal numerical signals, lacking fine-grained semantic steering
>     * **multi-model collaboration** methods can boost accuracy but add coordination or computational complexity.
>
>     Based on LRMs' inherent reasoning behaviors， Think-with-Me is an **interactive intervention mechanism** that applies external **semantic feedback** to **adapt reasoning depth and direction** during the reasoning process, achieving efficient reasoning **within a single model**.
>
> We update the **Related Work** in Appendix A with a systematic discussion and comparison of recent efficient reasoning approaches and other relevant parts, including **1. INTRODUCTION** and **4. EXPERIMENT** to reflect the rebuttal in our paper. We sincerely hope our clarifications above have addressed your concerns.
>
>
> Table B. The result of the comparison. Best and second-best results are in **bold** and \_underlined\_, respectively.
> |                | MATH500       |              | AIME24        |              | GPQA          |              | LIVECODE      |              |
> |----------------|---------------|--------------|---------------|--------------|---------------|--------------|---------------|--------------|
> |                | acc           | avg_len      | acc           | avg_len      | acc           | avg_len      | acc           | avg_len      |
> | DeepSeek-RI-7B | 84.05         | 1864.33      | 40.00         | 3374.50      | 22.25         | 5020.86      | 22.29         | 6305.82      |
> | s1-32b[r4]      | 90.00         | 2922.14      | \_55.00\_         | 6047.17      | **68.00**     | 3888.79      | 30.00         | 4019.5       |
> | SEAL[r6]           | 84.00         | \_1096.87\_      | 51.35         | \_2203.53\_     | 57.33         | 3338.8       | \_31.30\_  | 5987.83      |
> | DEER[r7]           | 87.00         | 1280.75      | 54.06         | 4127.76      | 63.33         | \_3136.17\_      | 30.80         | \_3487.17\_      |
> | specreason[r8]     | **92.00**         | 2380.78      | 50.00         | 6278.55      | 50.51         | 4450.72      | 30.00         | 6337.84      |
> | Think-with-Me(Ours)           | \_90.60\_  | **1081.87**      | **73.85**         | **1182.50**      | \_65.22\_  | **1136.38**      | **34.86**         | **1248.64**|
>
>
> [r6] SEAL: Steerable Reasoning Calibration of Large Language Models for Free. COLM, 2025.
>
> [r7] Dynamic Early Exit in Reasoning Models. Arxiv, 2025.
>
> [r8] Specreason: Fast and accurate inference-time compute via speculative reasoning. Nips, 2025.

---

> ### Author Response · Authors · 2025-11-25
>
> Dear Reviewer NoWM,
>
> We have tried our best to address all the concerns and provide explanations for all the questions. We sincerely hope that our answer can address your initial concerns. Kindly let us know if you have any other concerns, and we will do our best to address them.
>
> Best regards,
>
> The Authors

---

> ### Author Response · Authors · 2025-11-28
>
> Dear Reviewer NoWM,
>
> Thank you again for your insightful and detailed comments! To help you quickly grasp our response, we summarize the core points, manuscript revisions, and key contributions below:
>
>
> **Key Responses to Your Concerns**
> - **Empirical motivation (Q1):** Existing external feedback methods are **mismatched with LRMs’ reasoning behavior** (timing, form, content). Our Think-with-Me is an interactive intervention aligned with LRMs’ intrinsic reasoning behavior (balancing length-accuracy).
> - **Novelty of observations (Q2):** We integrated isolated prior observations into a **quantitative evidence chain** (fine-grained logical connective analysis) and derived **new insights** distinct from prior work.
> - **Baseline comparisons (Q3):** We **extended comparisons with representative baselines** (single-model: s1-32B, DEER, SEAL; multi-model: SpecReason). Think-with-Me **fundamentally differs**: it uses semantic-level interactive intervention aligned with LRMs’ intrinsic reasoning behaviors and thus **avoiding multi-model overheads** and **breaking single-model’s internal-signal-only limitations**.
>
>
>
> **Manuscript Revisions**
> - Updated **Introduction**, **Section 3.1 (Observations)**, **Related Work**, and **Appendix** (refined analysis, prior work distinctions, supplementary tables).
> - Enhanced the **Experiment** section with comparisons to representative baselines.
>
>
>
> **Key Contributions**
> - **Overcoming LRM Intrinsic Limitations**: Proposes an external intervention paradigm to address the core issue of "overthinking/overshoot" that existing methods fail to resolve.
> - **Behavior-Aligned Interactive Framework**: Designs intervention based on LRMs’ intrinsic reasoning behavior, transforming automatic reasoning into **transparent, steerable human-LLM collaboration**.
> -  **Superior Performance with Practical Advantages**: Achieves better accuracy-length trade-off across datasets, supports dynamic deployment, balances intervention costs with result reliability, and expands LRMs’ application boundaries.
>
> We have refined the manuscript accordingly and we’d warmly welcome any further feedback you may have. We would be deeply grateful if you could kindly re-assess our submission. Thank you once again for your valuable time and support throughout this process.
>
> Sincerely,
>
> The Authors

---

### Official Review · Reviewer_HGp3 · 2025-11-03

**Soundness:** 3
**Presentation:** 3
**Contribution:** 3
**Rating:** 6
**Confidence:** 3

**Summary:**

The paper introduces an innovative reasoning-enhanced framework, "Think-with-Me," designed to address common issues in large reasoning models (LRMs), such as "overthinking" and "overshoot" during the reasoning process. The framework incorporates an external feedback intervention mechanism that identifies key transitional words (such as "so") as intervention points during reasoning. This allows the reasoning process to pause, enabling human or LLM agents to provide feedback, dynamically adjusting the depth and direction of reasoning, reducing redundant reasoning steps while maintaining or improving accuracy. To ensure the model can effectively respond to external feedback, the authors employed the Group Relative Policy Optimization (GRPO) method to fine-tune the model, enhancing its adaptability in interactive reasoning modes. Experimental results show that Think-with-Me performs excellently across various tasks.

**Strengths:**

1、The study fully adheres to a rigorous logical chain of scientific exploration. The team first systematically analyzed the model’s reasoning behavior, identified that "transitional words" can serve as intervention nodes, and validated their effectiveness. This rigorous preliminary exploration laid a theoretical foundation for the design of Think-with-Me, forming a scientific closed loop.
2、The experimental validation is comprehensive, covering a wide range of tasks and comparing with various mainstream baseline models and efficient reasoning methods.
3、The paper provides extremely detailed appendices, including complete training details, reward function design, case studies, and even schematic diagrams of the interactive interface. This openness not only greatly enhances the transparency and credibility of the work, but also provides a valuable foundation for subsequent research and reproduction.

**Weaknesses:**

1、lthough the authors provide additional experimental details in the appendix, they do not release the source code, resulting in low reproducibility.
2、his approach relies on an external feedback mechanism, which may introduce new risks. If the LLM proxy generates incorrect feedback and the target model lacks a built-in correction mechanism, performance degradation or error propagation could occur; the authors offer no further discussion on this issue.

**Questions:**

1、In the Think-with-Me framework, feedback comes from humans or LLM proxies. If human subjectivity is too strong or the LLM's performance is poor, will the resulting erroneous feedback affect subsequent reasoning? If this problem exists, it should be pointed out in I.1

---

> ### Author Response · Authors · 2025-11-22
> **Responses to reviewer HGp3**
>
> We sincerely appreciate the reviewer's thoughtful comments and valuable suggestions. We organize all reviewer's questions and suggestions and provided point-by-point responses:
>
> >Q1: Although the authors provide additional experimental details in the appendix, they do not release the source code, resulting in low reproducibility.
>
> **A1.** We appreciate your valuable concern regarding reproducibility:
> * We have released an anonymous repository containing the main source code for review at: [https://anonymous.4open.science/r/Think-with-me-05FD].
> * All key implementation details are provided in Appendix B-G to support preliminary reproduction.
> * We will publicly release the complete source code upon acceptance.
>
>
> >Q2: This approach relies on an external feedback mechanism, which may introduce new risks. If the LLM proxy generates incorrect feedback and the target model lacks a built-in correction mechanism, performance degradation or error propagation could occur; the authors offer no further discussion on this issue.
>
> **A2.** Thank you for your great insight. Our response is grounded in methodological design and empirical evidence:
>
> * **Methodological robustness.** The target model is fine-tuned via GRPO with a **composite reward that includes task accuracy**. This enables the model to distinguish between misleading feedback during training and improves its final accuracy in the presence of noisy signals.
>
> * **Empirical validation.** As shown in Table A (excerpted from the original manuscript's Table 6), Think-with-Me maintains a **superior** accuracy-length **trade-off** across datasets **with LLM proxies of varying quality** over baseline, demonstrating its robustness.
>
> We update the discussion in Appendices J.1 & J.2 to reflect the rebuttal in our paper. We sincerely hope the clarifications above have addressed your concerns.
>
> Table A. Results of Think-with-Me with different feedback quality( **excerpted from the original manuscript** ). The base model refers to DeepSeek-R1-Distilled-Qwen2.5-7B.
> |  Method   | LLM proxy             |   AIME24 |         | LiveCode |         | MATH500  |         |
> | --- | --------------------- | --------:| -------:| --------:| ------- | -------- | -------:|
> |    |                       | accuracy |  length | accuracy | length  | accuracy |  length |
> |  base model    |    -       |    40.00 | 3374.50 |    22.29 | 6305.82 | 84.05    | 1864.33 |
> |  Think-with-Me（Ours）   | Qwen2.5-32b-Instruct) |    63.33 | 1241.48 |    29.14 | 1344.27 | 87.20    | 1177.60 |
> | Think-with-Me（Ours）    | Qwen2.5-72b-Instruct  |    73.85 | 1182.50 |    34.86 | 1248.64 | 90.60    | 1081.87 |
> | Think-with-Me（Ours）  | kimi-k2-instruct      |    76.56 | 1193.67 |    40.57 | 1277.33 | 92.67    |  762.60 |
>
>
>
> >Q3: In the Think-with-Me framework, feedback comes from humans or LLM proxies. If human subjectivity is too strong or the LLM's performance is poor, will the resulting erroneous feedback affect subsequent reasoning? If this problem exists, it should be pointed out in Appendix J.
>
> **A3.** Thank you for your insightful suggestions. While human subjectivity and weak LLM proxies introduce erroneous feedback, Think-with-Me remains empirically robust by generating feedback based on content-agnostic criteria:
>
> * **Content-agnostic feedback criteria reduce the negative impact of biases or poor performance.** Feedback is generated based on two universal criteria (Rationality and Completeness) rather than task-specific knowledge. This minimizes subjective biases from humans or LLMs, which only need to judge logical validity and answer readiness, rather than relying on domain expertise.
>
> * **Robustness to feedback quality** As shown in Table A, Think-with-Me outperforms the base model across all feedback quality levels， demonstrating its robustness.
>
> * **Possible mitigations** To further reduce the error feedback, feedback voting[r1], dynaminc weighting feedback[r2] and compression reasoning[r3] are potential enhance stratefies:
>
>     * **feedback voting**, which includes human or multiple LLMs to make feedback and vote on the most general feedback as the final feedback to avoid individual subjective errors or poor performance.
>     * **dynamic weighting feedback**, which includes human or multiple LLMs that generate feedback with different weights, and the final weight can be the composite one.
>     * **compression reasoning**, which reduces the burden to produce long reasoning content and then reduces the error risk for making feedback.
>
> We updated the discussion in Appendix J.1 and J.2 in the revision. We sincerely hope our clarifications above have addressed your concerns.
>
>
> [r1] LLM Voting: Human Choices and AI Collective Decision Making, 2024.
>
> [r2] DynaThink: Fast or Slow? A Dynamic Decision-Making Framework for Large Language Models, 2024.
>
> [r3] Reasoning Path Compression: Compressing Generation Trajectories for Efficient LLM Reasoning, 2025.

---

> ### Author Response · Authors · 2025-11-25
>
> Dear Reviewer HGp3,
>
> We would like to thank you for your valuable comments on our paper. We sincerely hope that our response can address your initial concerns. If there are still unclear parts for you, please let us know and we look forward to having further discussions with you.
>
> Best regards,
>
> The Authors

---

> ### Author Response · Authors · 2025-11-28
>
> Dear Reviewer HGp3,
>
> Thank you again for your insightful and constructive comments! To help you quickly grasp our responses, revisions, and core contributions, we summarize the key points below:
>
>
> **Key Responses to Concerns**
> - **Reproducibility (Q1):** We’ve shared an **anonymous repository (containing core code)** for preliminary reproducibility; full code will be released upon acceptance.
> - **Erroneous Feedback Impact (Q2):** Our **GRPO-based design** ensures robustness. Existing experiments verify **superior accuracy-length trade-offs** even with various-quality LLM proxies.
> - **External Feedback Risks (Q3):** We use **content-agnostic criteria** to reduce bias, and supplemented **voting/dynamic weighting mitigation strategies**.
>
>
> **Manuscript Revision**
> - Add discussions on potential feedback risks and potential enhance strategies in the **Appendix J.1 and J.2**
>
>
> **Core Contributions**
> - Identifies **LRMs’ intrinsic limitations** and **pervasively existing overthinking and overshoot issues**;
> - Builds an **interactive reasoning framework** aligned with LRMs’ reasoning behavior, enabling to turn automatic reasoning into **steerable human-LLM collaboration**;
> - Demonstrates **superior accuracy-length trade-offs** across datasets, with flexible deployment and practical value.
>
>
> We sincerely hope these clarifications address your concerns, and we’d greatly appreciate your further discussion or feedback!
>
>
> Sincerely,
>
> The Authors

---

### Author Response · Authors · 2025-11-26

Dear ACS and Reviewers,


We sincerely appreciate your time and effort in reviewing our paper and providing constructive feedback. Besides the response to each reviewer, here we would like to further 1) thank reviewers for their recognition of our work and 2) highlight the major modifications in our revision:

**1. Recognition of Our Work**

We are glad that all reviewers appreciate and recognize our novelty and contribution.


-  **Innovative Framework and Motivation**: "The paper introduces an innovative reasoning-enhanced framework, 'Think-with-Me', designed to address common issues in large reasoning models (LRMs)." [Reviewer **HGp3**]; "The proposed idea is interesting..." [Reviewer **Zmbd**]; "This paper addresses an important issue in optimizing the test-time reasoning efficiency"[Reviewer **7csp**]

- **Solid Experimental Results**: "Experimental results show that Think-with-Me performs excellently across various tasks." [Reviewer **HGp3**]; "The proposed method achieves a significant improvement in compressing reasoning length." [Reviewer **NoWM**]; "The experimental validation is comprehensive, covering a wide range of tasks and comparing with various mainstream baseline models." [Reviewer **HGp3**];"...demonstrates improvements in both performance and reasoning length" [Reviewer **Zmbd**]

-  **Clear Presentation and Scalability**: "The paper is well-written..." [Reviewers **NoWM**, **7csp**, **Zmbd**]; "It provides both LLM-based and human-in-the-loop feedback mechanisms, enhancing the scalability of the approach." [Reviewer **NoWM**]; "The paper provides extremely detailed appendices, enhancing transparency and credibility." [Reviewer **HGp3**]



**2. Major Modifications in Revision**

We summarize the main modifications in our revised paper (highlighted in blue).

- **Clarified Methodological Motivation and Novelty** We have strengthened the motivation and positioning of our approach in the **Introduction and Related Work (Appendix A)**, clearly articulating the core innovation of our method relative to existing external-feedback strategies. [Reviewers NoWM, Zmbd]

- **Expanded and Strengthened Baseline Comparisons** We include a wider range of strong and relevant baselines, such as **s1-32B**, **DEER**, **SEAL**, and **SpecReason**, in **Tables 1 and 2**, ensuring a more comprehensive and fair evaluation. [Reviewers NoWM, Zmbd]

- **Enhanced Motivation via External Feedback Comparison** We incorporate a comparison with **Self-Refine**—a strong baseline that also leverages external feedback (from both the base model itself and Qwen2.5-72B-Instruct)—in **Table 7**, and update the motivation in the **Introduction** accordingly. [Reviewer NoWM]

- **Extended Ablation Studies** We conduct extensive ablation experiments, including: (1) a comparison between **GRPO and SFT** in **Table 4**, and (2) an evaluation of **alternative intervention strategies** in **Table 12**. These analyses validate the contribution of each key component. [Reviewer 7csp]

- **Refined Theoretical Explanation** We provide a more rigorous theoretical rationale in **Proposition 1**, clarifying the working mechanism behind the proposed interactive intervention framework. [Reviewers 7csp, Zmbd]

- **Comprehensive Efficiency and Cost Analysis** We add a detailed breakdown of actual reasoning time and computational costs across all compared methods, reinforcing the practical efficiency of our approach. [Reviewer Zmbd]

- **Expanded Discussion of Limitations and Future Work** We have updated **Appendix J.1 and J.2** to explicitly outline key limitations and promising future research directions, enhancing the rigor and forward-looking impact of the work. [Reviewer HGp3]

Best regards,

The Authors

---

### Author Response · Authors · 2025-12-02

Dear ACs, SACs, and PCs,

Thank you for handling our submission. We sincerely appreciate the effort of all reviewers. We are glad that all the reviewers appreciate and recognize our work: **1) Well motivation and clear presentation** (HGp3, Zmbd, 7csp, NoWM), **2) An innovative framework and novel mechanism** (HGp3, Zmbd, 7csp), **3) Solid and comprehensive experimental results** (HGp3, NoWM, Zmbd).

We summarize the core contributions and rebuttal efforts to assist in your final assessment.

**Core contributions of our work**: We would like to take this opportunity to reiterate the core contributions of our work, which we believe make a significant contribution to the field:
* **A Novel Interactive Intervention Framework for LRM Reasoning Process** We propose **Think-with-Me**, a framework that tackles **overthinking** and **overshoot** of LRMs by **dynamic pausing, redirecting, or terminating** reasoning through external feedback from either LLM proxy or human beings.
* **Intervention Mechanism Inspired by Intrinsic Reasoning Behavior** Think-with-Me is rooted in a systematic analysis of LRM reasoning traces, where we identify ** the semantic role and dual impact on accuracy of transitional conjunctions**. This insight motivated an intervention mechanism with **explicit triggers, content, forms, and termination points** in reasoning.
* **Superior Efficiency-Performance Trade-off with Low Overhead** Comprehensive experiments demonstrate that Think-with-Me achieves a **superior length-accuracy trade-off** compared to strong baselines and the framework exhibits **adaptability across diverse scenarios and tasks**.
* **Enhanced Human-LLM Collaboration for Practical Applications** The framework supports feedback from human beings, **refining the granularity and mode** of **Human-LLM interaction**. It transforms LRMs into **steerable collaborative partners**, broadening the applicability of LRMs beyond reasoning tasks.

**Summary of Rebuttal Efforts and Revisions**: We address all reviewers’ concerns point-by-point through explanations and experimental evidence:
* **Enhanced Method Motivation and Observational Novelty**： We integrate prior observations into an **evidence chain**, providing **new insights** of reasoning behaviors. We enhance motivation by considering the **mismatch between existing external feedback methods and LRM reasoning behavior**, supported by experimental evidence. We update **Introduction** and **Table 7 in Appendix B.1**. [Reviewer NoWM]
*   **Clear Method Distinction & Empirical Validation**: We update the **related work in Appendix A** to clarify that Think-with-Me **avoids time overheads** in multi-model collaboration, **overcomes the signal limitations** of single-model optimization. Further comparisons with strong baselines confirm that our method is **superior accuracy-length trade-off**. We update **Table 1** and **Table 2** in **4.1 comparision experiment**.  [Reviewers Zmbd, NoWM]
*   **Theoretical Validity and Proposition Support**: We provide an **information-theoretic proof for Proposition 1**. Experiment shows that Think-with-Me **reduces reasoning uncertainty in 98.73% of cases**, demonstrating Proposition 1 offers a theoretical foundation for Think-with-Me. We update the clarification in **3.2.1**.[Reviewers Zmbd, 7cSP]
*   **Robustness for Error Feedback**: **GRPO-based training** and **content-agnostic criteria** enhance robustness against erroneous feedback and lower the usage threshold. Details are in **Appendix J.1 and J.2**. [Reviewer HGp3]
*   **Detailed Computational resource and time cost**:Despite the existing reasoning time analysis in Table 3, we report the average **reasoning time and GPU memory costs** for one instance of all methods, confirming the practicality of Think-with-Me. Details are in Appendix F, Table 11. [Reviewer Zmbd]
*   **Extended More Discussion**: We provide additional evidence on Think-with-Me: (1) support for **32k context windows**; (2) benefits of a **summary-before-feedback strategy**; (3) sub-optimal performance of **alternate intervene points**; (4) the superiority of **GRPO over SFT** for our task. Details are in **Appendix H.3 (Table 14), Appendix J, Sec. 4.3 (Table 4), and Appendix G.3 (Table 12)**. [Reviewer 7cSP]

**Conclusion**: While the new ICLR policy prevents further discussion, we are confident that our revisions address all concerns and enhance the clarity of our work.
We believe the proposed interactive reasoning intervention framework, **Think-with-Me**, that has been validated for effectiveness and practicality, represents **an innovation in human-LLM interaction** and provides a **new perspective and inspiration** for solving various tasks **beyond reasoning**.

Thank you for your time and we would be grateful for your fair consideration in the final decision.

Sincerely,

The Authors

---

### Meta-Review · Area_Chair_JYmZ · 2026-01-06

**Summary:**

This paper introduces ``Think-with-Me'', an interactive reasoning framework designed to address inefficiencies in Large Reasoning Models (LRMs), specifically ``overthinking'' (redundant reasoning extensions) and ``overshoot'' (corrupting correct answers through unbounded reasoning). The core mechanism pauses the model's generation at transitional conjunctions (e.g., ``so'', ``wait'') and injects external feedback from an LLM proxy or human to guide continuation, redirection, or termination. The model is fine-tuned via Group Relative Policy Optimization (GRPO) to respond effectively to such interventions. Evaluations on MATH500, AIME24, GPQA, and LiveCodeBench demonstrate improved accuracy-length trade-offs within constrained context windows. This paper would fit well in sessions on efficient inference, reasoning in LLMs, controllable generation, or human-AI collaboration.

Initial scores were split: HGp3 (6), NoWM (4), Zmbd (4), and 7csP (2). The primary criticisms, shared among several reviewers, centered on: (a) weak baseline comparisons that failed to situate the work; (b) a lack of analysis on practical costs like wall-clock time and GPU usage; (c) insufficient methodological justification for using GRPO over SFT; and (d) the theoretical depth of Proposition 1.

**Reviewer Concerns:**

The authors provided an thorough rebuttal that addressed nearly all concerns with new experimental evidence:
*   **Baselines and Costs (NoWM, Zmbd, 7csP)**: The authors added comparisons against a suite of strong competitors and provided a detailed breakdown of inference time and resource usage, directly resolving the most significant weaknesses.
*   **Methodology (7csP)**: New ablations compellingly demonstrated that GRPO outperforms SFT for this task and that the chosen intervention points are more effective than alternatives, validating the core design choices.
*   **Theory (7csP, Zmbd)**: A formal proof for Proposition 1 was provided, alongside empirical data showing entropy reduction in 98.73% of cases, grounding the theory in practice.

**Reviewer Scores:**

I estimate HGp3 would remain positive 6; NoWM and Zmbd would maintain 4 or increase to 6 as their main concerns about baselines and costs were responsed; and 7csP would increase to at least a 4, as their requests for ablations and cost analysis were fully met.

This would result in a lean to reject (6,4,4,4).

---

### Decision · Program_Chairs · 2026-01-26

Reject